# SELFCLEAN:
# A SELF-SUPERVISED DATA CLEANING STRATEGY

## ABSTRACT

Most benchmark datasets for computer vision contain irrelevant images, near duplicates, and label errors. Consequently, model performance on these benchmarks may not be an accurate estimate of generalization capabilities. This is a particularly acute concern in computer vision for medicine where datasets are typically small, stakes are high, and annotation processes are expensive and error-prone. In this paper we ~~propose SELFCLEAN, a general procedure to clean up image datasets exploiting a latent space learned with self-supervision. By relying on self-supervised learning, our approach focuses on intrinsic properties of the data and avoids annotation biases. We~~ formulate dataset cleaning as either a set of ranking problems, which significantly reduce human annotation effort, or a set of scoring problems, which enable fully automated decisions based on score distributions. We then propose SELFCLEAN, a general procedure to identify issues within image datasets which exploits the datasets' intrinsic context through self-supervision. SELFCLEAN consists of a novel thoughtful combination of dataset-specific representation learning and simple distance-based indicators which are surprisingly effective for finding noise without annotation biases. We demonstrate that SELFCLEAN achieves state-of-the-art performance in detecting irrelevant images, near duplicates, and label errors within popular computer vision benchmarks, retrieving both injected synthetic noise and natural contamination. In addition, we apply our method to multiple image datasets and confirm an improvement in evaluation reliability.

## 1 INTRODUCTION

In traditional machine learning (ML), data cleaning was an essential part of the development process since minor contaminations in the dataset, such as irrelevant samples, near duplicates, label issues, and missing values, may significantly impact model performance and robustness (Li et al., 2021). However, with the rise of deep learning (DL) and large-scale datasets, data cleaning has become less crucial as large models have shown to work relatively well even if data quality is limited (Rolnick et al., 2018). Validating and cleaning large datasets is challenging, especially for high-dimensional data such as images, where manual verification is often infeasible. Thus, a lot of research focused on learning from noisy data (Natarajan et al., 2013) rather than fixing quality issues, as the overwhelming benefits of large-scale datasets are believed to exceed the drawback of diminished control. However, the rapid growth of image collections also comes with a sharp decrease in their quality and consistency, especially if collection involves some degree of automation and thus is inherently prone to various errors (Sambasivan et al., 2021). Moreover, for many domains, the size of publicly available datasets is still one of the main limiting factors to the progress of artificial intelligence (AI). In these low-data regimes, the importance of clean data is more pronounced since even fractional amounts of poor-quality samples can substantially hamper performance (Karimi et al., 2020). This is especially relevant in high-stakes settings such as the medical domain, where high-quality data is needed to train robust models and validate their performance. However, many practitioners rather focus on data quantity as a key performance driver and implicitly assume a high-quality collection process (Pezoulas et al., 2019). Thus, even in sensitive domains such as dermatology, many existing datasets are known to contain varying noise levels, which can substantially undermine the progress of learning algorithms (Daneshjou et al., 2023).

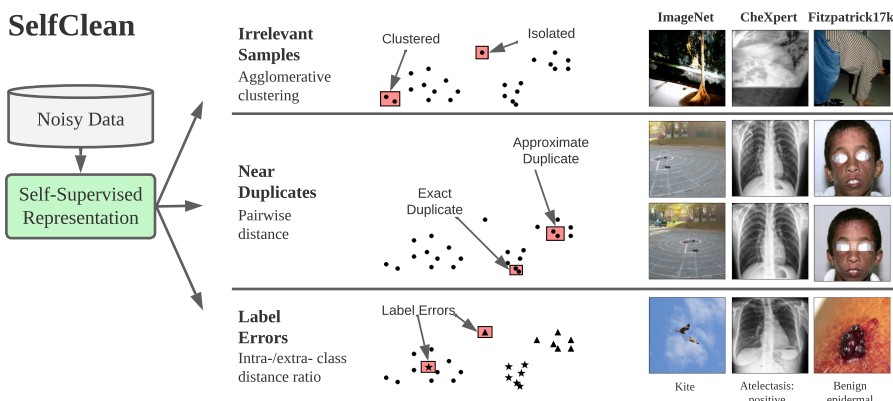

Figure 1: SELFCLEAN first trains a self-supervised encoder on noisy data to obtain latent representations for dataset samples. It then detects irrelevant samples with agglomerative clustering, near duplicates based on pairwise distances, and label errors using the intra-/extra- class distance ratio.

The necessity to report comparable results has led DL practitioners to leverage benchmark datasets despite their underlying issues. For example, ten of the most used benchmark datasets in ML were evaluated and found to have an average label error rate of 3.4% in the evaluation set (Northcutt et al., 2021). Errors in the evaluation and training set undermine the framework by which scientific progress is measured. Contamination in evaluation sets corrupts scores, making it unclear which methods can successfully handle edge cases and obscuring how close performance is to its theoretical optimum. This is especially relevant since many popular benchmarks are saturating, i.e. only saw minor relative changes in performance over the last few years (Ott et al., 2022). Data quality issues in training sets may hinder optimization and produce suboptimal results.

The three types of data quality issues addressed in this paper illustrate these mechanisms well. Irrelevant samples, i.e. inputs unsuitable to carry out valid tasks in the dataset context, add noise to evaluation metrics while slowing down and confusing training. Near duplicates, i.e. different views of the same object, produce arbitrary re-weighting in the evaluation set and reduce variability in the training set. Most importantly, they often introduce leaks between training and evaluation sets that can lead to overoptimistic results. Label errors, i.e. wrongly annotated samples, result in incorrect evaluation and poison the training process. We focus on these three data quality issues because we empirically found them to be frequent in existing benchmark datasets and, at the same time, not straightforward to detect. In the rest of this paper, we use the term "noise" for data quality issues including these three main categories. Notwithstanding the clear benefits of correcting noisy data, applying any automatic cleaning procedure on evaluation sets can be problematic. In fact, the model under evaluation may rely on similar mechanisms to the cleaning approach, invalidating the independence required for a performance estimate. An evaluation that ignores known data leaks is however also incorrect, leading to a conundrum.

This paper proposes a new holistic framework based on self-supervised learning (SSL) to clean multiple errors in image collections, which we call SELFCLEAN and summarize in figure 1. We formulate dataset cleaning as a set of ranking problems, which massively reduce the effort for manual correction, or alternatively as a set of scoring problems, which can be used for fully automatic decisions based on score distributions. We apply our approach to well-known benchmark datasets in computer vision and medical imaging, and discuss implications for reliability of results across these domains. The proposed method enables practitioners to audit data collections, increase evaluation reliability, and amend the training set to improve results. Our work contributes to data-centric ML (Jarrahi et al., 2022) and aims to bolster confidence in both newly-collected and existing datasets. In summary, our main contributions are:

- A new SSL-based data cleaning procedure called SELFCLEAN, which can be used to find irrelevant samples, near duplicates, and label errors by relying exclusively on inductive biases and the dataset itself. The novelty of this procedure consists of a nontrivial and

remarkably competitive combination of dataset-specific representations learned with self-supervision and simple heuristics based on nearest neighbor distance.

- A detailed comparison between the proposed cleaning method and competing approaches on synthetic and natural contamination, including validation against human experts.
- The application of SELFCLEAN to well-known benchmarks in computer vision, medical imaging, and dermatology, and open-source material to clean and correct their errors.
- A practical recommendation to clean development and evaluation splits of benchmark datasets as a reasonable trade-off to obtain more accurate performance estimates.

## 2 RELATED WORK

Data cleaning is a core component of data analytics and a topic of interest in the data management community (Chu et al., 2016). Recently, the data-centric AI initiative also brought it back to the attention of ML researchers, resulting in the development of various data cleaning tools. For instance, Vailoppilly et al. (2021) proposed an all-in-one "data cleansing" tool based on dimensionality reduction, a DL noise classifier, and a denoising model. Open-source tools for data cleaning also started to appear, including CleanLab (Mueller et al., 2018) and CleanVision (Mueller et al., 2022), Lightly (Susmelj et al., 2020), and FastDup (Alush et al., 2022). Most data cleaning approaches require dimensionality reduction to work with high-dimensional data such as images. This includes traditional approaches such as PCA (Maćkiewicz & Ratajczak, 1993) or t-SNE (Maaten & Hinton, 2008), and feature extraction with deep encoders, which are usually trained on natural image databases such as ImageNet (Deng et al., 2009). In the last few years, SSL was shown to learn more representative latent spaces compared to supervised training (Fernandez et al., 2022; Wang & Isola, 2020; Sorscher et al., 2022). Furthermore, Cao & Wu (2021) demonstrated that SSL could learn meaningful latent spaces even with small datasets, low resolution, and small architectures. Inspired by these results, we use SSL as a basis to detect three important types of data quality issues encountered in practice: irrelevant samples, near duplicates, and label errors (Chu et al., 2016). Since these sub-problems are typically addressed separately in the literature, we briefly review them in turn.

The problem of finding irrelevant samples is related to generalized out-of-distribution detection (Yang et al., 2022), in which context it is most similar to outlier detection because the starting point consists of both normal and anomalous samples without additional labels (Aggarwal, 2017). Outlier detection can be addressed with supervised, unsupervised, and semi-supervised learning and was initially motivated by data cleansing, i.e. enabling models to fit the data more smoothly (Boukerche et al., 2020). Supervised detectors, which require full ground truth labels, can identify known outlier types but risk missing unknown ones (Chalapathy et al., 2019; Liang et al., 2020). Unsupervised approaches assume outliers to be located in low-density regions and are based on either reconstruction errors (Hendrycks et al., 2019; Abati et al., 2019), classification (Ruff et al., 2018), or probabilistic methods (Ren et al., 2019). Semi-supervised algorithms can capitalize on partial labels and retain the ability to detect unseen outlier types. Prominent approaches include training on in-distribution data and detecting anomalies that deviate from the dataset representation learned during training (Akcay et al., 2019).

Near-duplicate detection is traditionally based on a representation-match strategy (Lowe, 2004; Ke et al., 2004). Most DL approaches follow a similar method, where feature vectors are extracted by a deep network and utilized as representations for content-based matching (Babenko et al., 2014). Another possible solution is to use Siamese neural networks to learn a similarity metric between samples (Žbontar & LeCun, 2015). A more recent approach was proposed for copy detection and uses a contrastive self-supervised objective along with entropy regularization to promote consistent separation of image descriptions (Pizzi et al., 2022). However, a caveat of this method is that it relies on a threshold that should be manually adapted for each dataset (Oquab et al., 2023).

The identification of label errors is generally focused on prediction-label agreement via confusion matrices and proceeds by removing samples with low recognition rate (Chen et al., 2019) or parts of the minority classes (Zhang et al., 2020). There are exceptions, such as recent approaches based on supervised contrastive learning for label error correction (Huang et al., 2022; Lee et al., 2021). Another prominent method is confident learning, which identifies label errors based on noisy data pruning, using probabilistic thresholds to estimate noise and ranking examples to train with confidence (Northcutt et al., 2022).

## 3 METHODOLOGY

Let $\mathbb{X} = \{(\boldsymbol{x}_i, l_i) \mid i \in \mathbb{I}\}$ be an image classification dataset to be cleaned, where $\mathbb{I} = \{1, \ldots, N\}$ is the index set, $l_i \in \{1, \ldots, L\}$ is the $i$-th label, and $\boldsymbol{x}_i \in \mathbb{R}^{h \times w \times c}$ is the $i$-th sample with height $h$, width $w$, and channels $c$.

**Representation learning.** We train a deep feature extractor $f$ with parameters $\theta$ on the dataset $\mathbb{X}$ using self-supervised learning (SSL), ~~which produces meaningful latent spaces by training features to be invariant to augmentations~~ which learns useful representations by solving pretext tasks. Furthermore, SSL does not suffer from semantic collapse like supervised learning, which removes any information unnecessary for class assignment (Fernandez et al., 2022). Note that SSL is always performed on the entire dataset without pre-filtering data quality issues. The hypothesis is that these dataset-specific representations based on invariance inductive bias are exceptionally useful for dataset cleaning, to the point that the following identification of issues can be carried out by appropriate, almost surprisingly simple methods. Let $\boldsymbol{e}_i = f(\boldsymbol{x}_i; \theta) \in \mathbb{R}^D$ be the representation of sample $\boldsymbol{x}_i$ obtained with $f$, where $D$ denotes the latent dimension.

In this work, we consider pre-trained features from SimCLR (Chen et al., 2020) and DINO (Caron et al., 2021), which have been shown to produce meaningful latent spaces (Fernandez et al., 2022; Wang & Isola, 2020), but in principle any SSL method can be used. SimCLR is a popular discriminative SSL approach, that uses a contrastive loss to compare different views of the same image against other randomly sampled images and their transformations. DINO instead belongs to the distillation SSL family and trains a student network to match the outputs of a teacher network on different views of the same image. SELFCLEAN requires to pre-train a dataset-specific encoder with SSL, but after cleaning this model can also be used as a starting point for downstream tasks. In appendix F.1, we compare the performance of such a dataset-specific encoder against general-purpose features. For simplicity we use vision transformers (ViTs) throughout our experiments as detailed in Appendix B.

As feature normalization is built into the SimCLR training objective, it is natural to compare points in its latent space using cosine similarity, $\text{sim}(\boldsymbol{u}, \boldsymbol{v}) = \boldsymbol{u}^\top \boldsymbol{v} / (||\boldsymbol{u}||_2 ||\boldsymbol{v}||_2)$, and the associated distance scaled to $[0, 1]$, $\text{dist}(\boldsymbol{u}, \boldsymbol{v}) = (1 - \text{sim}(\boldsymbol{u}, \boldsymbol{v}))/2$. To facilitate comparison, we explicitly include $L_2$ normalization during training and inference for DINO, such that its latent space is a unit hypersphere of dimension $D - 1$. In appendix F.2, we present an ablation study of the $L_2$ normalization and investigate the influence of the distance function.

**Irrelevant samples.** We define samples to be irrelevant when they are unsuitable for a task that is valid in the dataset context. For example, irrelevant samples could be images from different modalities or without any object of interest. Our irrelevant sample ranking is induced by agglomerative clustering with single linkage (Gower & Ross, 1969) in representation space. The idea is that the later a cluster is merged with a larger one, the more it can be considered an outlier (Jiang et al., 2001). The ranking is given by sorting the clustering dendrogram such that, at each merge, the elements of the cluster with fewer leaves appear first. When a merge occurs among sub-clusters with the same number of elements, we sort the sub-cluster created at the larger distance first, and we allow for ties when only leaves are involved. Finally, for fully automatic determination of irrelevant samples, we also associate each sample with a numerical score, which takes small values for abnormal instances and is compatible with our ranking. In appendix J, we construct such a score $s_{\text{is}}(\boldsymbol{e}_i)$ starting from the idea that merges, which happen at very different distances or between clusters of very different sizes, should produce large numerical variations.

**Near duplicates.** We define near duplicates as pairs of images that contain different views of the same object. In this sense, exact duplicates are a special case of near duplicates. We rank potential near duplicates by sorting each pair of distinct samples $(i, j), i \neq j$ in ascending order according to the distance between their representations in the latent space, $s_{\text{nd}}(\boldsymbol{e}_i, \boldsymbol{e}_j) = \text{dist}(\boldsymbol{e}_i, \boldsymbol{e}_j)$. Thus, the first pair in the ranking has the smallest cosine distance compared to every other pair in the dataset.

**Label errors.** Label errors are samples annotated with a wrong class label. We rank potential label errors by sorting samples in ascending order according to their intra-/extra- class distance ratio (Ho & Basu, 2002). This is a ratio that, for an anchor point $\boldsymbol{e}_i$, compares the distances to the nearest representation of a different label $m_{\neq}(\boldsymbol{e}_i)$ and the distance to the nearest representation of the same

label $m_=(\boldsymbol{e}_i)$:

$$m_=(\boldsymbol{e}_i) = \min_{j \in \mathbb{I}} \left[ \delta_{l_i l_j} \cdot \mathrm{dist}(\boldsymbol{e}_i, \boldsymbol{e}_j) \right], \qquad s_{\mathrm{le}}(\boldsymbol{e}_i) = \frac{m_{\neq}^2(\boldsymbol{e}_i)}{m_=^2(\boldsymbol{e}_i) + m_{\neq}^2(\boldsymbol{e}_i)}, \qquad (1)$$
$$m_{\neq}(\boldsymbol{e}_i) = \min_{j \in \mathbb{I}} \left[ (1 - \delta_{l_i l_j}) \cdot \mathrm{dist}(\boldsymbol{e}_i, \boldsymbol{e}_j) \right],$$

where $\delta_{l_i l_j}$ is the indicator function that evaluates to 1 if, and only if, $l_i = l_j$. The score $s_{\mathrm{le}}$ is bound between $[0, 1]$ where label errors ought to be scored lower than true labels.

Note that, in all three cases, SELFCLEAN leverages the local structure of the embedding space: cluster distances are computed only using the closest samples during agglomeration for irrelevant samples, near-duplicates are identified among sample pairs with the smallest distances, and finding label errors only considers the nearest examples of the same and a different class.

## 3.1 Operation modes

The criteria above rank and score candidate issues, but do not specify which ones are inferred to be actual issues. This can be achieved with two operating modes: human-in-the-loop or fully automatic.

In the human-in-the-loop mode, ranking is essential as an exhaustive manual search is often infeasible in practice, especially when considering pairwise relationships such as near duplicates. A human curator can then inspect issue-enriched data subsets, either confirming and correcting problems or looking for a specific rank threshold that gives the desired balance between precision and recall.

To perform automatic cleaning, specifying a fraction of data to drop or correct without a detailed inspection is suboptimal, as sensitivity to this parameter is expected to be very high. In this work, we make and empirically verify the hypothesis that (pairs of) samples can be scored with functions based on the embedding space distance, which produce a smooth distribution for clean samples, and relegate contaminated ones to significantly lower score values. Depending on the contaminated data distribution, it may then be possible to isolate problematic samples with statistical arguments based on two robust hyperparameters, namely the contamination rate guess $\alpha$ and the significance level $q$, as detailed in appendix H. In short, we first use a logit transformation on scores to induce a separation gap between normal and problematic samples. We then set an upper bound for the left tail of the score distribution using a logistic functional form, and we estimate its parameters using quantiles. Afterward, we identify outliers based on their violation of the upper probability bound.

## 4 Experimental setup

**Datasets.** We experiment on a total of ten datasets described in appendix D, including

- 3 large-scale vision benchmarks: ImageNet (Deng et al., 2009), CelebA (Liu et al., 2015), and Food-101N (Bossard et al., 2014; Lee et al., 2018);

- 2 general medical datasets of X-rays and histopathological images: CheXpert (Irvin et al., 2019) and PatchCamelyon (Veeling et al., 2018; Ehteshami Bejnordi et al., 2017);

- 5 dermatology datasets including dermatoscopy and clinical images: HAM10000 (Tschandl et al., 2018), ISIC-2019 (Tschandl et al., 2018; Combalia et al., 2019), Fitzpatrick17k (Groh et al., 2021), DDI (Daneshjou et al., 2022), and PAD-UFES-20 (Pacheco et al., 2020).

**Synthetic experiment setup.** To compare SELFCLEAN against other methods, we create synthetic datasets by altering clean benchmarks for dermatology and initially consider each of the three data quality issues separately. We start with two dermatology datasets which can be assumed to be of high quality, since their size is relatively small and were exhaustively curated by multiple experts: High-Quality Fitzpatrick17k (HQ-FST) (Groh et al., 2021) and DDI (Daneshjou et al., 2022). The generation of synthetic contamination is inspired by typical noise present in the medical domain. For each of the three noise types, we define two distinct contamination strategies applied to both datasets, resulting in a total of twelve evaluations. These evaluations are then repeated for different percentages of contamination, i.e. 5% and 10%, mimicking real-world noise prevalence estimates (Han et al., 2022). For each noise category, we compare against other unsupervised methods that have performed well on the given task. A detailed description of these competing approaches can be

found in appendix E. SELFCLEAN employs self-supervised pre-training carried out on the contaminated dataset. Specifically, we train a separate model for every noise category, contamination level, and synthetic contamination strategy.

The first synthetic contamination strategy for irrelevant samples, *XR*, consists of adding images of a different modality, in our case lung X-rays of COVID-19 patients (Ozturk et al., 2020). The second strategy for irrelevant samples, *CMED*, adds a number of samples from disparate datasets, consisting of surgical tools (Lavado, 2018), X-ray images from arbitrary body parts (Ozturk et al., 2020), ImageNet samples (Deng et al., 2009), histopathological images (Spanhol et al., 2016), segmentation masks (Codella et al., 2019), and pictures of PowerPoint slides[1] (Araujo et al., 2016). Note that *XR* produces clustered outliers and *CMED* more isolated ones. The first contamination strategy for near duplicates, called *MED*, adds samples from the original dataset after augmenting them with rotation, flipping, resizing, padding, and Gaussian blur. The second approach for near duplicates, *ARTE*, consists of adding samples from the original dataset after including artifacts such as watermarks, color bars, and rulers, followed by scaling and composition with other images to create a collage. For the first contamination strategy to evaluate label errors, named *LBL*, we take a predefined percentage of samples of the original dataset and randomly change their labels, chosen uniformly from all others. The second, more challenging strategy to evaluate label errors, *LBLC*, consists of changing a predefined fraction of labels with values randomly extracted according to class prevalence in the original dataset. Detailed configurations and sample augmentations can be found in appendix L.

Different contamination strategies can be applied sequentially to create a dataset with more challenging and realistic constellation of artificial data quality issues, resulting in a so-called "mixed-contamination strategy". We include such a scenario by choosing the first strategy for each of the three considered data problems. In order to consider all interactions, we start with adding irrelevant samples, proceed by creating near duplicates, and finally introduce label errors. Thus, there is the possibility, for example, that an irrelevant sample is further augmented, resulting in near-duplicates of an irrelevant sample. To preserve the meaning of the overall contamination rate $C$, each contamination in the sequence is added with prevalence $C_S$ such that $(1 + C_S)^S = (1 + C)$, where $S$ is the number of contamination steps.

**Natural experiment setup.** To further validate the proposed approach, we evaluate it on data quality issues naturally found in benchmark datasets. To this end, we devise two different experiments. In the first experiment, we measure how well the ranking matches available metadata, e.g. if two images show the same person or if the label was obtained using gold-standard tools in medical diagnosis. This experiment is, however, specific to each dataset depending on the applicability of the available metadata. Therefore, in a second experiment, we use SELFCLEAN to propose a ranking for some datasets and gather partial human annotations for validation. We collect annotations from medical experts for the medical datasets and rely on crowd workers for general image datasets. We then evaluate the proposed ranking using human annotations as described in appendix I.

## 5 RESULTS

### 5.1 SYNTHETIC CONTAMINATION

**Comparison on data quality issues.** Table 11 shows the results of the best two competing methods compared to SELFCLEAN using supervised ImageNet (IN), SimCLR and DINO pre-training. Here performance is reported for the twelve synthetic datasets based on two dermatology benchmarks described in section 4 with a contamination rate of 10%; Table 4 in appendix G.1 includes results for all competing approaches for both 5% and 10% contamination. SELFCLEAN with DINO pre-training performs roughly on par or better than the considered competing methods for irrelevant-sample, near-duplicate, and label-error detection. Some competing approaches for irrelevant sample detection perform better on clustered outliers and worse on isolated outliers or vice versa. In contrast, our method is able to detect both clustered and isolated ones. SELFCLEAN with supervised ImageNet features performs surprisingly well for irrelevant sample detection. On the other hand, SimCLR does not seem competitive compared to the other two pre-training strategies. This could be due to the small dataset size, which may not be sufficient for SimCLR to be effective. In particular, the batch size cannot be large, which is crucial for the local contrastive loss over sampled mini-batch data.

---

[1]Common contamination since medical datasets are often crawled from existing PowerPoint collections.

Table 1: Performance of various models on the detection of synthetic data quality issues. Evaluation is performed for each of the three considered issue types across two clean benchmark datasets, DDI and High-Quality Fitzpatrick17k (HQ-FST), augmented with two strategies for 10% synthetic contamination each (XR, CMED, MED, ARTE, LBL, and LBLC). Consult section 4 for more details on the contamination, appendix E for details on competing approaches, and table 4 for more results.

| | | DDI + XR | | DDI + CMED | | HQ-FST + XR | | HQ-FST + CMED | |
|---|---|---|---|---|---|---|---|---|---|
| Irrelevant Samples | Method | AUROC (%) | AP (%) | AUROC (%) | AP (%) | AUROC (%) | AP (%) | AUROC (%) | AP (%) |
| | IN + IForest | 89.5 | 37.0 | 89.6 | 52.3 | 88.1 | 37.2 | 96.3 | 74.8 |
| | IN + HBOS | 92.5 | 46.7 | 80.7 | 42.9 | 84.3 | 27.3 | 94.0 | 69.7 |
| | SELFCLEAN (IN) | **100.0** | **100.0** | **100.0** | **99.8** | 99.2 | 80.3 | **99.7** | **96.7** |
| | SELFCLEAN (SimCLR) | 90.8 | 32.8 | 77.8 | 31.6 | 95.5 | 48.1 | 81.4 | 26.0 |
| | SELFCLEAN (DINO) | **100.0** | **100.0** | 97.4 | 89.5 | **100.0** | **100.0** | 98.5 | 84.1 |

| | | DDI + MED | | DDI + ARTE | | HQ-FST + MED | | HQ-FST + ARTE | |
|---|---|---|---|---|---|---|---|---|---|
| Near Duplicates | Method | AUROC (%) | AP (%) | AUROC (%) | AP (%) | AUROC (%) | AP (%) | AUROC (%) | AP (%) |
| | pHashing | 69.2 | 9.8 | 68.9 | 33.4 | 71.6 | 12.5 | 80.5 | 30.6 |
| | SSIM | 85.7 | 25.7 | 84.2 | 35.3 | 92.6 | 36.1 | 88.5 | 32.9 |
| | SELFCLEAN (IN) | 96.7 | 6.0 | 94.9 | 29.2 | 97.8 | 27.9 | 93.3 | 27.2 |
| | SELFCLEAN (SimCLR) | 96.1 | 13.7 | 88.6 | 37.5 | 97.4 | 22.4 | 87.2 | 17.3 |
| | SELFCLEAN (DINO) | **99.6** | **63.0** | **99.6** | **65.4** | **99.3** | **61.2** | **95.3** | **38.9** |

| | | DDI + LBL | | DDI + LBLC | | HQ-FST + LBL | | HQ-FST + LBLC | |
|---|---|---|---|---|---|---|---|---|---|
| Label Errors | Method | AUROC (%) | AP (%) | AUROC (%) | AP (%) | AUROC (%) | AP (%) | AUROC (%) | AP (%) |
| | IN + Confident Learning | 71.4 | 17.8 | 72.0 | 23.6 | 77.0 | 31.7 | 83.8 | **38.6** |
| | FastDup | 68.6 | 14.6 | 71.4 | 28.8 | **85.2** | 32.3 | **84.7** | 28.9 |
| | SELFCLEAN (IN) | 60.3 | 22.9 | 43.8 | 16.7 | 48.5 | 10.7 | 49.1 | 12.8 |
| | SELFCLEAN (SimCLR) | 39.6 | 8.0 | 42.8 | 16.0 | 56.3 | 17.7 | 47.2 | 13.2 |
| | SELFCLEAN (DINO) | **77.2** | **26.3** | **78.7** | **50.0** | 85.1 | **39.7** | 79.9 | 35.2 |

**Influence of contamination rate.** Figure 2 illustrates the influence of the contamination rate for SELFCLEAN and the best two competing models for each contamination category, i.e. IForest and HBOS for irrelevant samples, SSIM and pHASH for near duplicates, and FastDup and confident learning for label errors. Wherever appropriate we compare performance using both supervised ImageNet (IN) and self-supervised DINO features. Central value and error bars are obtained from training with three different random initializations and synthetic datasets. Note that this experiment is run on the mixed-contamination dataset, which includes interaction effects. Again, our method outperforms most competing approaches for the three noise types over various contamination rates. The difference is apparent for irrelevant samples and near duplicates, where SELFCLEAN scores on average 84.0% and 19.3% higher in AP than the best alternative for the considered contamination rates. In the case of label errors, confident learning performs well for lower contamination rates, whereas FastDup works better for higher ones. SELFCLEAN is a good compromise throughout and results, on average, in 11.7% higher AP than FastDup.

Figure 2 also elucidates the interplay between the two core components of SELFCLEAN, i.e. representation learning and the three ranking criteria for candidate problems. A sizeable improvement is obtained already when using our ranking criteria with ImageNet features, and an additional considerable benefit is obtained by switching to the dataset-specific self-supervised image encoder. This holds across all three plots, although it is less clear for label errors, where the other methods challenge SELFCLEAN for high contamination rates. Note that tuned SSL features do not deliver the same benefit for alternative approaches. These results indicate that both pre-training and ranking criteria significantly contribute to SELFCLEAN's success.

## 5.2 NATURAL CONTAMINATION

**Comparison with metadata.** We validate our label error ranking in a more realistic setting using annotations for this type of issue available in the literature, specifically Northcutt et al. (2021) which verified 5,440 samples of ImageNet's validation set and Lee et al. (2018) 57,608 of Food-101N. Performance is relatively poor for ImageNet with AUROC = 67.3% and AP = 8.4%, and better for Food-101N where AUROC = 79.8% and AP = 47.8% indicate that verified label errors are identified well. However compared against competitive approaches SELFCLEAN achieves almost double the performance in terms of AP for both datasets. Additionally, we evaluate near-duplicate detection

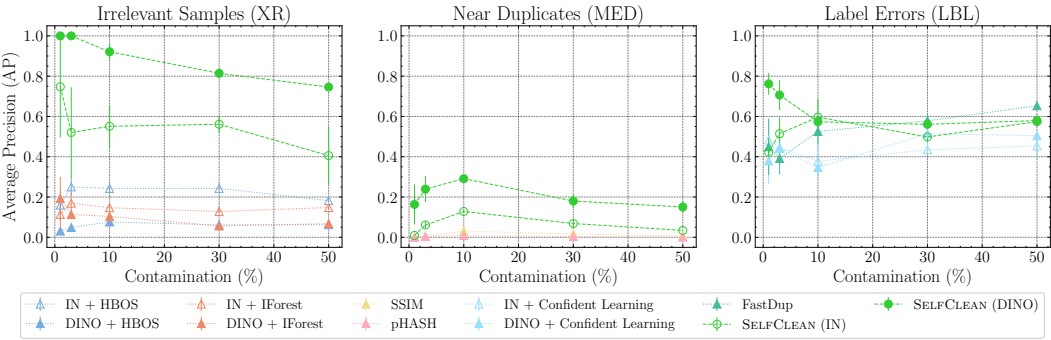

Figure 2: Performance of the best two competing approaches for each noise type compared to SELF-CLEAN measured on two different representations in terms of AP for a "mixed-contamination strategy" when varying the contamination rate. The artificial dataset is created from DDI by adding X-ray images (XR), then injecting augmented duplicates (MED), and finally changing labels at random (LBL).

against CelebA labels that indicate images of the same celebrity. Here we achieve an AUROC = 78.8% and AP = 30.9%, which is remarkably high considering that the model effectively performs facial recognition without supervision. For medical datasets, we first check how well SELFCLEAN can find pairs of images showing the same skin lesion. We obtain good results for HAM10000 and ISIC-2019, with AUROC = 98.7%, AP = 28.4% and AUROC = 98.2%, AP = 26.6% respectively. On the other hand, for PAD-UFES-20 we only achieve AUROC = 71.0% and AP = 10.0%, further investigated in appendix G.2 and likely caused by faulty metadata. We also attempt to identify X-rays from the same patient within CheXpert and find some agreement with AUROC = 70.5% and AP = 7.5%, suggesting again that a case-by-case analysis should be performed. Furthermore, we investigate again the correspondence between our label error ranking and the labeling method. For PAD-UFES-20, we observe that gold-standard labeling does not correlate very well with our label error ranking (AUROC = 57.0%, AP = 61.3%). On the other hand, for the HAM10000 dataset, we observe that the labels obtained from follow-up, which are arguably very accurate since the considered nevi did not show any change during three follow-up visits or 1.5 years (Tschandl et al., 2018), are found by SELFCLEAN to be unlikely label errors (AUROC = 90.8%, AP = 86.3%). A table with results can be found in appendix G.2.

**Comparison with human annotators.** We evaluate SELFCLEAN rankings based on human annotations across two medical and two common vision benchmarks as described in appendix I. The evaluation reveals that images ranked by SELFCLEAN as the most likely to contain data quality issues are also identified by human experts as problematic significantly more often ~~than random images~~ than an uninformed selection of images. ~~As shown in table~~ 12, we find 95% significant differences in nine of twelve evaluations for comparing the lowest 50 ranked images to a random selection and six of ten evaluations for comparing images ranked 1-25 to images ranked 26-50. More precisely, we compare the number of confirmed issues in the lowest 50 ranked images and in 50 images extracted at random, and find 95% significant differences in nine of twelve cases as shown in table 12. We then repeat the comparison for images ranked 1-25 against images ranked 26-50, and observe significance for six out of ten evaluations. Two cases in the second comparison are excluded because too many positive samples give undefined metrics. Results indicate that the proposed ranking, to a large degree, coincides with human understanding of these three noise types. Therefore using SELFCLEAN can increase efficiency when analyzing and fixing data quality issues.

## 6 DISCUSSION

**Application to benchmark datasets.** We now apply the fully automatic mode of SELFCLEAN to well-known image benchmark datasets and estimate the prevalence of data quality issues. The contamination rate guess $\alpha = 0.10$ and significance level $q = 0.05$ are conservative choices based on general noise level estimates and the study in appendix H. For highly curated datasets involving extensive manual verification, such as DDI, PAD-UFES-20, HAM10000, CheXpert, and ImageNet-

Table 2: Influence of removing samples detected in the automatic cleaning mode with $\alpha = 0.10$ and $q = 0.05$ on downstream tasks. We report macro-averaged F1 scores for linear and $k$NN classifiers on DINO features over 100 random training/evaluation splits with 80% and 20% fractions respectively. We compute paired performance differences before and after cleaning the evaluation set, and before and after cleaning also the training set. We report the median and the intervals to the 5% (subscript) and 95% (superscript) percentiles. Additionally, we indicate significance of a paired permutation test on the difference sign with $^*p < 0.05$, $^{**}p < 0.01$, and $^{***}p < 0.001$.

| | $\Delta$ $k$**NN Classifier (%)** | | $\Delta$ **Linear Classifier (%)** | |
|---|---|---|---|---|
| **Dataset** | Clean Eval | Clean Train | Clean Eval | Clean Train |
| DDI | $+1.2^{+1.9}_{-1.2}$ *** | $+0.0^{+1.7}_{-1.4}$ *** | $+1.0^{+11.1}_{-11.2}$ | $-0.7^{+7.7}_{-10.8}$ |
| HAM10000 | $+0.2^{+0.5}_{-0.4}$ *** | $+0.2^{+1.3}_{-0.8}$ ** | $+0.1^{+3.2}_{-3.5}$ | $-0.1^{+3.9}_{-3.6}$ |
| Fitzpatrick17k | $-4.1^{+1.2}_{-1.3}$ *** | $+0.1^{+2.0}_{-1.7}$ | $-0.6^{+2.9}_{-3.6}$ ** | $+0.2^{+3.3}_{-3.9}$ * |
| Food-101N | $+0.1^{+0.1}_{-0.1}$ *** | $+0.1^{+0.2}_{-0.2}$ *** | $+0.2^{+0.6}_{-0.5}$ *** | $+0.1^{+0.6}_{-0.5}$ ** |
| ImageNet-1k | $-0.4^{+0.1}_{-0.2}$ *** | $+0.4^{+0.3}_{-0.4}$ *** | $-0.4^{+0.6}_{-0.6}$ *** | $-0.0^{+0.9}_{-0.5}$ |

1k, we find noise levels below 1%. However, for ISIC-2019 and PatchCamelyon, we estimate 5.4% and 3.9% of near duplicates not accounted for in the metadata. Such errors should be addressed to regain confidence in the obtained scores. On the other hand, when considering datasets involving less manual curation, such as Fitzpatrick17k, CelebA, and Food-101N, we find less than 1% of irrelevant samples, approximately 14.8%, 0.4%, and 1.4% near duplicates, and 0.6%, 0.5%, and 0.9% label errors, respectively. The abundance of near duplicates in these benchmarks can often be traced back to crawled data of different pages using the same illustration or thumbnail images. Since some datasets, such as Fitzpatrick17k, do not have fixed data splits considering these near duplicates, most random splits suffer from data leaks. Thus, models evaluated on these benchmarks are optimistically biased. Details can be found in appendix G.5.

**Influence of dataset cleaning.** To better understand the relevance of data cleaning, we examine the impact of using our method on performance estimates for downstream tasks in table 2. We train linear and $k$NN classifiers based on our encoder on multiple classification benchmarks and measure the performance difference in F1 score when cleaning either the evaluation set only or both splits. We use the automatic mode of SELFCLEAN with contamination rate guess $\alpha = 0.10$ and significance level $q = 0.05$. For most benchmark datasets, cleaning the evaluation set and removing data leaks significantly alters scores. This is most apparent for datasets involving less curation, e.g. obtained from web crawling. Cleaning the training set has a significant positive impact for many benchmarks, indicating that noise in the training set hindered optimization. Details are reported in appendix G.4.

**Recommended use.** Our method determines context based on the dataset rather than a specific task, so the candidates it provides for correction may represent desired features (e.g. rare diseases, longitudinal studies, or multiple object views). By no means should the identification of a data quality issue be automatically considered a suggestion to remove it. Instead, discovering relationships among samples is always an advantage, as it can inform proper action. While undesirable behavior may occur with the automatic mode of SELFCLEAN, this is similar to other methods such as supervised learning when applied without control, and such biases can be mitigated with the human-in-the-loop approach.

The conflict between resolving data quality issues and the veto against the examination of evaluation data, mentioned in the introduction, has no easy resolution. We suggest the following compromise, which avoids obvious pitfalls, as an improvement to the current situation. To refine performance estimates for a benchmark dataset, an SSL model can be trained on the training set. SELFCLEAN can then be used to clean both training and evaluation sets, where for the latter the human-in-the-loop mode is required and label errors should not be considered. The number of problems found for each set separately and across them for near duplicates should be reported. Note that even with human confirmation and refraining from the correction of label errors, the cleaning procedure introduces some degree of bias due to the sampling of the candidate issues to be confirmed. We believe that in many practical cases, the benefit of data cleaning outweighs this bias. ~~This procedure further~~

~~enables~~ Incidentally, the procedure described allows using the SSL pre-trained backbone as starting point for dataset tasks.

**Limitations.** SELFCLEAN hinges on the considered dataset and inherits biases from its intrinsic composition. For example, given an image collection with a minority group that can be easily distinguished from the rest, the minority samples could be ~~classified as~~ suggested to be irrelevant. This risk is studied in a limited number of cases in Appendix G.3. ~~Likewise, our method identifies context based on the dataset rather than a specific task, so the candidates it provides for correction may represent desired features (e.g. rare diseases). While undesirable behavior may occur with the automatic mode of SELFCLEAN, this is similar to other methods such as supervised learning when applied without control, and such biases can be mitigated with the human-in-the-loop approach.~~ Additional noise types such as ambiguous labels may be integrated in our framework in future work. Furthermore, the current formulation of SELFCLEAN does not scale well with dataset size, which potentially could be improved with approximation methods or by relying on the distribution of the shortest distance to any data point. Also, no case-specific tuning was carried out for hyperparameters or augmentations after ensuring convergence for small datasets. However, we expect improved representations to positively influence the performance of SELFCLEAN. Likewise, the relationship between the granularity of concepts learned by SSL (Cole et al., 2022), the difficulty in finding data quality issues, and the performance of SELFCLEAN was not explored. We however stress that identifying issues even as simple as the ones we simulate in synthetic experiments is valuable, as they occur in practice as illustrated in Appendix M. Finally, while SSL strategies are readily available for other data modalities, we focused on a thorough analysis for images.

## 7    CONCLUSION AND OUTLOOK

This paper proposed a novel data-cleaning strategy called SELFCLEAN, based on self-supervised learning. Our method can be used to detect irrelevant samples, near duplicates, and label errors either automatically or in collaboration with humans. We compared our proposal to state-of-the-art methods across multiple general and medical image benchmarks in both synthetic and natural contamination settings. Results showed that SELFCLEAN outperformed the considered competing approaches for synthetic data quality issues and demonstrated very good correspondence to metadata and expert annotations in natural settings. Furthermore, our methodology has notably advanced the state-of-the-art in label error detection, achieving a twofold increase in AP over existing approaches on known ImageNet-1k and Food-101N errors. Moreover, applying the cleaning strategy to highly-curated medical datasets and general vision benchmarks revealed multiple data quality issues with significant impact on model scores. By correcting these data collections, confidence can be regained in reported benchmark performances. In the future, we plan to incorporate SELFCLEAN in annotation processes for higher data quality and use it to detect problems during inference to enhance model robustness.

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

## A    BROADER IMPACT

SELFCLEAN is a new data-cleaning procedure that can be applied to any visual data collection. The procedure relies on SSL and therefore does not inherit annotation bias. Practitioners can choose if they want the cleaning process to happen fully automatically or with human intervention. Gaining insights into data collections of unknown quality can lead to the production of more reliable benchmarks and, in turn, results in performance measurements that are more meaningful and translate better to real-world performance. Moreover, reported benchmark results can be invalidated if they contain substantial contamination. SELFCLEAN is therefore a valuable tool to clarify which methods are the most valuable and to steer future research directions, as well as to accelerate AI applications.

The near-duplicate detection component of our approach could be used for person re-identification and data de-anonymization, even if it was not designed for this purpose. Although new in peer-reviewed publications and lacking extensive evaluation for data cleaning to the best of our knowledge, it can already be found in at least one publicly available tool (Susmelj et al., 2020). We therefore conclude that increased awareness outweighs increased chances of malignant use.

For all experiments we used about 5400 GPU hours which roughly corresponds to 600kg $CO_2$.

## B    TRAINING DETAILS

We use a randomly initialized ViT tiny with a patch size of $16 \times 16$ as encoder (Dosovitskiy et al., 2021) for all experiments. Additionally, we compare against the same ViT supervised pre-trained on ImageNet. The latent representation is given by the class token output from the encoder's last layer, which has dimension 192 for a ViT tiny. The self-supervised pre-training follows the introductory papers of the respective methods (Chen et al., 2020; Caron et al., 2021). All SSL models were pre-trained for 100 epochs with only minor manual hyperparameter tuning to ensure proper convergence. Since we also consider small datasets, we increase the strength of augmentations to train with samples of sufficient variation. Details on hyperparameters can be found in table 13 of the appendix. We resize images to $224 \times 224$ pixels and normalize them using the mean and standard deviation of ImageNet (Deng et al., 2009). The implementation is based on PyTorch 1.9 (Paszke et al., 2019), and experiments are performed on an Nvidia DGX station, which features eight V100 GPUs, each with 32 GB of memory, 512 GB of system memory, and 40 CPU cores. Code and models for replicating our results can be found at `https://github.com/***/***`.

## C    EVALUATION METRICS

Our evaluation relies on ranking metrics as the ranking constitutes the core of SELFCLEAN independently of the operation mode. The proposed method and the competing approaches are therefore evaluated in terms of AUROC, AP, Recall@$k$, and Precision@$k$, following standard practice (Rendle, 2019). The area under the ROC curve (AUROC) measures the likelihood that a randomly relevant item is ranked higher than a randomly irrelevant item. Average precision (AP) at $k$ measures the precision at all ranks that hold a relevant item. Precision@$k$ measures the fraction of relevant items among the top $k$ predicted items, and Recall@$k$ measures the fraction of all relevant items recovered in the top $k$.

## D    DATASETS

In this study, we selected ten well-known open-source image datasets comprising three general-purpose vision benchmarks and seven medical datasets. The selection of datasets contains different modalities such as smartphone, high-resolution, X-ray, histopathology, dermatoscopy, and clinical images. The diversity of the datasets and domains should illustrate the proposed approach's versatility. Furthermore, some datasets were chosen because of their very high quality standards, and their curation involved large amounts of manual correction including validation by multiple domain experts. These high-quality datasets are mainly used to evaluate the proposed approach after injecting synthetic noise since they are assumed to be almost free of natural contamination.

**Diverse Dermatology Images (DDI)** is a public, deeply-curated, and pathologically-confirmed image dataset with diverse skin tones (Daneshjou et al., 2022). It contains 656 clinical images of 570 unique patients with 78 common and uncommon diseases originating from pathology reports of the Stanford Clinics.

**PAD-UFES-20** is a public benchmark dataset composed of clinical images collected from smartphone devices including patient clinical data (Pacheco et al., 2020). The dataset comprises 1,373 patients, 1,641 skin lesions, and 2,298 images for six different diagnoses: three skin diseases and three skin cancers.

**HAM10000** is a public benchmark dataset consisting of 10,015 dermatoscopic images collected from different populations and institutions (Tschandl et al., 2018). The collected cases include a representative sample of seven categories of pigmented lesions. Images were collected with the approval of the Ethics Review Committee of the University of Queensland (Protocol-No. 2017001223) and the Medical University of Vienna (Protocol-No. 1804/2017).

**Fitzpatrick17k (FST)** is a public benchmark dataset containing 16,577 clinical images with skin condition annotations and skin type labels based on the Fitzpatrick scoring system (Groh et al., 2021). The images originate from two online dermatology atlases and thus are known to contain noise (Daneshjou et al., 2023). In this study, we used the middle granularity level, which partitions the labels into nine disease categories.

**High-Quality Fitzpatrick17k (HQ-FST)** is a subset of the Fitzpatrick17k dataset used in the paper (Groh et al., 2021) as a data quality check. It was obtained by randomly selecting 3% of the images (504 samples) and gathering annotations by two board-certified dermatologists to evaluate diagnostic accuracy. This subset is assumed to be of much higher quality than its original, larger counterpart.

**ISIC-2019** is a public benchmark dataset of 25,331 dermoscopic images with metadata split into eight diagnostic categories. Additionally, the test set contains an additional outlier class not represented in the training data. The images originate from the HAM10000 (Tschandl et al., 2018) and the BCN_20000 (Combalia et al., 2019) datasets.

**PatchCamelyon** consists of 327,680 color image patches extracted from histopathologic scans of lymph node sections (Veeling et al., 2018) from the Camelyon16 dataset (Ehteshami Bejnordi et al., 2017). Each patch is annotated with a binary label indicating the presence of metastatic tissue. Camelyon16 contains 399 whole-slide images and corresponding glass slides of sentinel axillary lymph nodes, which were retrospectively sampled from 399 patients who underwent breast cancer surgery at two hospitals in the Netherlands. All metastases in the slides were annotated under the supervision of multiple expert pathologists.

**CheXpert** is a large public dataset for chest radiograph interpretation, consisting of 224,316 X-ray scans from 65,240 patients (Irvin et al., 2019). The authors retrospectively collected the chest radiographic examinations from Stanford Hospital, performed between October 2002 and July 2017 in both inpatient and outpatient centers, along with their associated radiology reports. Labels were extracted from the free-text radiology reports with an automated rule-based system. The dataset further contains radiologist-labeled reference evaluation sets.

**CelebFaces Attributes Dataset (CelebA)** is a large-scale face attributes dataset with more than 200,000 celebrity images, each with 40 attribute annotations (Liu et al., 2015). The images in this dataset cover large pose variations and mixed backgrounds. With 10,177 identities and 202,599 face images, CelebA includes many diverse samples and features rich annotations.

**Food-101N** is an image dataset that contains 310,009 images of food recipes divided into 101 classes (Lee et al., 2018). Both Food-101N and the Food-101 (Bossard et al., 2014) dataset share the same 101 classes. However, Food-101N has a significantly larger number of images and contains more noise. The pictures were scraped from Google, Bing, Yelp, and TripAdvisor, and 60,000 of them were manually verified for evaluation. The dataset includes information for each sample on whether or not it features a label problem.

**ImageNet-1k (IN)** is a well-known benchmark image database with 1,000 classes (Deng et al., 2009). Images were scraped by querying words from WordNet's "synonym set" (synsets) on several image search engines. The images were labeled by Amazon Mechanical Turk workers, who were asked whether each image contained objects of a given synset.

**STL-10** is a benchmark collection consisting of 10 classes, each with 500 training images, 800 test images, and an additional 100,000 unlabeled images for unsupervised learning (Coates et al., 2011). It focuses on higher resolution images (96x96 pixels) compared to other similar collections like CIFAR-10. The images in STL-10 are sourced from labeled examples in ImageNet and are chosen to represent a broad range of object categories and real-world scenarios.

# E    COMPETING APPROACHES

We selected different competitive and popular approaches to detect each of the three data quality issue categories, i.e. irrelevant samples, near duplicates, and label errors. Some of these methods require to encode images in a low-dimensional latent space. For this projection we used a ViT-tiny, the same architecture used for the proposed methodology, pre-trained on ImageNet or DINO, referred to as "IN + BASE" or "DINO + BASE" respectively, where BASE is the corresponding detection approach. In this section, we briefly summarize each competing approach used in this work.

## E.1    APPROACHES FOR IRRELEVANT SAMPLES

**Isolation Forest (IForest)** isolate observations by randomly selecting a feature and split value between the minimum and maximum of the selected feature, where the number of splits required to isolate a sample corresponds to the path length from the root node to the leaf node in a tree (Liu et al., 2008). This path length averaged over a forest of random trees is a measure of normality, where noticeably shorter paths are produced for anomalies.

**Histogram-based outlier detection (HBOS)** is an efficient unsupervised method that creates a histogram of the feature vector for each dimension and then calculates a score based on how likely a particular data point is to fall within the histogram bins for each dimension (Goldstein & Dengel, 2012). The higher the score, the more likely the data point is an outlier, i.e. a feature vector coming from an anomaly will occupy unlikely bins in one or several of its dimensions and thus produce a higher anomaly score.

**Deep One-Class Classifier with Auto Encoder (DeepSVDD)** is a type of neural network for learning useful data representations in an unsupervised way (Ruff et al., 2018). DeepSVDD trains a neural network while minimizing the volume of a hypersphere that encloses the network representations of the data, forcing the network to extract the common factors of variation. DeepSVDD then detects outliers in the data by calculating the distance from the center.

**Empirical Cumulative Distribution Functions (ECOD)** is a parameter-free, highly-interpretable unsupervised outlier detection algorithm (Li et al., 2022). It estimates an empirical cumulative distribution function (ECDF) for each variable in the data separately. To generate an outlier score for an observation, it computes the tail probability for each variable using the univariate ECDFs and multiplies them together. This calculation is done in log space, accounting for each dimension's left and right tails.

## E.2    APPROACHES FOR NEAR DUPLICATES

**Perceptual Hash (pHashing)** is a type of locality-sensitive hash, which is similar if features of the sample are similar (Marr et al., 1997). It relies on the discrete cosine transform (DCT) for dimensionality reduction and produces hash bits depending on whether each DCT value is above or below the average value. In this paper, we use pHash with a hash size of 8.

**Structural Similarity Index Measure (SSIM)** is a type of similarity measure to compare two images with each other based on three features, namely luminance, contrast, and structure (Wang et al., 2004). Instead of applying SSIM globally, i.e. all over the image at once, one usually applies the metrics regionally, i.e. in small sections of the image, and takes the mean overall. This variant of SSIM is often called Mean Structural Similarity Index. In this paper, we in fact apply SSIM locally to 8x8 windows but still refer to the method as SSIM for simplicity.

### E.3 APPROACHES FOR LABEL ERRORS

**Confident Learning (CL)** is a data-centric approach that focuses on label quality by characterizing and identifying label errors in datasets based on the principles of pruning noisy data, counting with probabilistic thresholds to estimate noise, and ranking examples to train with confidence (Northcutt et al., 2022). It builds upon the assumption of a class-conditional noise process to directly estimate the joint distribution between noisy (given) and uncorrupted (unknown) labels, resulting in a generalized CL that is provably consistent and experimentally performant. In this study, we use AdaBoost (Freund & Schapire, 1999) as a classifier to estimate the probabilities. We did not observe any significant performance difference when using different classifiers similarly to Northcutt et al. (2022).

**NoiseRank (Noise)** is a method for unsupervised label noise detection using Markov Random Fields (Sharma et al., 2020). It constructs a dependence model to estimate the posterior probability of an instance being incorrectly labeled, given the dataset, and then ranks instances based on this probability.

### E.4 APPROACHES FOR MULTIPLE NOISE TYPES

**FastDup** is a powerful free and open-source tool designed to rapidly extract valuable insights from image and video datasets, aiming to increase the dataset quality and reduce data operations costs at an unparalleled scale (Alush et al., 2022). It can detect and eliminate anomalies and outliers, identify duplicate and near-duplicate images and videos, and find wrongly-labeled samples.

## F SELF-SUPERVISED ENCODER

This section presents further investigation of the encoder pre-training and its influence on downstream tasks.

### F.1 PERFORMANCE OF PRE-TRAINED ENCODER

Since SELFCLEAN relies on dataset-specific SSL to produce meaningful latent spaces, the learned features can further be used for transfer learning on the same dataset. In table 3, we consider DINO pre-training, as performed for SELFCLEAN, on different medical and vision benchmarks, i.e. DDI, HAM10000, Fitzpatrick17k, Food-101N, and ImageNet-1k. The features obtained are compared to a random and supervised ImageNet initialized ViT tiny in terms of linear performance on the respective classification task of the datasets, i.e. binary classification (benign/malignant) for DDI, multi-class classification for HAM10000, Fitzpatrick17k, Food-101N, and ImageNet-1k. The results show that ImageNet features perform better for the DDI and ImageNet-1k datasets. This is not unexpected since DDI is relatively small and the representation was obtained specifically for supervised ImageNet classification. In contrast, for HAM10000, Fitzpatrick17k, and Food-101N, the features produced by SELFCLEAN are lead to improved results. This indicates a further benefit of using SELFCLEAN, since the features used for cleaning can be recycled for the considered downstream task where they potentially improve performance.

Table 3: Macro-averaged test F1 scores for multiple vision benchmarks, obtained with linear classifiers on top of different pre-trained feature sets. Representations are extracted from a ViT tiny using random initialization weights as a baseline, transfer learning from supervised ImageNet, and DINO SSL pre-training as performed by SELFCLEAN. Results are obtained from runs with three different random seeds, of which we report the average performance and standard error.

| Method | DDI (%) | HAM10000 (%) | Fitzpatrick17k (%) | Food-101N (%) | ImageNet-1k (%) |
|---|---|---|---|---|---|
| Random | $52.5 \pm 1.0$ | $21.9 \pm 0.5$ | $28.9 \pm 0.2$ | $5.1 \pm 0.1$ | $0.8 \pm 0.1$ |
| IN | $\mathbf{63.6} \pm \mathbf{1.1}$ | $57.5 \pm 0.4$ | $52.3 \pm 0.1$ | $39.1 \pm 0.1$ | $\mathbf{49.1} \pm \mathbf{0.1}$ |
| SELFCLEAN | $59.2 \pm 1.9$ | $\mathbf{65.2} \pm \mathbf{0.4}$ | $\mathbf{53.8} \pm \mathbf{0.4}$ | $\mathbf{53.9} \pm \mathbf{0.2}$ | $34.9 \pm 0.1$ |

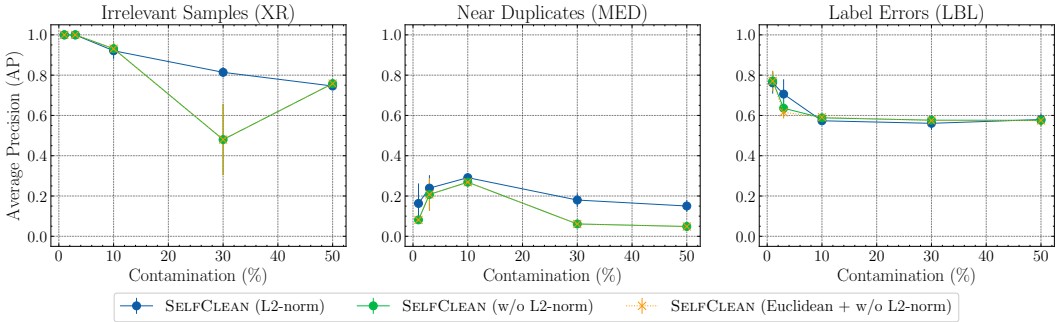

Figure 3: Performance of SELFCLEAN when changing the distance function used for comparison and removing the $L_2$-normalization. The performance is measured in terms of average precision (AP) for a "mixed-contamination strategy" when varying the contamination rate. The artificial dataset is created from DDI by adding X-ray images (XR), then injecting augmented duplicates (MED), and finally changing labels at random (LBL).

## F.2  INFLUENCE OF $L_2$-NORMALIZATION AND DISTANCE FUNCTIONS

To compare DINO with SimCLR, we included explicit $L_2$ normalization in the latent space during both training and inference for DINO. A similar explicit $L_2$ normalization for representation layers is also used in theoretical works on SSL (Dubois et al., 2022), where it was inherited from the neural collapse literature (E & Wojtowytsch, 2022). We investigate the influence of this $L_2$ normalization on the detection performance for the different dataset quality issues. Figure 3 shows the performance of SELFCLEAN with and without normalization for the mixed-contamination dataset constructed in the paragraph "influence of contamination rate" of section 5.1. The results show that $L_2$ normalization increases robustness of irrelevant samples detection, improves performance in finding near duplicates, and it has little effect on label error detection. One possible explanation for the improved performance is that limiting the latent space to the unit hypersphere enforces a more direct relation between the training objective and the relative distances of encoded samples.

In a second experiment, we examined the influence of the choice of the distance function between cosine and Euclidean distance. Since the Euclidean and cosine distance on a $L_2$ normalized space are equivalent and produce the same ranking, we only show the results of different distance functions for the non-normalized latent space. Figure 3 shows that performance is almost unaffected by the choice of distance function, and suggests that normalization does not help by enforcing a specific distance, but rather by constraining the latent space.

## F.3  COMPARISON OF SELF-ATTENTION

In this experiment, we qualitatively compare on selected target datasets the last self-attention block of the encoders trained with DINO and supervised ImageNet classification. This experiment illustrates the difference between self-supervised training on the target domain and domain-agnostic pre-training. Figure 4 shows the self-attention masks for a random sample from ImageNet, CheXpert, and Fitzpatrick17k. A clear difference is visible for the self-attention masks obtained using the two pre-training strategies. For example, the masks show that encoders pre-trained with self-supervision attend to many more visual features than their supervised pre-trained counterparts, indicating that self-supervised representations contain more holistic information.

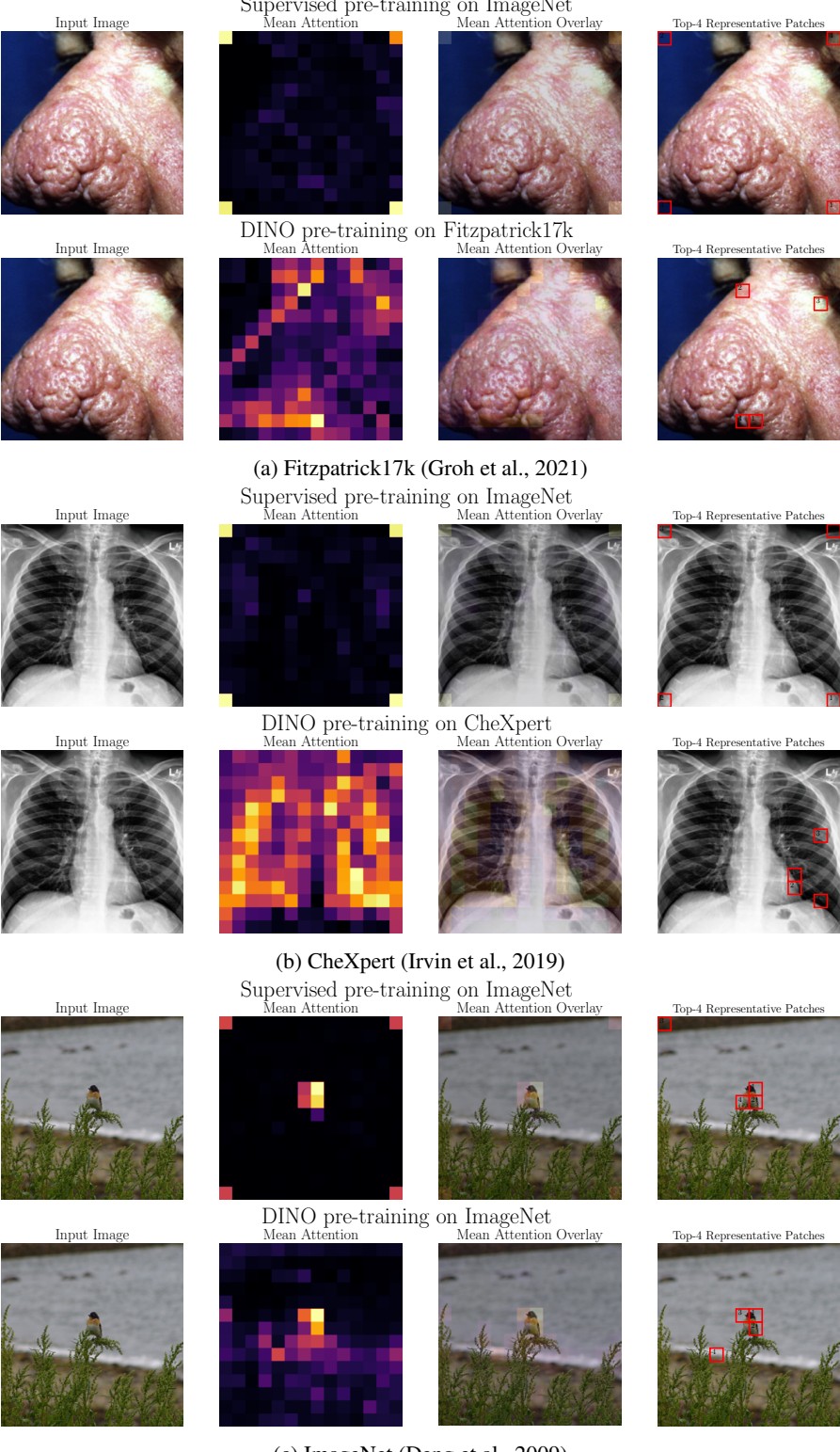

Figure 4: Visualization of the last self-attention block of the encoder pre-trained with supervision on ImageNet and with DINO self-supervision on domain-specific datasets, for representative images of different datasets. The first column shows the input image, the second the focus of the model computed as the mean self-attention of the last block, the third is an overlay of the first two, and the fourth shows the four most representative patches.

# G   DETAILED DATASET CLEANING RESULTS

This section provides extended tables with performance results related to dataset cleaning. More precisely, section G.1 investigates synthetic contamination detection with different methods, metrics, and noise levels, expanding on section 5.1. Section G.2 presents in tabular form the comparison of SELFCLEAN with available metadata as discussed in section 5.2. Section G.4 extends table 2 in section 6 by including information on the performances used to compute paired differences. Finally, G.5 summarizes the number of data quality issues found in the considered benchmark datasets.

## G.1   DETAILED COMPARISON ON SYNTHETIC DATA QUALITY ISSUES

Table 4: Performance of various models on the detection of synthetic data quality issues. Evaluation is performed for each of the three considered issue types across two clean benchmark datasets, DDI and HQ-FST, augmented with two strategies for synthetic contamination each (XR, CMED, MED, ARTE, LBL, and LBLC). All scores are reported in percentages (%), and P@$k$ and R@$k$ indicate precision and recall at $k$ respectively. Consult section 4 for more details.

### Contamination 5%

**Irrelevant Samples**

| Method | DDI + XR | | | | DDI + CMED | | | | HQ-FST + XR | | | | HQ-FST + CMED | | | |
|---|---|---|---|---|---|---|---|---|---|---|---|---|---|---|---|---|
| | P@20 | R@20 | AUROC | AP | P@20 | R@20 | AUROC | AP | P@20 | R@20 | AUROC | AP | P@20 | R@20 | AUROC | AP |
| IN + IForest | 30.0 | 18.2 | 92.3 | 26.9 | 55.0 | 36.7 | 91.8 | 53.1 | 15.0 | 11.5 | 81.5 | 12.8 | 70.0 | 53.8 | 96.3 | 68.4 |
| IN + HBOS | 50.0 | 30.3 | 95.4 | 47.4 | 50.0 | 33.3 | 78.9 | 35.3 | 20.0 | 15.4 | 86.5 | 16.9 | 55.0 | 42.3 | 94.3 | 60.1 |
| IN + DeepSVDD | 0.0 | 0.0 | 71.3 | 10.5 | 10.0 | 6.7 | 69.4 | 11.9 | 5.0 | 3.8 | 85.8 | 19.2 | 10.0 | 7.7 | 60.6 | 11.4 |
| IN + ECOD | 40.0 | 24.2 | 91.5 | 28.4 | 70.0 | 46.7 | 97.9 | 68.2 | 10.0 | 7.7 | 79.7 | 12.1 | 45.0 | 34.6 | 94.3 | 42.9 |
| FastDup | 0.0 | 0.0 | 9.7 | 2.7 | 60.0 | 40.0 | 74.6 | 45.2 | 0.0 | 0.0 | 7.5 | 2.7 | 85.0 | 65.4 | 99.2 | 86.8 |
| SELFCLEAN (IN) | 100.0 | 60.6 | 100.0 | 100.0 | 100.0 | 66.7 | 100.0 | 100.0 | 80.0 | 61.5 | 99.2 | 70.6 | 100.0 | 76.9 | 99.9 | 98.8 |
| SELFCLEAN (SimCLR) | 65.0 | 39.4 | 91.8 | 49.0 | 45.0 | 30.0 | 73.3 | 36.8 | 90.0 | 69.2 | 96.0 | 81.3 | 40.0 | 33.3 | 93.8 | 42.4 |
| SELFCLEAN (DINO) | 100.0 | 60.6 | 100.0 | 100.0 | 100.0 | 66.6 | 98.9 | 92.8 | 100.0 | 76.9 | 100.0 | 100.0 | 85.0 | 70.8 | 99.1 | 83.2 |

**Near Duplicates**

| Method | DDI + MED | | | | DDI + ARTE | | | | HQ-FST + MED | | | | HQ-FST + ARTE | | | |
|---|---|---|---|---|---|---|---|---|---|---|---|---|---|---|---|---|
| | P@20 | R@20 | AUROC | AP | P@20 | R@20 | AUROC | AP | P@20 | R@20 | AUROC | AP | P@20 | R@20 | AUROC | AP |
| pHashing | 20.0 | 12.1 | 65.1 | 9.3 | 40.0 | 24.2 | 62.7 | 24.3 | 5.0 | 3.8 | 71.0 | 1.1 | 40.0 | 30.8 | 80.9 | 30.5 |
| SSIM | 45.0 | 27.3 | 85.9 | 22.1 | 40.0 | 24.2 | 74.6 | 25.4 | 35.0 | 26.9 | 87.0 | 24.7 | 40.0 | 30.8 | 81.2 | 32.9 |
| FastDup | 45.0 | 27.3 | 70.8 | 16.9 | 15.0 | 9.1 | 53.7 | 7.6 | 15.0 | 11.5 | 56.1 | 3.8 | 10.0 | 7.7 | 44.6 | 3.4 |
| SELFCLEAN (IN) | 20.0 | 12.1 | 94.8 | 7.0 | 35.0 | 21.2 | 91.9 | 21.7 | 30.0 | 23.1 | 99.1 | 18.2 | 40.0 | 30.8 | 96.1 | 29.1 |
| SELFCLEAN (SimCLR) | 35.0 | 21.2 | 98.5 | 21.6 | 30.0 | 18.2 | 95.5 | 18.1 | 35.0 | 26.9 | 95.7 | 18.5 | 15.0 | 11.5 | 93.6 | 9.1 |
| SELFCLEAN (DINO) | 80.0 | 48.5 | 99.8 | 55.9 | 95.0 | 57.6 | 99.5 | 71.0 | 50.0 | 38.5 | 99.3 | 38.2 | 50.0 | 38.5 | 96.7 | 44.1 |

**Label Errors**

| Method | DDI + LBL | | | | DDI + LBLC | | | | HQ-FST + LBL | | | | HQ-FST + LBLC | | | |
|---|---|---|---|---|---|---|---|---|---|---|---|---|---|---|---|---|
| | P@20 | R@20 | AUROC | AP | P@20 | R@20 | AUROC | AP | P@20 | R@20 | AUROC | AP | P@20 | R@20 | AUROC | AP |
| IN + Confident Learning | 15.0 | 9.1 | 74.6 | 16.9 | 10.0 | 6.1 | 75.2 | 12.5 | 30.0 | 24.0 | 76.5 | 18.9 | 20.0 | 12.9 | 69.8 | 16.3 |
| IN + NoiseRank | 5.0 | 3.1 | 50.3 | 5.4 | 10.0 | 6.2 | 56.2 | 7.1 | 15.0 | 11.5 | 60.9 | 11.2 | 25.0 | 16.7 | 62.0 | 18.0 |
| FastDup | 15.0 | 9.1 | 77.9 | 12.9 | 35.0 | 7.5 | 73.8 | 26.3 | 15.0 | 11.5 | 82.8 | 17.3 | 25.0 | 16.7 | 86.2 | 22.6 |
| SELFCLEAN (IN) | 10.0 | 6.1 | 60.4 | 13.3 | 20.0 | 4.3 | 44.8 | 13.9 | 20.0 | 15.4 | 53.8 | 13.5 | 15.0 | 9.7 | 56.6 | 8.9 |
| SELFCLEAN (SimCLR) | 10.0 | 6.1 | 50.9 | 6.9 | 20.0 | 4.3 | 42.8 | 14.0 | 25.0 | 19.2 | 48.4 | 13.2 | 15.0 | 10.0 | 46.7 | 10.5 |
| SELFCLEAN (DINO) | 15.0 | 9.1 | 72.2 | 13.6 | 75.0 | 16.3 | 75.4 | 41.0 | 30.0 | 23.1 | 80.8 | 20.4 | 40.0 | 26.7 | 86.5 | 33.4 |

### Contamination 10%

**Irrelevant Samples**

| Method | DDI + XR | | | | DDI + CMED | | | | HQ-FST + XR | | | | HQ-FST + CMED | | | |
|---|---|---|---|---|---|---|---|---|---|---|---|---|---|---|---|---|
| | P@20 | R@20 | AUROC | AP | P@20 | R@20 | AUROC | AP | P@20 | R@20 | AUROC | AP | P@20 | R@20 | AUROC | AP |
| IN + IForest | 50.0 | 15.4 | 89.5 | 37.0 | 65.0 | 21.7 | 89.6 | 52.3 | 55.0 | 21.6 | 88.1 | 37.2 | 85.0 | 33.3 | 96.3 | 74.8 |
| IN + HBOS | 55.0 | 16.9 | 92.5 | 46.7 | 65.0 | 21.7 | 80.7 | 42.9 | 25.0 | 9.8 | 84.3 | 27.3 | 90.0 | 35.3 | 94.0 | 69.7 |
| IN + DeepSVDD | 10.0 | 3.1 | 52.4 | 9.8 | 20.0 | 6.7 | 75.9 | 21.0 | 5.0 | 2.0 | 51.3 | 9.0 | 20.0 | 7.8 | 66.5 | 23.1 |
| IN + ECOD | 25.0 | 7.7 | 85.1 | 27.1 | 75.0 | 25.0 | 96.9 | 68.5 | 0.0 | 0.0 | 71.9 | 15.6 | 55.0 | 21.6 | 93.8 | 53.1 |
| FastDup | 0.0 | 0.0 | 7.3 | 4.9 | 60.0 | 20.0 | 43.8 | 26.6 | 0.0 | 0.0 | 4.5 | 4.9 | 90.0 | 35.3 | 95.3 | 80.7 |
| SELFCLEAN (IN) | 100.0 | 30.8 | 100.0 | 100.0 | 100.0 | 33.3 | 100.0 | 99.8 | 80.0 | 31.4 | 99.2 | 80.3 | 100.0 | 39.2 | 99.7 | 96.7 |
| SELFCLEAN (SimCLR) | 5.0 | 1.5 | 90.8 | 32.8 | 45.0 | 15.0 | 77.8 | 31.6 | 0.0 | 0.0 | 95.5 | 48.1 | 30.0 | 12.5 | 81.4 | 26.0 |
| SELFCLEAN (DINO) | 100.0 | 30.8 | 100.0 | 100.0 | 100.0 | 33.3 | 97.4 | 89.5 | 100.0 | 39.2 | 100.0 | 100.0 | 95.0 | 39.6 | 98.5 | 84.1 |

**Near Duplicates**

| Method | DDI + MED | | | | DDI + ARTE | | | | HQ-FST + MED | | | | HQ-FST + ARTE | | | |
|---|---|---|---|---|---|---|---|---|---|---|---|---|---|---|---|---|
| | P@20 | R@20 | AUROC | AP | P@20 | R@20 | AUROC | AP | P@20 | R@20 | AUROC | AP | P@20 | R@20 | AUROC | AP |
| pHashing | 30.0 | 9.2 | 69.2 | 9.8 | 95.0 | 29.2 | 68.9 | 33.4 | 35.0 | 13.7 | 71.6 | 12.5 | 75.0 | 29.4 | 80.5 | 30.6 |
| SSIM | 70.0 | 21.5 | 85.7 | 25.7 | 100.0 | 30.8 | 84.2 | 35.3 | 65.0 | 25.5 | 92.6 | 36.1 | 80.0 | 31.4 | 88.5 | 32.9 |
| FastDup | 40.0 | 12.3 | 58.6 | 10.0 | 40.0 | 12.3 | 59.6 | 5.6 | 25.0 | 9.8 | 56.1 | 7.2 | 35.0 | 13.7 | 52.9 | 8.9 |
| SELFCLEAN (IN) | 15.0 | 4.6 | 96.7 | 6.0 | 75.0 | 23.1 | 94.9 | 29.2 | 60.0 | 23.5 | 97.8 | 27.9 | 60.0 | 23.5 | 93.3 | 27.2 |
| SELFCLEAN (SimCLR) | 40.0 | 12.3 | 96.1 | 13.7 | 95.0 | 29.2 | 88.6 | 37.5 | 55.0 | 21.6 | 97.4 | 22.4 | 75.0 | 18.2 | 87.2 | 17.3 |
| SELFCLEAN (DINO) | 95.0 | 29.3 | 99.6 | 63.0 | 100.0 | 31.0 | 99.6 | 65.4 | 95.0 | 37.3 | 99.3 | 61.2 | 80.0 | 31.4 | 95.3 | 38.9 |

**Label Errors**

| Method | DDI + LBL | | | | DDI + LBLC | | | | HQ-FST + LBL | | | | HQ-FST + LBLC | | | |
|---|---|---|---|---|---|---|---|---|---|---|---|---|---|---|---|---|
| | P@20 | R@20 | AUROC | AP | P@20 | R@20 | AUROC | AP | P@20 | R@20 | AUROC | AP | P@20 | R@20 | AUROC | AP |
| IN + Confident Learning | 25.0 | 7.9 | 71.4 | 17.8 | 30.0 | 9.7 | 72.0 | 23.6 | 35.0 | 15.6 | 77.0 | 31.7 | 40.0 | 15.1 | 83.8 | 38.6 |
| IN + NoiseRank | 10.0 | 3.3 | 48.8 | 9.9 | 15.0 | 4.7 | 47.5 | 10.9 | 15.0 | 6.2 | 60.0 | 16.1 | 20.0 | 7.7 | 59.3 | 17.7 |
| FastDup | 15.0 | 4.7 | 68.6 | 14.6 | 25.0 | 4.4 | 71.4 | 28.8 | 40.0 | 16.0 | 85.2 | 32.3 | 30.0 | 11.8 | 84.7 | 28.9 |
| SELFCLEAN (IN) | 40.0 | 12.9 | 60.3 | 22.9 | 15.0 | 2.5 | 43.8 | 16.7 | 15.0 | 6.1 | 48.5 | 10.7 | 20.0 | 7.8 | 49.1 | 12.8 |
| SELFCLEAN (SimCLR) | 10.0 | 3.3 | 39.6 | 8.0 | 15.0 | 2.6 | 42.8 | 16.0 | 10.0 | 4.1 | 56.3 | 17.7 | 15.0 | 6.0 | 47.2 | 13.2 |
| SELFCLEAN (DINO) | 30.0 | 9.8 | 77.2 | 26.3 | 85.0 | 14.8 | 78.7 | 50.0 | 50.0 | 20.4 | 85.1 | 39.7 | 40.0 | 16.0 | 79.9 | 35.2 |

### G.2 DETAILED COMPARISON WITH METADATA

Table 5 details the comparison of the SELFCLEAN ranking with metadata from multiple benchmark datasets as discussed in section 5.2. For PAD-UFES-20, we conducted additional investigations on SELFCLEAN's lower performance, as discussed in "Comparision with metadata" of section 5.2. We provided the near-duplicate rankings of PAD-UFES-20 to three practicing dermatologists, asking them to verify if given samples are near duplicates. The experts reached an inter-annotator agreement as Krippendorff's alpha of $> 0.6$, which can be considered good. Of the samples they unanimously agreed on were near duplicates (56 samples), 32% had faulty metadata where the lesion ID was not correctly maintained. Thus, we find evidence that the poor alignment of SELFCLEAN and the metadata of PAD-UFES-20 is likely caused by imperfect metadata.

Table 5: Comparison of the SELFCLEAN ranking with metadata from multiple benchmark datasets. For reference, we include the proportion of positive samples, also corresponding to the not-informed baseline performing best in terms of AP. Please consult section 5.2 for guidance on interpretation.

| Dataset | Metadata | Positive Samples (%) | AUROC (%) | AP (%) |
|---|---|---|---|---|
| PAD-UFES-20 | Same Lesion | 0.06 | 71.0 | 10.0 |
| PAD-UFES-20 | Not Biopsied | 41.60 | 57.0 | 61.3 |
| HAM10000 | Same Lesion | 0.01 | 98.8 | 28.4 |
| HAM10000 | Not Follow-Up | 63.89 | 90.8 | 86.3 |
| ISIC-2019 | Same Lesion | 0.01 | 98.2 | 26.6 |
| CheXpert | Same Patient | 0.01 | 70.5 | 7.5 |
| ImageNet-1k[2] | Verified Label Errors | 88.73 | 67.3 | 8.4 |
| Food-101N[3] | Verified Label Errors | 18.51 | 79.8 | 47.8 |
| CelebA | Same Person | 0.02 | 78.8 | 30.9 |

---

[2]Refers to the subset of ImageNet-1k validation set which was verified by Northcutt et al. (2021).
[3]Refers to the subset of Food-101N set which was verified by Lee et al. (2018).

Table 6: Comparison of SELFCLEAN and competitor rankings with metadata from multiple benchmark datasets. For reference, we include the proportion of positive samples, also corresponding to the not-informed baseline performing best in terms of AP. Please consult section 5.2 for guidance on interpretation.

| Dataset | Metadata | Positive Samples (%) | Method | AUROC (%) | AP (%) |
|---|---|---|---|---|---|
| PAD-UFES-20 | Same Lesion | 0.06 | SELFCLEAN | **71.0** | **10.0** |
| | | | SSIM | 63.7 | 0.3 |
| | | | pHASH | 56.6 | 0.2 |
| HAM10000 | Same Lesion[4] | 0.01 | SELFCLEAN | **98.7 [97.9, 99.0]** | **30.0 [23.9, 34.3]** |
| | | | SSIM | 67.3 [66.3, 72.4] | 7.7 [4.4, 7.8] |
| | | | pHASH | 71.3 [69.9, 74.2] | 2.7 [2.6, 7.6] |
| ISIC-2019 | Same Lesion[4] | 0.01 | SELFCLEAN | **98.9 [97.4, 98.9]** | **28.6 [26.6, 29.2]** |
| | | | SSIM | 69.1 [66.3, 70.0] | 1.3 [0.3, 1.6] |
| | | | pHASH | 62.3 [58.5, 63.4] | 0.1 [0.0, 2.1] |
| CheXpert | Same Patient[4] | 0.01 | SELFCLEAN | **86.5 [85.5, 88.1]** | **1.9 [0.3, 2.3]** |
| | | | SSIM | 65.7 [64.7, 66.1] | 0.2 [0.2, 0.3] |
| | | | pHASH | 54.7 [53.8, 57.0] | 0.2 [0.1, 0.4] |
| CelebA | Same Person[4] | 0.02 | SELFCLEAN | **81.0 [80.6, 81.2]** | **0.6 [0.6, 0.6]** |
| | | | SSIM | 56.3 [55.3, 58.3] | 0.0 [0.0, 0.0] |
| | | | pHASH | 53.3 [52.8, 54.7] | 0.0 [0.0, 0.0] |
| ImageNet-1k[2] | Verified Label Errors | 88.73 | SELFCLEAN | **67.3** | **8.4** |
| | | | IN+CL | 49.8 | 4.4 |
| | | | IN+NOISE | 46.4 | 3.7 |
| | | | FastDup | 42.6 | 3.6 |
| Food-101N[3] | Verified Label Errors | 18.51 | SELFCLEAN | **79.8** | **47.8** |
| | | | IN+CL | 61.0 | 25.2 |
| | | | IN+NOISE | 48.8 | 18.9 |
| | | | FastDup | 72.1 | 30.7 |

---

[4]As the number of near duplicates for comparison exceeds memory limitations for the baseline methods, they were subsampled three times with the same percentage of positive samples to 2,000 samples (i.e. 1,999,000 comparisons) and report the median and the min-max variation in brackets.

## G.3   POTENTIAL BIAS OF IRRELEVANT RANKING

There is a chance that irrelevant sample detection may exacerbate data distribution biases, because underrepresented samples are more likely to be proposed as candidate issues. For this reason, we investigate if some specific attributes of the dataset correlate with the irrelevant sample ranking, assessing for example if the prevalence of pigment-rich skin is higher in the earlier rankings than in the latter. We focus on the demographics of CheXpert and skin types in DDI and Fitzpatrick17k. We compare the ranking of the feature attribute using AP and AUROC, similar to the comparison with metadata in Appendix G.2. The results show no evidence for an increased likelihood of underrepresented groups to appear earlier in the ranking, as the average precision is overall very similar to the non-informed baseline i.e. the percentage of samples belonging to the group.

Table 7: Comparison of the SELFCLEAN ranking with various demographic attributes. For reference, we include the prevalence of each group, also corresponding to the not-informed baseline performing best in terms of AP.

| Dataset | Attribute | Value | Prevalence (%) | AUROC (%) | AP (%) |
|---|---|---|---|---|---|
| DDI | Skin Tone | Fitzpatrick Type 3&4 | 36.7 | 46.8 | 35.4 |
| | | Fitzpatrick Type 1&2 | 31.7 | 52.5 | 31.2 |
| | | Fitzpatrick Type 5&6 | 31.6 | 50.9 | 35.9 |
| Fitzpatrick17k | Skin Tone | Fitzpatrick Type 2 | 29.0 | 53.2 | 31.1 |
| | | Fitzpatrick Type 3 | 20.0 | 47.5 | 19.1 |
| | | Fitzpatrick Type 1 | 17.8 | 52.8 | 18.9 |
| | | Fitzpatrick Type 4 | 16.8 | 45.4 | 15.2 |
| | | Fitzpatrick Type 5 | 9.2 | 46.3 | 8.5 |
| | | Fitzpatrick Type 6 | 3.8 | 50.8 | 3.8 |
| | | Fitzpatrick Type Unknown | 3.4 | 57.5 | 4.3 |
| CheXpert | Ethnicity | Non-Hispanic/Non-Latino | 72.9 | 50.0 | 72.8 |
| | | Unknown | 14.2 | 53.3 | 15.4 |
| | | Hispanic/Latino | 12.1 | 46.5 | 11.1 |
| | | Patient Refused | 0.3 | 43.5 | 0.2 |
| | | Not Hispanic | 0.0 | 35.1 | 0.0 |
| | | Hispanic | 0.0 | 9.1 | 0.0 |
| CheXpert | Gender | Male | 55.2 | 43.5 | 51.4 |
| | | Female | 44.3 | 56.4 | 48.5 |
| | | Unknown | 0.0 | 17.7 | 0.0 |
| CheXpert | Primary Race | White | 45.5 | 47.7 | 44.0 |
| | | Other | 12.9 | 46.4 | 11.9 |
| | | White, non-Hispanic | 10.0 | 55.7 | 11.5 |
| | | Asian | 9.5 | 51.7 | 9.7 |
| | | Unknown | 6.6 | 52.5 | 7.1 |
| | | Black or African American | 4.0 | 47.4 | 3.8 |
| | | Race and Ethnicity Unknown | 3.9 | 53.8 | 4.3 |
| | | Other, Hispanic | 1.7 | 49.6 | 1.6 |
| | | Asian, non-Hispanic | 1.2 | 56.0 | 1.4 |
| | | Native Hawaiian or Other Pacific Islander | 1.2 | 44.1 | 1.0 |
| | | Black, non-Hispanic | 0.8 | 55.5 | 1.0 |
| | | White, Hispanic | 0.5 | 53.7 | 0.6 |
| | | Other, non-Hispanic | 0.3 | 56.4 | 0.4 |
| | | Patient Refused | 0.2 | 44.4 | 0.2 |
| | | American Indian or Alaska Native | 0.2 | 46.5 | 0.2 |
| | | Pacific Islander, non-Hispanic | 0.1 | 51.4 | 0.2 |
| | | Native American, non-Hispanic | 0.0 | 60.7 | 0.0 |
| | | Black, Hispanic | 0.0 | 60.7 | 0.0 |
| | | Native American, Hispanic | 0.0 | 60.2 | 0.0 |
| | | Asian, Hispanic | 0.0 | 63.4 | 0.0 |
| | | White or Caucasian | 0.0 | 18.9 | 0.0 |
| | | Pacific Islander, Hispanic | 0.0 | 65.1 | 0.2 |
| | | Asian - Historical Conv | 0.0 | 58.7 | 0.0 |

## G.4 DETAILED INFLUENCE OF DATASET CLEANING

Table 8: Influence of removing samples detected in the automatic cleaning mode with $\alpha = 0.10$ and $q = 0.05$ on downstream tasks. We report macro-averaged F1 scores for linear and $k$NN classifiers on DINO features over 100 random training/evaluation splits with 80% and 20% fractions respectively. We compute paired performance differences before and after cleaning the evaluation set, and before and after cleaning also the training set. We report the median and the intervals to the 5% (subscript) and 95% (superscript) percentiles. Additionally, we indicate significance of a paired permutation test on the difference sign with $^*p < 0.05$, $^{**}p < 0.01$, and $^{***}p < 0.001$.

| | $k$**NN Classifier** | | | | |
| | **Scores (%)** | | | **Differences (%)** | |
| | Cont + Cont | Cont + Clean | Clean + Clean | Clean Eval | Clean Train |
|---|---|---|---|---|---|
| DDI | $58.2^{+7.7}_{-8.3}$ | $59.2^{+7.5}_{-8.3}$ | $59.7^{+7.3}_{-8.8}$ | $+1.2^{+1.9}_{-1.2}$ *** | $+0.0^{+1.7}_{-1.4}$ *** |
| HAM10000 | $58.3^{+3.4}_{-4.9}$ | $58.3^{+3.7}_{-4.7}$ | $58.7^{+3.1}_{-4.6}$ | $+0.2^{+0.5}_{-0.4}$ *** | $+0.2^{+1.3}_{-0.8}$ ** |
| Fitzpatrick17k | $60.2^{+1.8}_{-1.9}$ | $56.1^{+1.9}_{-2.2}$ | $56.1^{+2.0}_{-2.3}$ | $-4.1^{+1.2}_{-1.3}$ *** | $+0.1^{+2.0}_{-1.7}$ |
| Food-101N | $40.3^{+0.8}_{-0.9}$ | $40.4^{+0.7}_{-1.1}$ | $40.5^{+0.7}_{-1.1}$ | $+0.1^{+0.1}_{-0.1}$ *** | $+0.1^{+0.2}_{-0.2}$ *** |
| ImageNet-1k | $31.2^{+0.8}_{-0.9}$ | $30.8^{+0.9}_{-0.9}$ | $31.1^{+0.8}_{-0.9}$ | $-0.4^{+0.1}_{-0.2}$ *** | $+0.4^{+0.3}_{-0.4}$ *** |

| | **Linear Classifier** | | | | |
| | **Scores (%)** | | | **Differences (%)** | |
| **Dataset** | Cont + Cont | Cont + Clean | Clean + Clean | Clean Eval | Clean Train |
|---|---|---|---|---|---|
| DDI | $59.2^{+9.6}_{-10.2}$ | $59.6^{+12.0}_{-11.2}$ | $58.9^{+9.0}_{-9.7}$ | $+1.0^{+11.1}_{-11.2}$ | $-0.7^{+7.7}_{-10.8}$ |
| HAM10000 | $62.6^{+4.2}_{-4.2}$ | $63.0^{+3.3}_{-4.0}$ | $62.8^{+3.2}_{-3.8}$ | $+0.1^{+3.2}_{-3.5}$ | $-0.1^{+3.9}_{-3.6}$ |
| Fitzpatrick17k | $52.8^{+2.6}_{-3.1}$ | $52.5^{+2.5}_{-4.1}$ | $52.6^{+2.9}_{-2.8}$ | $-0.6^{+2.9}_{-3.6}$ ** | $+0.2^{+3.3}_{-3.9}$ * |
| Food-101N | $50.0^{+0.9}_{-1.2}$ | $50.1^{+1.1}_{-1.0}$ | $50.4^{+0.8}_{-1.2}$ | $+0.2^{+0.6}_{-0.5}$ *** | $+0.1^{+0.6}_{-0.5}$ ** |
| ImageNet-1k | $42.4^{+0.7}_{-0.9}$ | $42.0^{+0.9}_{-0.9}$ | $42.2^{+0.6}_{-1.0}$ | $-0.4^{+0.6}_{-0.6}$ *** | $-0.0^{+0.9}_{-0.5}$ |

## G.5 DETAILS ON THE APPLICATION TO BENCHMARK DATASETS

Table 9: Estimated percentage of data quality issues in multiple vision benchmarks obtained using SELFCLEAN's automatic mode with $\alpha = 0.10$ and $q = 0.05$. Images marked as originating from the same person, patient, or lesion were excluded from the near-duplicate count whenever this metadata was available. See appendix M for examples of the problems which were automatically found.

| | **Estimated Errors** | | |
| **Dataset** | Irrelevant Samples | Near Duplicates | Label Errors |
|---|---|---|---|
| DDI | 1 (0.2%) | 4 (0.6%) | 5 (0.8%) |
| PAD-UFES-20 | 0 (0.0%) | 0 (0.0%) | 5 (0.4%) |
| HAM10000 | 0 (0.0%) | 1 (0.6%) | 17 (0.2%) |
| Fitzpatrick17k | 18 (0.1%) | 2,446 (14.8%) | 103 (0.6%) |
| ISIC-2019 | 0 (0.0%) | 1,200 (5.4%) | 97 (0.4%) |
| PatchCamelyon | 98 (0.3%) | 12,845 (3.9%) | 589 (0.2%) |
| CheXpert[4] | 6 (0.0%) | 0 (0.0%) | 303 (0.6%) |
| CelebA | 2 (0.0%) | 810 (0.4%) | 1,033 (0.5%) |
| Food-101N | 310 (0.1%) | 4,433 (1.4%) | 2,728 (0.9%) |
| ImageNet-1k[5] | 0 (0.0%) | 36 (0.1%) | 262 (0.5%) |

---

[4]Refers to the label errors in atelectasis.
[5]Refers to the ImageNet-1k validation set, similar to Northcutt et al. (2021).

## G.6 INVESTIGATION OF FURTHER SYNTHETIC AUGMENATIONS

The synthetic experiment G.1 is here repeated with an additional irrelevant sample augmentation consisting of strong Gaussian blur with a kernel size of 100 to simulate badly out-of-focus pictures. This is applied to dataset images at random turning them into irrelevant samples, unlike the other synthetic contamination strategies for irrelevant samples, i.e. CMED and XR, which add irrelevant samples from other sources.

Results show that most baselines fail to detect strongly blurred samples, except for FastDup, which performs well across contaminations and datasets. This experiment also highlights the benefit of using dataset-specific pre-training as both SimCLR and DINO reach competitive results, whereas the domain-agnostic ImageNet representation achieves significantly lower performance.

Table 10: Performance of various models on the detection of strongly blurred images as irrelevant samples. Evaluation is performed for each of the three considered issue types across two clean benchmark datasets, DDI and HQ-FST. All scores are reported in percentages (%), and P@$k$ and R@$k$ indicate precision and recall at $k$ respectively. Consult section 4 for more details.

| | Method | Contamination 5% | | | | | | | | Contamination 10% | | | | | | | |
| | | DDI + BLUR | | | | HQ-FST + BLUR | | | | DDI + BLUR | | | | HQ-FST + BLUR | | | |
| | | P@20 | R@20 | AUROC | AP | P@20 | R@20 | AUROC | AP | P@20 | R@20 | AUROC | AP | P@20 | R@20 | AUROC | AP |
|---|---|---|---|---|---|---|---|---|---|---|---|---|---|---|---|---|---|
| Irrelevant Samples | IN + IForest | 0.0 | 0.0 | 57.3 | 5.6 | 0.0 | 0.0 | 68.6 | 7.6 | 0.0 | 0.0 | 36.6 | 7.3 | 0.0 | 0.0 | 52.4 | 9.8 |
| | IN + HBOS | 0.0 | 0.0 | 69.3 | 8.0 | 0.0 | 0.0 | 60.1 | 6.2 | 0.0 | 0.0 | 52.0 | 9.4 | 0.0 | 0.0 | 44.7 | 8.5 |
| | IN + DeepSVDD | 0.0 | 0.0 | 58.3 | 5.9 | 0.0 | 0.0 | 76.9 | 10.1 | 0.0 | 0.0 | 30.4 | 6.7 | 10.0 | 3.9 | 53.4 | 13.1 |
| | IN + ECOD | 0.0 | 0.0 | 73.2 | 9.0 | 0.0 | 0.0 | 66.5 | 7.3 | 0.0 | 0.0 | 60.6 | 11.3 | 5.0 | 2.0 | 56.0 | 10.5 |
| | FastDup | 100.0 | 60.6 | 98.9 | 97.2 | 100.0 | 76.9 | 97.5 | 94.5 | 100.0 | 30.8 | 99.4 | 97.8 | 100.0 | 39.2 | 99.7 | 98.0 |
| | SELFCLEAN (IN) | 0.0 | 0.0 | 15.6 | 3.1 | 0.0 | 0.0 | 28.7 | 3.8 | 0.0 | 0.0 | 17.5 | 6.3 | 0.0 | 0.0 | 32.6 | 7.6 |
| | SELFCLEAN (SimCLR) | 50.0 | 30.3 | 81.9 | 35.4 | 70.0 | 53.9 | 75.6 | 53.1 | 100.0 | 30.8 | 100.0 | 99.7 | 100.0 | 39.2 | 100.00 | 100.0 |
| | SELFCLEAN (DINO) | 100.0 | 60.6 | 94.8 | 88.6 | 50.0 | 53.9 | 97.5 | 75.0 | 30.8 | 100.0 | 100.0 | 100.0 | 90.0 | 35.3 | 97.7 | 86.2 |

## G.7 ANNOTATION EFFORT SAVED

When potential data quality issues are verified by a human, it is interesting to quantify the annotation effort saved thanks to the ranking.

The annotation effort saved thanks to the ranking should be considered a function of the residual contamination that can be tolerated in the dataset, i.e. of the recall for data quality issues. We quantify effort using the number of annotations required rather than the actual time spent on the task or concentration, as this is a good proxy which is more directly related to the ranking. Namely, we calculate the fraction of effort (FE) needed to achieve a given recall as the number of annotations needed to achieve it using the ranking divided by the number of annotations needed to achieve it when candidate issues are sorted randomly. The term of comparison is therefore the baseline which sorts the sequence randomly, which always requires the confirmation of a number of examples equal to the target recall times the number of potential issues, since the density of actual issues is uniform along the sequence. The fraction of effort equals one when issue confirmation using the ranking is just as expensive as the baseline. It is below one when the ranking is beneficial for cleaning, and above one when it is detrimental. The best and worst cases are obtained by a ranking algorithm that sorts all positive samples first or last respectively. They obtain FEs equal to $\alpha_+$ and $[1 - (1 - R)\alpha_+]/R$, where R is the recall and $\alpha_+$ is the contamination in the dataset, i.e. the number of actual data quality issues divided by the number of possible data quality issues. Note that the fraction of effort saved by a method compared to another can easily be obtained by dividing the two corresponding FEs.

To summarize the annotation effort savings in a single number, we compute the average fraction of effort (AFE) over all possible recalls, i.e. the area under the FE–R curve. To this end, we proceed as is often done for the computation of average precision, and define

$$\text{AFE} = \sum_i (\text{R}_{i+1} - \text{R}_i)\text{FE}_i. \tag{2}$$

Table 11: Model comparison on the detection of synthetic data quality issues. Evaluation is performed for a mixed contamination strategy on DDI, consisting of adding X-ray images (XR), injecting augmented duplicates (MED), and finally changing labels at random (LBL). Performance is measured in terms of AUROC, AP, and the average fraction of effort (AFE) for a single random seed and 10% synthetic contamination.

|  | Method | ↑ AUROC (%) | ↑ AP (%) | ↓ AFE (%) |
|---|---|---|---|---|
| **Irrelevant Samples** | IN + IForest | 90.3 | 25.3 | 16.8 |
| | IN + HBOS | 91.6 | 30.2 | 14.5 |
| | SELFCLEAN (IN) | 99.6 | 72.2 | 4.6 |
| | SELFCLEAN (DINO) | **100.0** | **100.0** | **3.0** |
| **Near Duplicates** | Method | ↑ AUROC (%) | ↑ AP (%) | ↓ AFE (%) |
| | pHashing | 56.9 | 0.0 | 71.8 |
| | SSIM | 72.5 | 0.0 | 39.6 |
| | SELFCLEAN (IN) | 96.7 | 11.7 | 3.4 |
| | SELFCLEAN (DINO) | **99.6** | **32.5** | **0.4** |
| **Label Errors** | Method | ↑ AUROC (%) | ↑ AP (%) | ↓ AFE (%) |
| | IN + Confident Learning | 85.9 | 51.6 | 18.3 |
| | FastDup | **95.9** | **50.7** | **8.2** |
| | SELFCLEAN (IN) | 92.3 | 50.4 | 11.9 |
| | SELFCLEAN (DINO) | 89.0 | 49.1 | 15.6 |

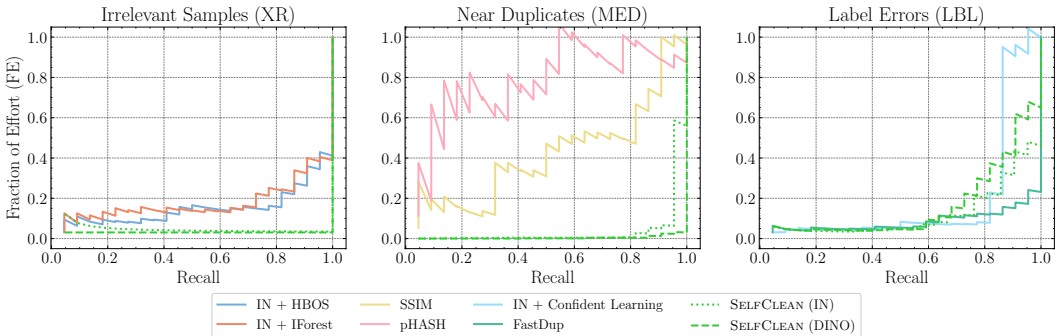

Figure 5: Fraction of effort of the best two competing approaches for each noise type compared to SELFCLEAN for a "mixed-contamination strategy" at 10% level. The artificial dataset is created from DDI by adding X-ray images (XR), injecting augmented duplicates (MED), and finally changing labels at random (LBL). The closer the curves are to zero, the less effort is needed to find data quality issues.

## G.8 MIXED CONTAMINATION EXPERIMENT IN NATURAL IMAGE DOMAIN

In Figure 6 we perform the mixed contamination experiment detailed in Section 4 for a natural image dataset, namely STL-10 (Coates et al., 2011). Different from experiment 2, we are using synthetic augmentations more closely related to the target domain and thus use BLUR to generate irrelevant samples, MED for near-duplicates, and LBL for label errors.

Results show that SELFCLEAN outperforms the competitive baselines for irrelevant samples and near duplicates. For label errors we can see that FastDup is very competitive and SELFCLEAN with DINO is not able to outperform it. However we can also see that SELFCLEAN with ImageNet features is performing on par with FastDup further highlighting the need for indomain knowledge as ImageNet features are more closely related to STL-10.

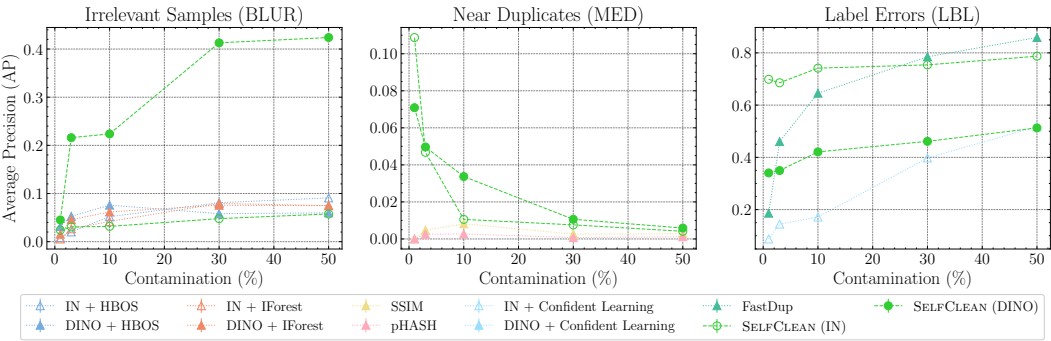

Figure 6: Performance of the best two competing approaches for each noise type compared to SELF-CLEAN measured on two different representations in terms of AP for a "mixed-contamination strategy" when varying the contamination rate. The artificial dataset is created from STL-10 by adding strongly blurred images (BLUR), then injecting augmented duplicates (MED), and finally changing labels at random (LBL).

# H AUTOMATIC CLEANING

In section 3 we constructed scores which take extreme values for candidate problematic data points. Isolating data issues using such a score is essentially a one-dimensional anomaly detection problem. Here we construct a procedure to detect outlier scores which works well in our case, and demonstrate that despite the introduction of two hyperparameters our results do not depend much on arbitrary choices.

## H.1 AUTOMATIC CLEANING PROCEDURE

We start with the intuition that detecting problematic samples is easy if the scores are smoothly distributed for normal data, but are far from the bulk for data with quality issues. However, all our scores range from 0 to 1 and increasingly extreme issues are simply closer and closer to zero without leaving large gaps on the score scale. For this reason, we expand the neighborhoods of 0 and 1 using a logit transformation, $\tilde{s} = \log[s/(1-s)]$. The transformed scores $\tilde{s}$ then range over the whole real axis enabling a better separation between normal and problematic samples.

Since the logit transformation has jacobian $|\,\mathrm{d}\tilde{s}\,/\,\mathrm{d}s\,| = e^{-\tilde{s}}/(1+e^{-\tilde{s}})^2$, under broad assumptions we expect the dominant behavior of the transformed score distribution to drop at least as quickly as a logistic probability density function for $\tilde{s} \to \pm\infty$. Note that this is the case even if the original score distribution is not just constant but presents an integrable power-law singularity for $s \to 0, 1$.

In order to identify the samples which are very unlikely to belong to normal data, we first attempt to isolate a region on the left tail of the distribution which is not affected by noise. To this end, we introduce a hyperparameter $\alpha$, the "contamination rate guess", which represents a generous estimate of the fraction of dirty data in the dataset. For data quality issues where a score is associated to each sample, we simply drop the lowest $\lfloor \alpha_1 N \rfloor$ scores with $\alpha_1 = \alpha$, while when a score is associated to a pair of samples, we discard the lowest $\lfloor \alpha_1 N(N-1)/2 \rfloor$ scores with $\alpha_1 = \alpha^2$. Indeed, in the case where there are no interactions (e.g., only pairs of near-duplicates) we expect $\alpha N$ abnormally low scores, but in the worst case interaction scenario (e.g., all views of the very same sample) we await $\alpha N(\alpha N - 1)/2$ low out-of-distribution scores, which reduces to the above expression for $\alpha_1$ when $\alpha N \gg 1$. Besides dropping the potentially problematic samples, we also select an upper score bound for the range of interest, since we aim at reproducing only the smooth *left* tail of the distribution. Reasonable choices are values between the lower score cutoff determined by $\alpha_1$ and the median, paying attention that enough data is included for sufficient robustness to noise. For this reason, we choose the upper score cutoff to be the quantile corresponding to a fraction of data $\alpha_2$ which is the geometric mean between $\alpha_1$ and $1/2$, i.e. $\alpha_2^2 = \alpha_1/2$. We observe that the range produces robust statistical information if the number of samples is sufficiently large and $\alpha \ll 1/2$, where in practice $\alpha \lesssim 1/4$ is already stable.

Following our heuristic argument, we approximate the smooth component of the left tail of the distribution using a logistic distribution with suitably chosen scale and location parameters, which has probability density function

$$\mathrm{pdf}(\tilde{s}; \mu, \sigma) = \frac{1}{\sigma}\mathrm{pdf}\Big(\frac{\tilde{s}-\mu}{\sigma}\Big), \qquad \mathrm{pdf}(\hat{s}) = \frac{e^{-\hat{s}}}{(1+e^{-\hat{s}})^2}. \tag{3}$$

Given the score cutoffs $\bar{s}_1$ and $\bar{s}_2$ corresponding to the quantiles $\alpha_1$ and $\alpha_2$ of the empirically observed distribution, the scale $\sigma$ and location $\mu$ can be estimated as

$$\sigma = \frac{\bar{s}_2 - \bar{s}_1}{\bar{s}(\alpha_2) - \bar{s}(\alpha_1)}, \qquad \mu = \frac{\bar{s}_1 \bar{s}(\alpha_2) - \bar{s}_2 \bar{s}(\alpha_1)}{\bar{s}(\alpha_2) - \bar{s}(\alpha_1)}, \qquad \bar{s}(\alpha_m) = \log\frac{\alpha_m}{1-\alpha_m} \quad \text{for } m = 1, 2. \tag{4}$$

Here $\bar{s}(\alpha_m)$ indicates the percentage point function of the logistic distribution, i.e. the inverse of its cumulative distribution function. Note that the whole estimation procedure for the left tail of the distribution relies exclusively on quantiles and is therefore naturally robust to outliers.

With an estimate of the smooth score distribution for normal data, we can identify abnormal samples by requesting that they be very unlikely generated by the same random process. This is achieved by demanding that the probability of obtaining a score below an outlier cutoff $s_{\mathrm{cut}}$ be less than a significance level $q$ times the expected fraction of outliers, which is $2\alpha/(N-1)$ in the case of pairs of samples and $\alpha$ otherwise. We set the hyperparameter $q$ to 0.05 corresponding to a 95% one-sided

confidence level and study the influence of this choice in section H.4. All samples with scores lower than the outlier cutoff will be then classified as problematic.

The value of the procedure described above lies in the fact that, despite requiring several engineering steps and introducing two additional hyperparameters, it detects a number of outliers which is largely independent of any reasonable choice for $\alpha$ and $q$. The remaining parts of appendix H are dedicated to showing that the procedure is intuitive and assumptions are empirically acceptable (H.2), and to demonstrating that detected outliers exhibit low sensitivity to the contamination rate guess $\alpha$ (H.3) and to the significance level $q$ (H.4).

## H.2 AUTOMATIC CLEANING EXAMPLES

In figure 7 we illustrate the fit to the left tails of distributions for representative datasets, together with the relevant range used to estimate scale and location, and the position of the outlier cutoff to classify data quality issues. We observe that the probability density function is a qualitatively good estimate of the density-normalized histograms in the expected range, i.e. for the smooth component of the histogram's left tail, within sampling uncertainties. The fit quality is somewhat lower for irrelevant samples, probably due to the score range which is all above $\tilde{s} = 0$. We also carried out experiments with a gaussian functional form for score distribution tails, and observed only minor changes which resulted in a slightly reduced number of detected problems.

## H.3 INFLUENCE OF THE CONTAMINATION RATE GUESS $\alpha$

In figure 8 we analyze the sensitivity of the number of detected data quality problems on the contamination rate guess $\alpha$, for all noise types and representative datasets analyzed in this paper. In these plots, the significance level $q$ is fixed to its default value of $0.05$. We observe that the fraction of found problems does not depend much on $\alpha$ over several orders of magnitude, suggesting a sensitivity to this hyperparameter that is approximately vanishing or at most logarithmic. It is by virtue of this reduced dependence that we can fix $\alpha = 0.10$ throughout the rest of the paper, and that fully-automatic cleaning is able to produce stable results with limited prior knowledge of dataset quality.

## H.4 INFLUENCE OF THE SIGNIFICANCE LEVEL $q$

In figure 9 we report the fraction of detected problematic samples as a function of the significance level $q$, for all considered noise types and for representative datasets. We can see that this hyperparameter essentially determines the number of outliers found, which is monotonically increasing with $q$. Moreover, the number of identified issues has, in most cases, a dependence on $q$ which is less than linear. In some cases, especially when the number of detected outliers is below percent level or $q$ approaches 1, we see more severe sensitivity to the specific value. This may be because the empirical score distribution changes more abruptly than estimated by the logistic fit, as happens for irrelevant samples, or because the region immediately below the lower score cutoff $\bar{s}(\alpha_1)$ (which corresponds to $q = 1$) is densely populated almost by construction. It is however clear that $q$ regulates how extreme scores need to be for a sample to be considered problematic. A value of $q = 10^{-3}$ will only select very apparent data quality issues, $q = 1/4$ will almost certainly also include a significant fraction of valid samples, and our choice of $q = 0.05$ strikes a compromise between precision and recall.

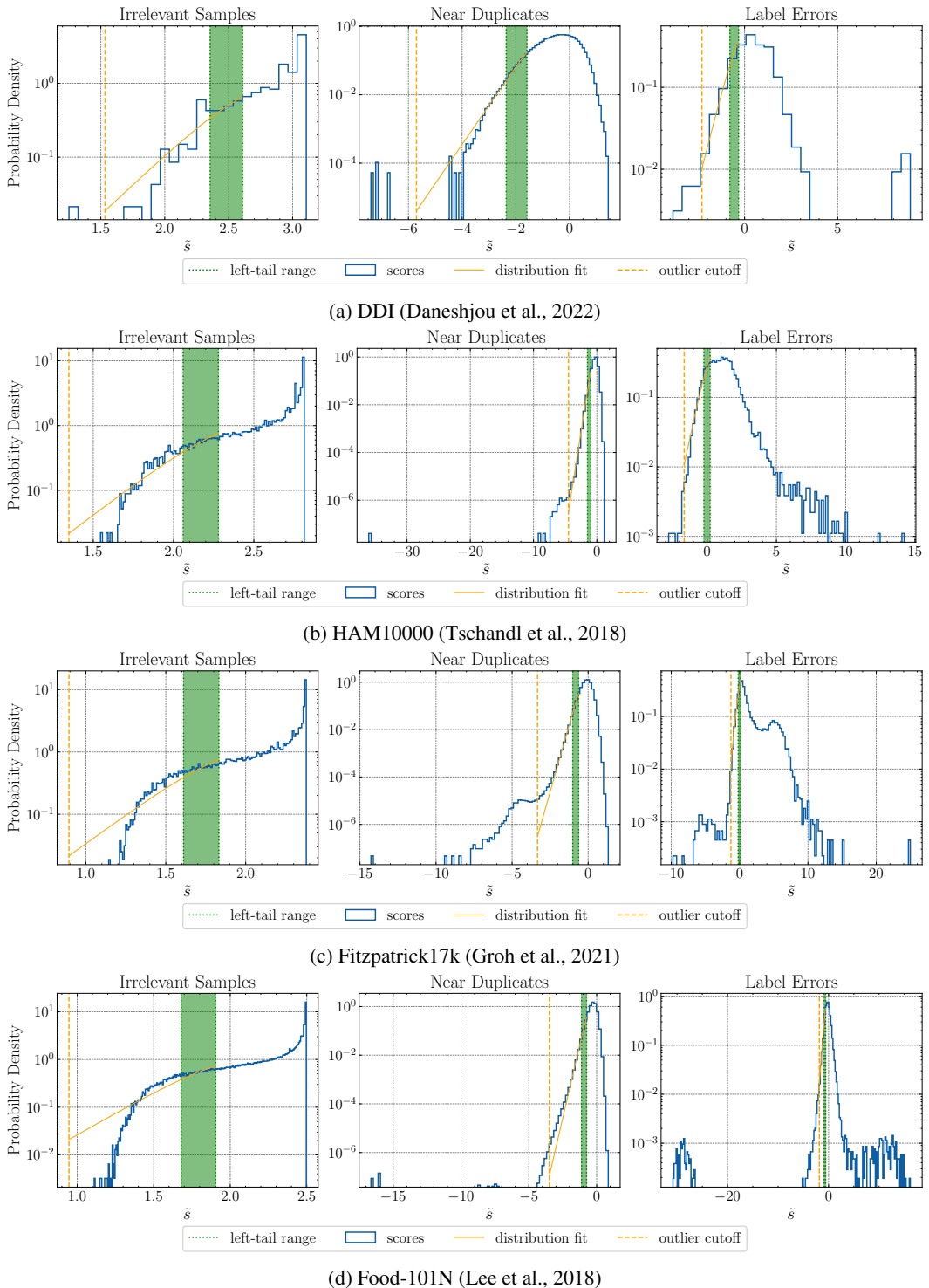

Figure 7: Score histogram (blue) and associated left-tail distribution fit (solid orange) with outlier cutoff (dashed orange) for all considered noise types and representative datasets. The green shaded area represents the range $[\bar{s}_1, \bar{s}_2]$ which is used to determine location and scale of the associated logistic distribution. The values $\alpha = 0.10$ and $q = 0.05$ are used throughout.

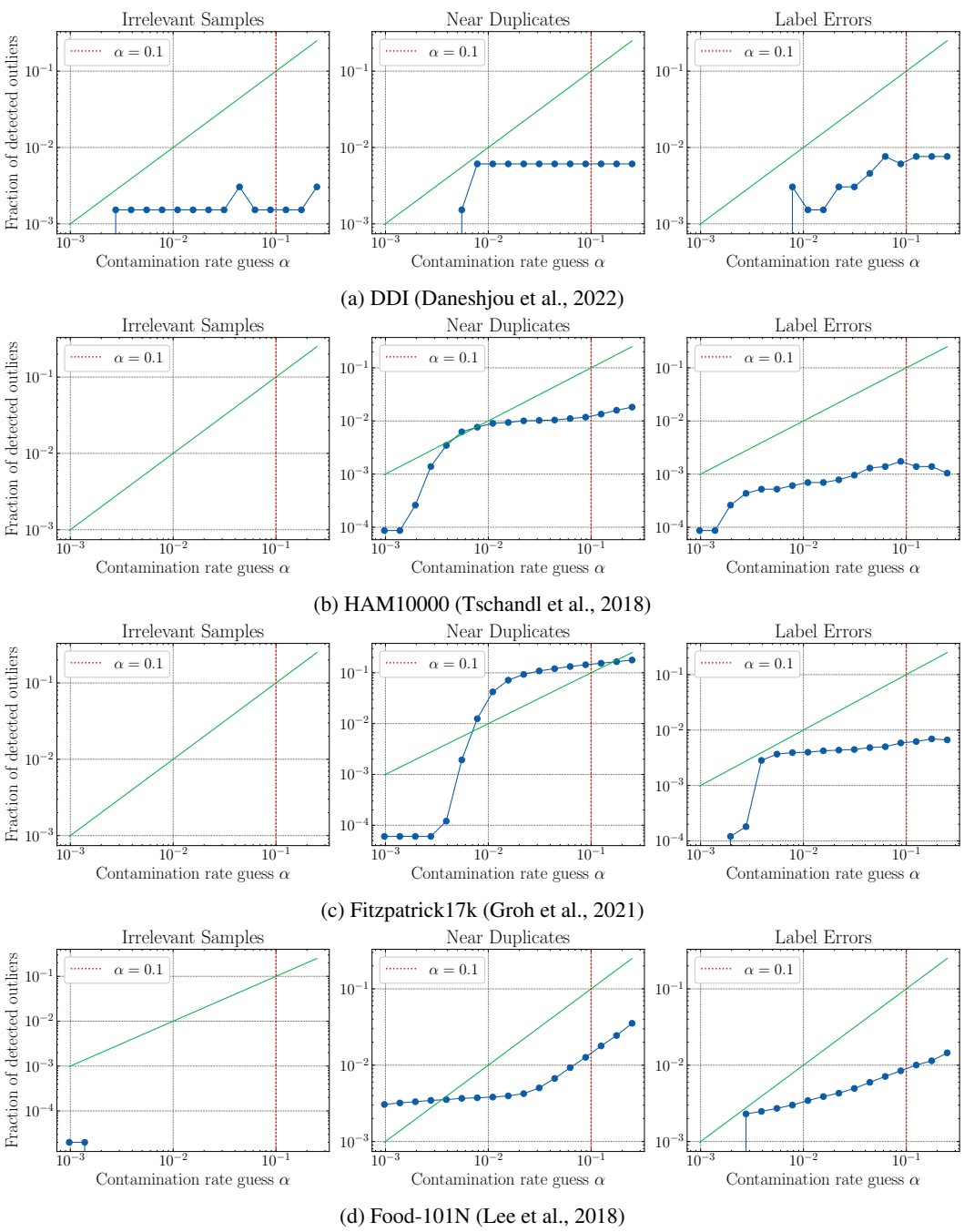

Figure 8: Dependence of the fraction of detected data quality issues on the contamination rate guess $\alpha$ for all considered noise types and representative datasets, at a fixed significance level $q = 0.05$. The observed behavior is reported in blue. It is outside of the lower margin of the plots when no problems are found. The green solid line represents a fraction of detected issues which is equal to the contamination rate guess for reference. The vertical dotted red line indicates the default value $\alpha = 0.10$ used in the rest of the paper.

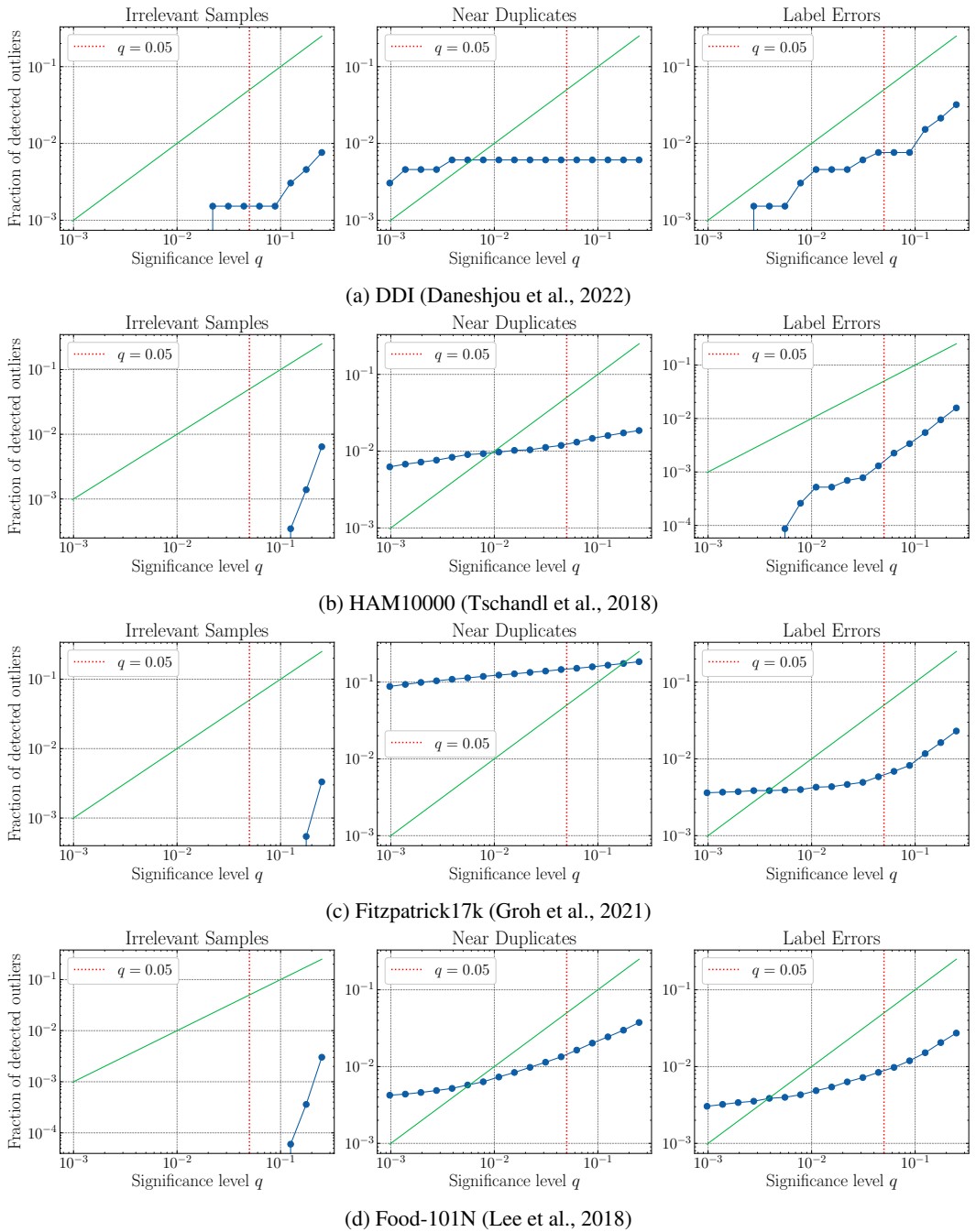

Figure 9: Impact of the choice of the significance level $q$ on the fraction of detected data quality issues, across noise types and for representative datasets, for a fixed contamination rate guess $\alpha = 0.10$. The observed dependency on $q$ is reported in blue, and it is below the lower margin of the plots when no problematic samples are found. The diagonal green solid line is just a reference to guide reading, and the dotted red line indicates the default choice $q = 0.05$.

# I    VALIDATING ALGORITHMIC RANKINGS WITH HUMANS

In this section, we describe the procedure used to confirm that SELFCLEAN assigns low ranks to problematic samples and high ranks to normal data, as discussed in the second part of section 5.2. To this end, for each data quality issue type, we collect human annotations for the first 50 images and for 50 images randomly sampled from the dataset. Annotators use a custom tool which is shown in figure 10. The annotation process starts with the selection of a dataset and data quality issue (e.g. the Fitzpatrick17k dataset and irrelevant samples) and then proceeds with binary questions about single images or pairs thereof depending on the task. Section I.1 shows the task descriptions for each quality issue. Note that samples ranks are not displayed to avoid potential bias.

We paid crowd workers 0.03 US dollars per annotation for images from ImageNet and Food-101N, which roughly corresponds to 9 US dollars per hour. Medical experts were not compensated financially but were instead acknowledged with co-authorship in a labeling consortium.

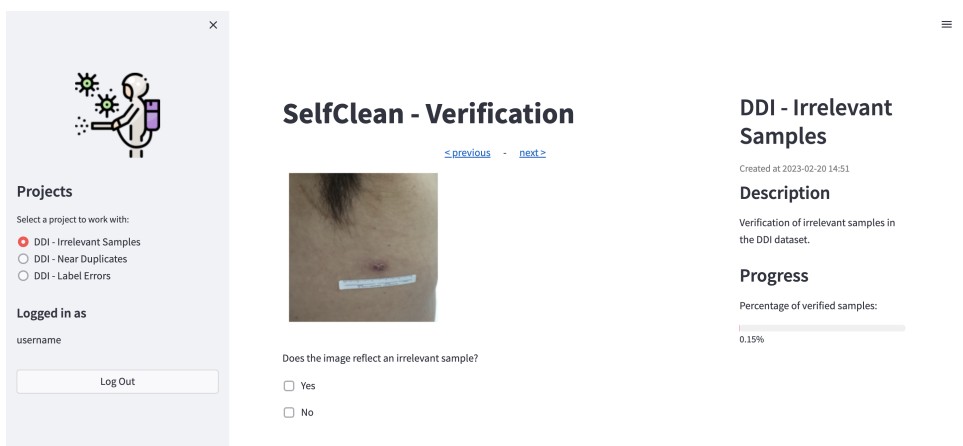

Figure 10: Screenshot of verification tool used by medical experts to annotate data quality issues in the order proposed by SELFCLEAN.

## I.1    TASK DESCRIPTIONS

This section gives detailed information about each task description shown to the annotators:

- Irrelevant samples: "Your task is to judge if the image shown is irrelevant. Select *yes* when the image is not a valid input for the task at hand."

- Near duplicates: "Your task is to judge whether the two images shown together are pictures of the same object. Note that pictures of the same object can be identical or different shots with the same object of interest."

- Label errors: "Your task is to judge whether the image's label is correct. Please select that the label is an error only if you think it is wrong and not when there is low uncertainty or ambiguity."

## I.2    DETAILED RESULTS

In order to verify that problematic samples tend to appear first in the ranking provided by SELFCLEAN, for each noise type, we first consider all 100 annotated images and then the first 50 in the ranking only. We conduct Mann-Whitney $U$ statistical tests to verify that humans are more likely to identify data quality issues in samples that appear first in the SELFCLEAN ranking. In order to gain more intuitive understanding, we also report the fraction of samples that were found to be problematic within the first 50 and the 50 random samples, and within samples ranked 1 through 25 and 26 through 50. Finally, we visualize the distribution of human-confirmed problems through the ranking by plotting the fraction of confirmed problems in a rolling window of ten ranks in figure 11.

Table 12: Comparison of the percentage of errors found by ~~experts~~ humans in the 50 lowest-ranked samples with 50 random samples, and in samples 1 to 25 with samples 26 through 50. We report the percentage of errors in each sample and the corresponding $p$-value of a Mann–Whitney $U$ test, which represents the probability for the ranking to be unrelated to the position of positive samples.

| Dataset | Data Quality Issue | Lowest 1-50 (%) | Random Sample (%) | $p$-value | Lowest 1-25 (%) | Lowest 26-50 (%) | $p$-value |
|---|---|---|---|---|---|---|---|
| | | **Percentage of Human-Confirmed Problems** | | | | | |
| DDI | Irrelevant Samples | 12 | 8 | 0.25 | 20 | 4 | **0.04** |
| DDI | Near Duplicates | 12 | 0 | **0.006** | 24 | 0 | **0.005** |
| DDI | Label Errors | 22 | 32 | 0.86 | 20 | 24 | 0.63 |
| Fitzpatrick17k | Irrelevant Samples | 14 | 4 | **0.04** | 12 | 16 | 0.65 |
| Fitzpatrick17k | Near Duplicates | 100 | 0 | $\mathbf{1.3 \times 10^{-23}}$ | 100 | 100 | undef |
| Fitzpatrick17k | Label Errors | 54 | 12 | $\mathbf{4.4 \times 10^{-6}}$ | 52 | 56 | 0.61 |
| ImageNet | Irrelevant Samples | 62 | 48 | 0.08 | 56 | 68 | 0.80 |
| ImageNet | Near Duplicates | 92 | 0 | $\mathbf{2.1 \times 10^{-20}}$ | 100 | 84 | **0.02** |
| ImageNet | Label Errors | 36 | 0 | $\mathbf{1.6 \times 10^{-6}}$ | 48 | 24 | **0.04** |
| Food-101N | Irrelevant Samples | 24 | 4 | **0.002** | 36 | 12 | **0.02** |
| Food-101N | Near Duplicates | 100 | 0 | $\mathbf{1.3 \times 10^{-23}}$ | 100 | 100 | undef |
| Food-101N | Label Errors | 72 | 34 | $\mathbf{7.6 \times 10^{-5}}$ | 80 | 64 | 0.61 |

We observe very good performance for near-duplicate detection throughout the considered datasets. Label-error identification is very convincing in all cases but for DDI. The different concentration of problems is mostly observed between images with low ranking and random samples, while the difference between samples 1-25 and 26-50 is less pronounced. We observe that identifying label errors in a highly-curated dataset such as DDI is a highly-nontrivial task which might exceed the design of our experiment. Finally, the detection of irrelevant samples is the task where SELFCLEAN achieves the lowest agreement with human annotators. Nevertheless, our results still suggest a significant separation of irrelevant samples within the ranking in at least half of the cases.

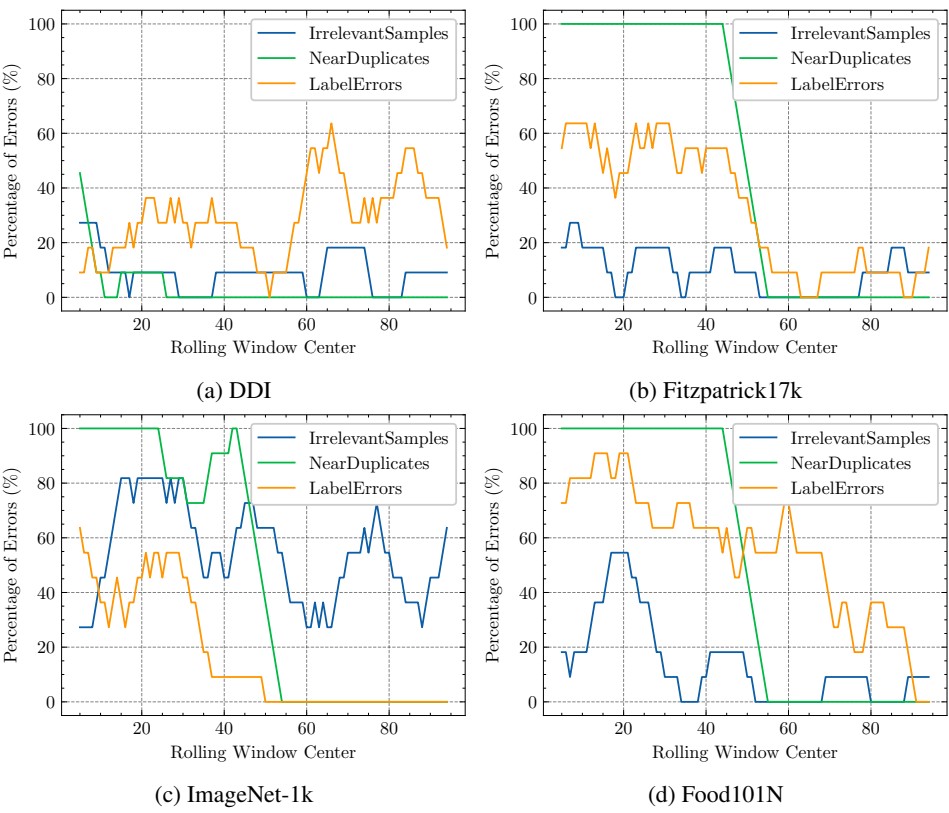

(a) DDI

(b) Fitzpatrick17k

(c) ImageNet-1k

(d) Food101N

Figure 11: Visualization of the percentage of quality issues found across the first 50 samples in the SELFCLEAN ranking and in 50 random samples, using a rolling window of size 10. Results are reported across four datasets and for each noise type.

## J  Scoring for irrelevant samples

This section describes how to construct a score based on hierarchical clustering, such that samples with a high probability of being irrelevant have significantly lower values compared to the bulk of the data. Note that, although in practice we use single-linkage agglomerative clustering, our heuristic construction can be applied to any hierarchical clustering and is formulated accordingly.

Hierarchical clustering over a set of ~~leaves~~ data points numbered $\{1, \ldots, N\}$ ~~is defined by~~ can be represented with a hierarchy of sets $C_n$ which specify clusters at each step of the algorithm. Here $n$ is chosen to indicate the number of clusters at a specific step, $C_n = \{C_{1n}, \ldots, C_{nn}\}$ ~~for~~, in which case $n$ ~~going~~ runs from $N$ to 1 as the algorithm proceeds and more data points are merged. Without loss of generality, ~~we assume~~ it is always possible to rename data points such that indices of merged sets are always consecutive, and the other sets in $C_n$ do not change their relative order

$$C_{in} = \begin{cases} C_{i(n+1)} & \text{if } i < i_n, \\ C_{i(n+1)} \cup C_{(i+1)(n+1)} & \text{if } i = i_n, \quad \text{for } i = 1, \ldots, n \text{ and } n = 1, \ldots, N. \\ C_{(i+1)(n+1)} & \text{if } i > i_n, \end{cases} \quad (5)$$

With this notation, the clusters merged from step $n + 1$ to step $n$ are the one indexed by $i_n$ and $i_n + 1$. The hierarchy of sets $C_n$ induces a dendrogram, i.e. a tree graph where each cluster is a node connected to its direct parent and children. Each element $n$ of the hierarchy (except for the first one, $C_N$) ~~is~~ can also be associated with a ~~merge~~ distance ~~$d_{n-1} = d(C_{i_n n}, C_{(i_n+1)n})$~~ $d_n$ which is the one at which the last two clusters were merged, $d_n = d(C_{i_n(n+1)}, C_{(i_n+1)(n+1)})$. To define a ranking, we sort the dendrogram such that ~~$|C_{i(n+1)}| \leq |C_{(i+1)(n+1)}|$~~ at every step $|C_{i_n(n+1)}| \leq |C_{(i_n+1)(n+1)}|$, i.e. the cluster which contains the least leaves comes first, based on the idea that outliers are associated with merges containing fewer leaves (Jiang et al., 2001). In case of ties, the cluster which was created at the largest distance precedes the other.

To produce a scor~~ing~~ e for each ~~sample from~~ node in the dendrogram, natural building blocks are the merge distance, the sizes of the merged clusters, and their interactions (Tokuda et al., 2020). Accordingly, we define scores by drawing the dendrogram in a $[0, 1] \times [0, 1]$ square where the horizontal axis is one minus the (merge) distance $d$ and the vertical axis is the weight $w_{in}$ of cluster $in$ which is defined recursively below. Note that the possible values for the distance range from 0 to 1, which can be achieved in general with a simple transformation, and also guarantees that $1 - d$ spans the same range. This graphical construction is illustrated in the right panel of figure 12. The score of each leaf ~~$s_{is}(e_i)$~~ is determined ~~as~~ at each merge distance $d$ by the weight of the cluster $C_{jn}$ it belongs to between merge distance $d_n$ and $d_{n-1}$. Formally, the irrelevant sample score $s_{is}(e_i)$ is then given by the area under the curve $f_i(d)$ where

$$f_i(d) = w_{jn} \quad \text{if} \quad d_n \leq d < d_{n-1} \quad \text{and} \quad i \in C_{jn}, \qquad n = 1, \ldots, N, \qquad (6)$$

with $d_N = 0$ and $d_0 = 1$. For convenience, we define $p_{in} = |C_{in}|/N$ to be the probability of cluster $in$ and set $w_{0n} = 0$ and $w_{11} = 1$. To define the weights, we propose a rule which we call leaves and distances (LAD) and reads

$$w_{i(n+1)} = \begin{cases} w_{in} & \text{if } i < i_n, \\ w_{(i_n-1)n} + (w_{i_n n} - w_{(i_n-1)n})p_{i(n+1)}/p_{i_n n} & \text{if } i_n \leq i \leq i_n + 1, \\ w_{(i-1)n} & \text{if } i > i_n + 1. \end{cases} \qquad (7)$$

Essentially, at each split, children cluster $i$ receives a weight $w_{i(n+1)}$ which is proportional to its relative size $p_{i(n+1)}/p_{i_n n}$ with respect to the parent cluster, while bound between the previous cluster weight $w_{(i_n-1)n}$ and the parent cluster weight $w_{i_n n}$. This completes our construction of the scores $s_{is}(e_i)$ for irrelevant samples.

We note that although the formulation given here is limited to the case of binary merges and no ties among distances, generalization to these cases is straightforward except for resolving how to sort the clusters.

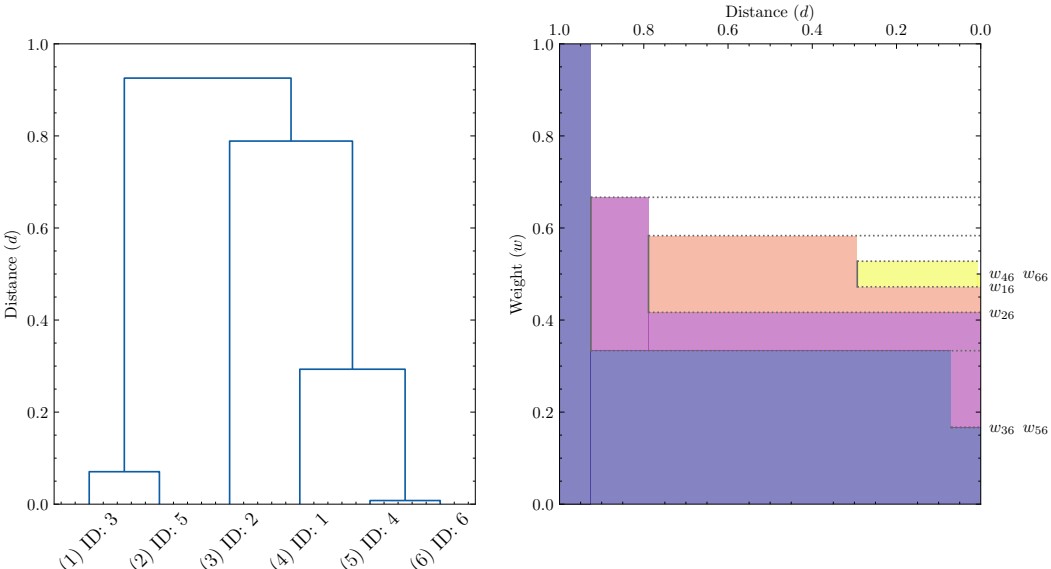

Figure 12: The left plot shows an example of a dendrogram for hierarchical clustering, and the right plot an illustration of the LAD scoring. In the left plot, the x-axis shows the ranking of the different points in brackets and the corresponding identification number. In the right plot, the right side of the y-axis shows the weights $w_{in}$ corresponding to equation 7.

## K HYPERPARAMETERS

In this section we specify parameter values used throughout the paper for both pre-training and evaluation. Table 13 lists the hyperparameters used for pre-training for both DINO and SimCLR. The used parameters are very similar to the introductory papers of these approaches (Chen et al., 2020; Caron et al., 2021) with the exception that the global crop scale is larger and there are more local crops, which we have found to be beneficial for smaller datasets ($< 20,000$). However, we observed no benefit or harm for larger datasets with these minor hyperparameter changes. Table 14 lists the configuration for producing near-duplicate images using the MedAugment and ArtefactAugment strategies. The configuration was chosen to mimic the natural contamination of near duplicates in benchmark datasets.

Table 13: Hyperparameters used for pre-training using SimCLR and DINO on the dataset to clean. Here "-" indicates that the respective parameter is not used for the corresponding pre-training strategy. Parameters not given in this list follow the introductory paper. More detailed information about the hyperparameters can be found in the open-sourced codebase.

| Hyperparameter | SimCLR | DINO |
|---|---|---|
| Batch size | 90 | 64 |
| Epochs | 100 | 100 |
| Optimizer | Adam | AdamW |
| Learning rate | 0.001 | 0.0005 |
| Min. learning rate | 1e-6 | 1e-6 |
| Weight decay | 0.04 | 0.04 |
| Weight decay end | 0.4 | 0.4 |
| Warmup epochs | 10 | 10 |
| Momentum teacher | - | 0.996 |
| Clip grad. | 3.0 | 3.0 |
| Base model | ViT tiny | ViT tiny |
| Model embedding dim. | 192 | 192 |
| Model output dim. | 128 | 4096 |
| Model patch size | 16 | 16 |
| Model drop path rate | 0.1 | 0.1 |
| Norm last layer | - | True |
| Global crops scale | - | (0.7, 1.) |
| Local crops scale | - | (0.05, 0.4) |
| Global crops number | - | 2 |
| Local crops number | - | 12 |
| Warmup teacher temp. | - | 0.04 |
| Teacher temp. | - | 0.04 |
| Warmup teacher temp. epochs | - | 30 |
| Contrastive temp. | 0.5 | - |

Table 14: Parameter values used for the MedAugment (MED) (14a) and ArtefactAugment (ARTE) (14b) strategies. More detailed configurations can be found in the open-sourced codebase.

(a) MedAugment (MED)

| Hyperparameter | MED |
|---|---|
| Rotation proba. | 0.5 |
| Padding proba. | 0.5 |
| Blur proba. | 0.5 |
| Rotation degree range | (0, 180) |
| Scale range | (0.5, 0.9) |
| Padding | 3 |
| Gaussian kernel size | 5 |

(b) ArtefactAugment (ARTE)

| Hyperparameter | ARTE |
|---|---|
| Watermark proba. | 0.5 |
| Colorbar proba. | 0.5 |
| Collage proba. | 0.5 |
| Watermark max. scale | 0.5 |
| Collage max. scale | 0.5 |

## L    VISUALIZATION OF ARTIFICIAL AUGMENTATION STRATEGIES

This section visualizes the different synthetic contamination strategies used for the evaluation of 5.1. Figure 13 visualizes both near-duplicate augmentation strategies, namely MedAugment (MED) and ArtefactAugment (ARTE). Figure 14 shows a random sample of the irrelevant images from the combined medical images (CMED) strategy added to a dataset for contamination. The gallery further shows some irrelevant X-ray images from the XR augmentation strategy.

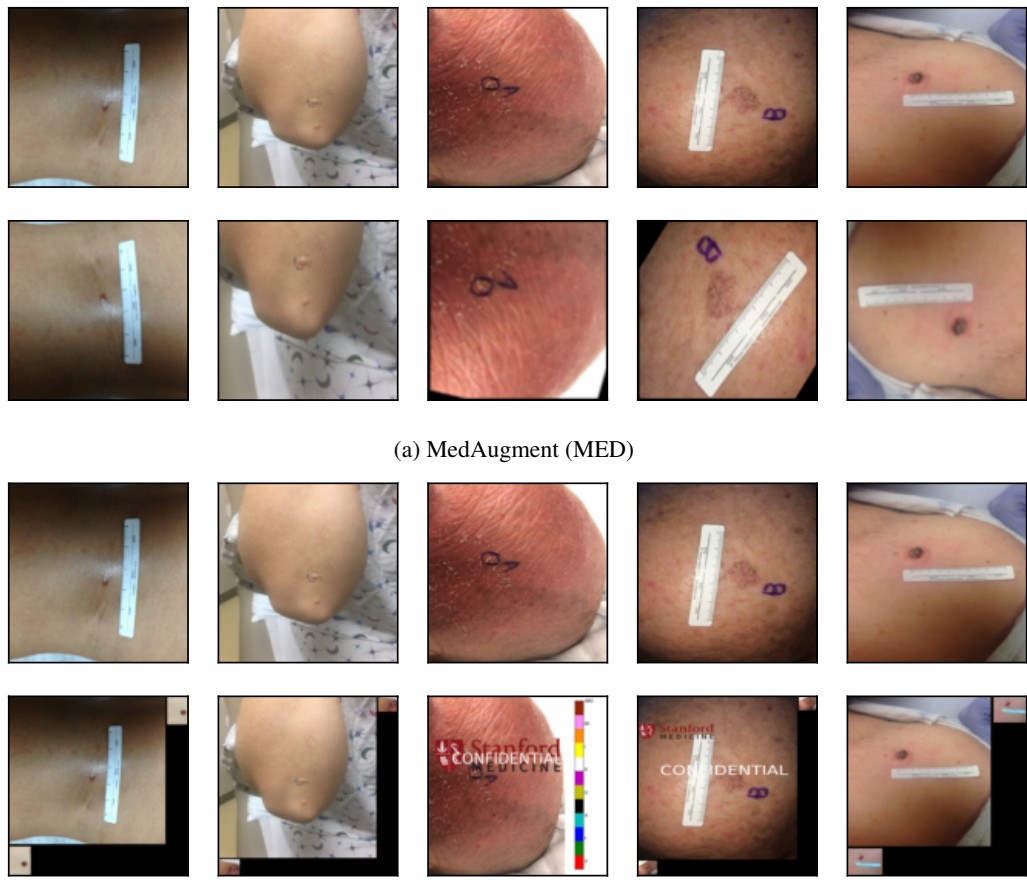

(a) MedAugment (MED)

(b) ArtefactAugment (ARTE)

Figure 13: Random samples of near duplicates produced by synthetic augmentation strategies and used to evaluate near-duplicate detection. Figure 13a shows examples of the MedAugment strategy (MED), consisting of random rotation, flipping, resizing, applying padding, and Gaussian blur. Figure 13b shows examples of the ArtefactAugment strategy (ARTE), which consists of adding typical artifacts found in medical image collections. This includes adding watermarks, color bars, and rulers and then scaling the image and randomly adding additional images to create a collage. Images one, two, four, and five show examples of collages.

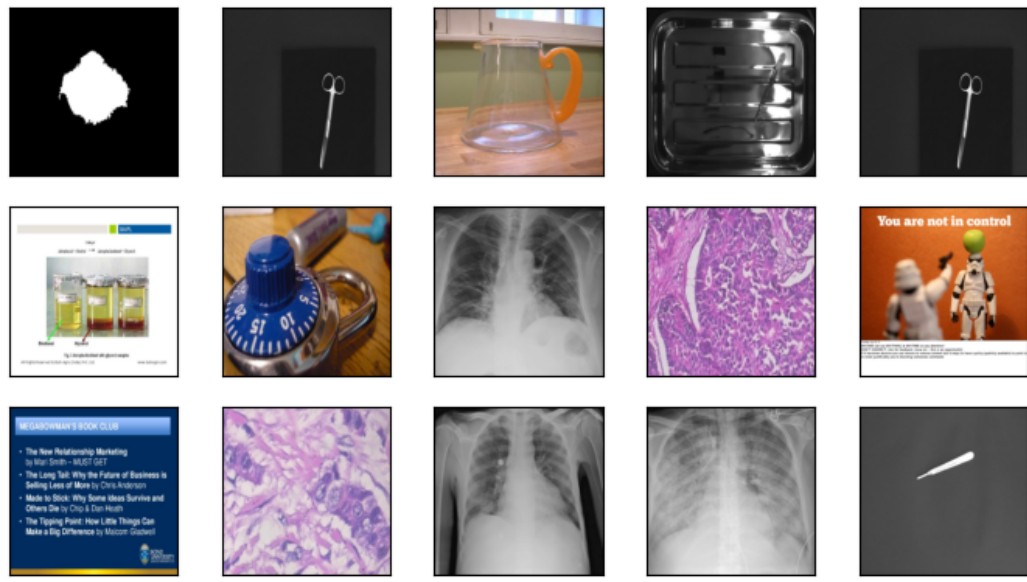

Figure 14: A random sample of irrelevant samples from the *CMED* contamination strategy. The images originate from multiple datasets, consisting of surgical tools (Lavado, 2018), X-ray images (Ozturk et al., 2020), ImageNet samples (Deng et al., 2009), histopathological images (Spanhol et al., 2016), segmentation masks (Codella et al., 2019) and pictures of PowerPoint slides (Araujo et al., 2016).

## M    INSPECTION OF BENCHMARK DATASETS

This section contains illustrations of the rankings produced by SELFCLEAN for multiple vision benchmarks, divided into the three main noise categories of irrelevant samples, near duplicates, and label errors.

Applying SELFCLEAN to multiple benchmark datasets across different domains has led to different insights on why some of these data quality issues may occur. Irrelevant samples in the medical domain are often caused by device malfunctions (figure 19 Rank 2-6) or wrong device configuration (figure 19 Rank 7, 8, and 10, figure 22 Rank 1, 5, and 9). Near duplicates can often be traced to data acquisition problems, such as crawling both an image and its thumbnail (figure 24) or the metadata failing to correctly flag that two images have a common origin (figure 15 and 21). The most apparent label errors in our experience are near-duplicate samples with different labels (figure 26 Rank 1&2, 4&6, 9&10, and 8&13), which indicate (understandable) difficulties in the annotation process, or irrelevant samples with a label (figure 20 Rank 2-4), which easily arise in (semi-)automated annotation procedures.

## M.1 IMAGENET-1K

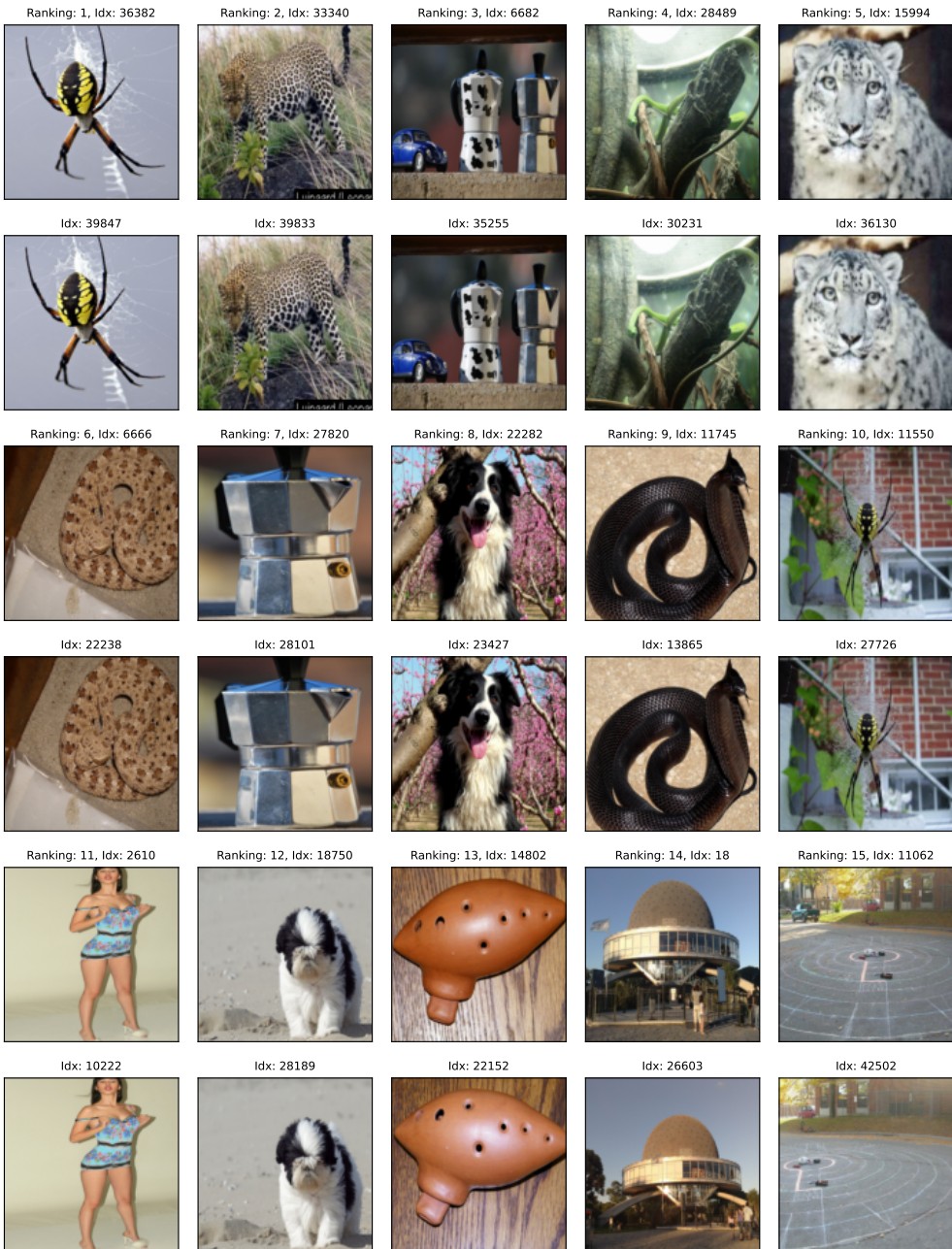

Figure 15: Ranking produced by SELFCLEAN for near duplicates in the ImageNet-1k validation set, of which the top-15 are shown along with the respective rank and index.

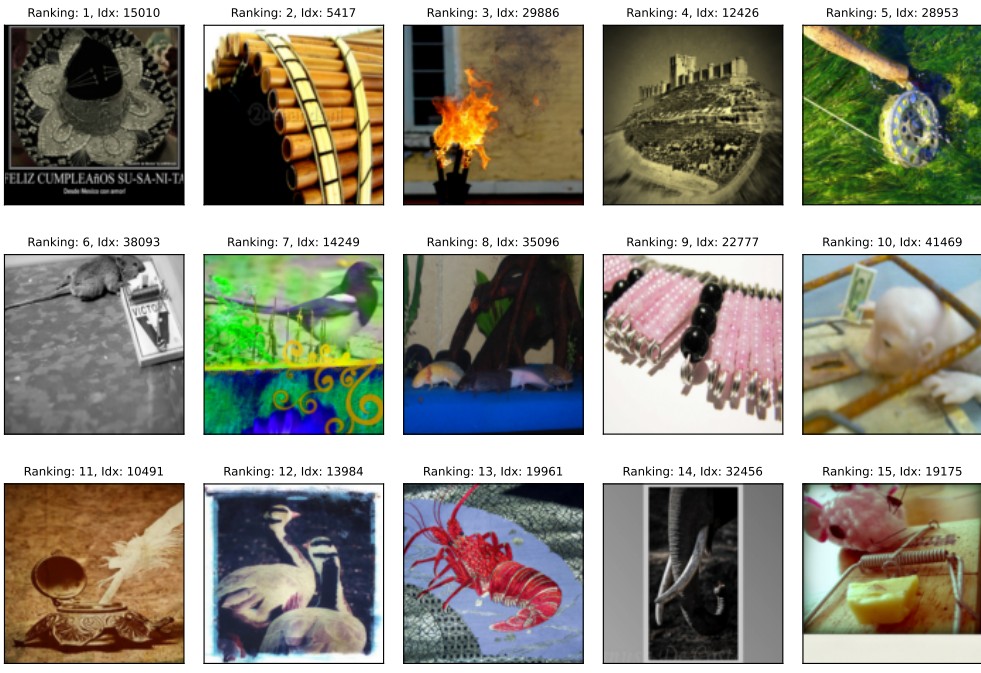

Figure 16: Ranking produced by SELFCLEAN for irrelevant samples in the ImageNet-1k validation set, of which the top-15 are shown along with the respective rank and index.

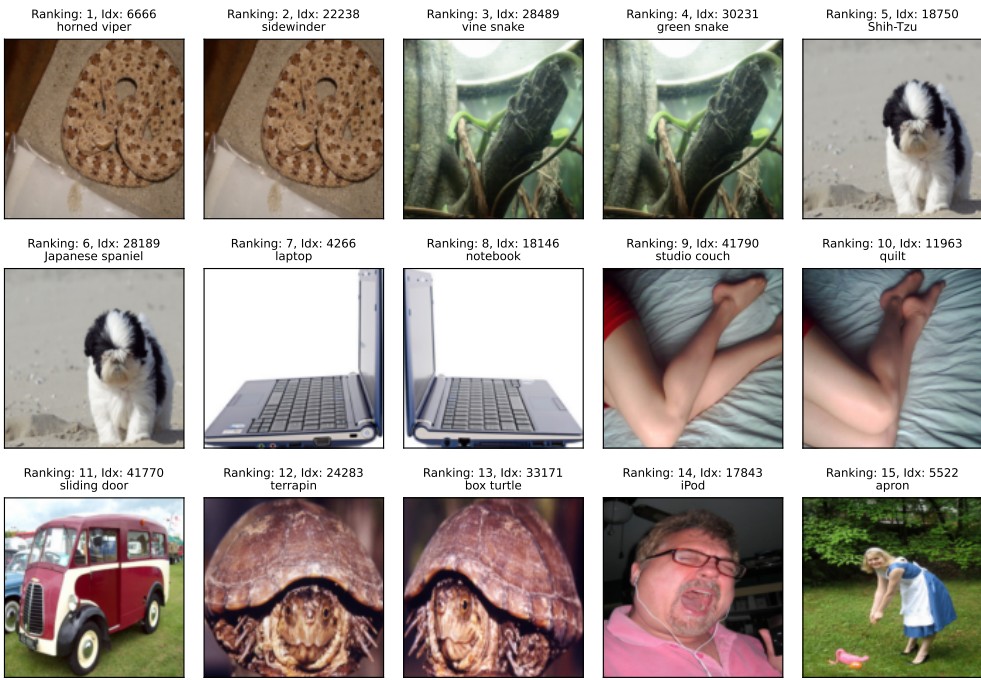

Figure 17: Ranking produced by SELFCLEAN for label errors in the ImageNet-1k validation set, of which the top-15 are shown along with the respective rank, index, and original label.

## M.2 CHEXPERT

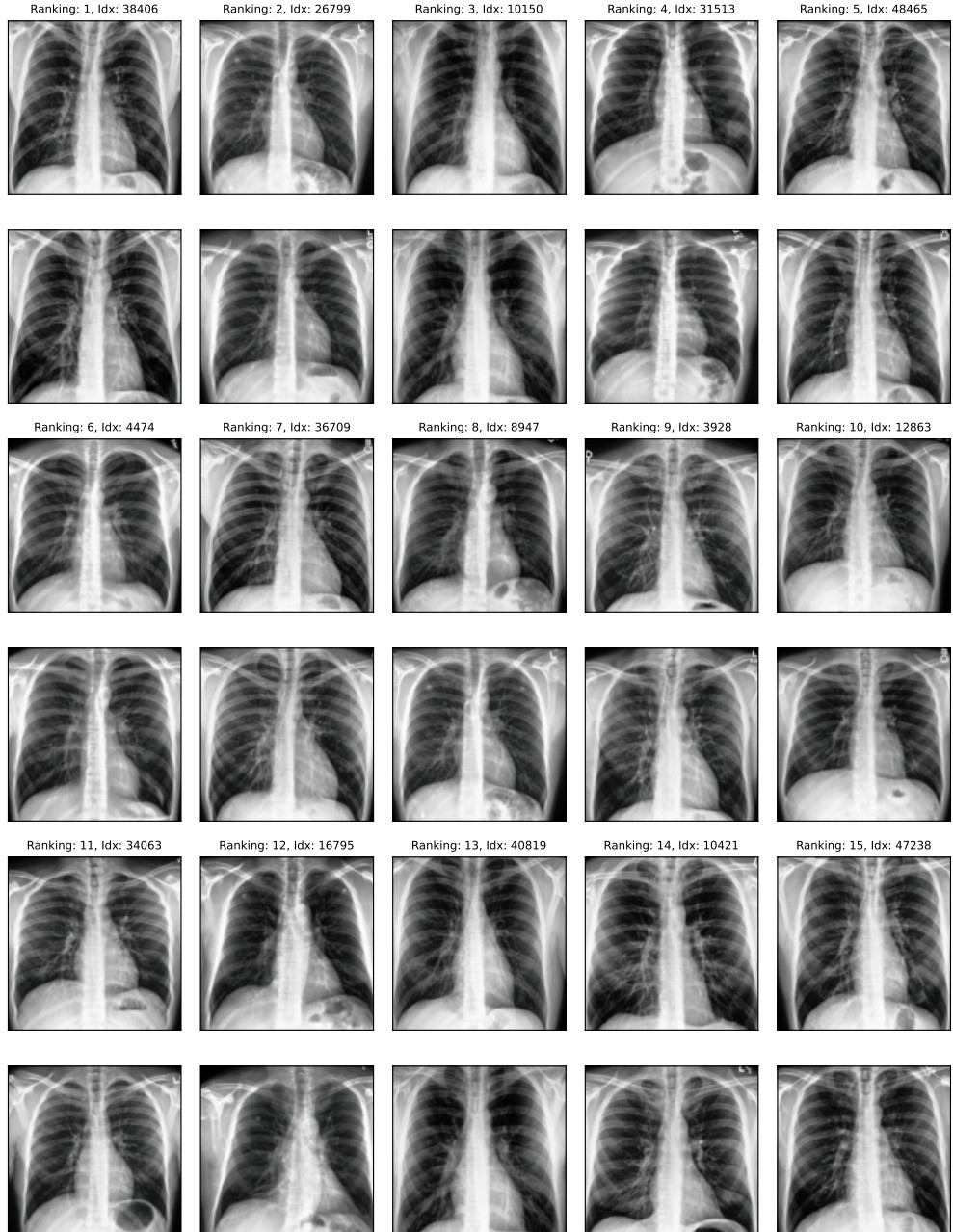

Figure 18: Ranking produced by SELFCLEAN for near duplicates in CheXpert, of which the top-15 are shown along with the respective rank and index.

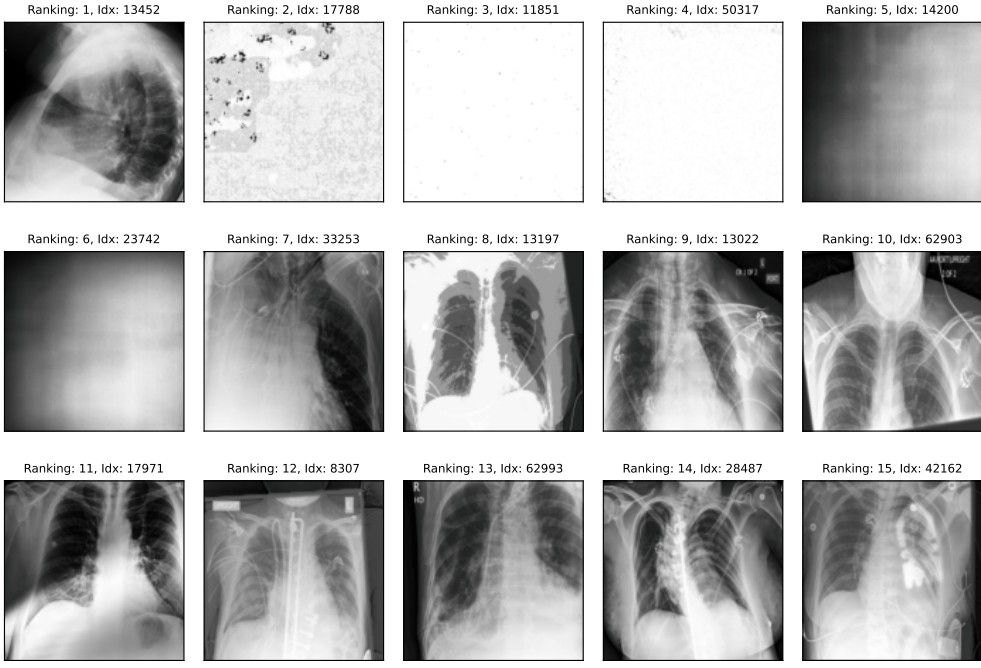

Figure 19: Ranking produced by SELFCLEAN for irrelevant samples in CheXpert, of which the top-15 are shown along with the respective rank and index.

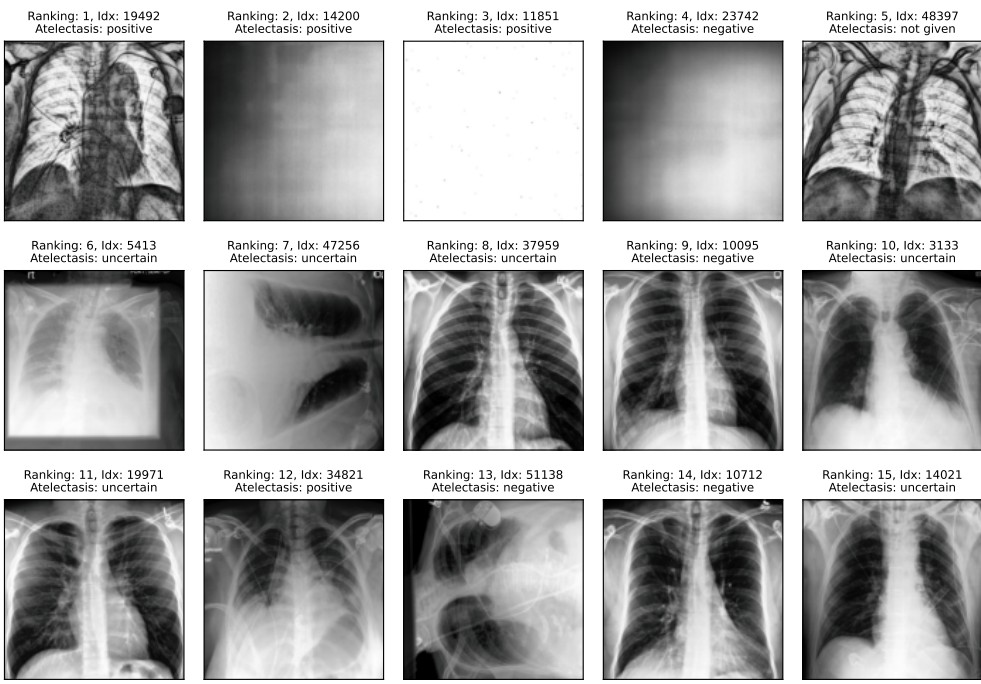

Figure 20: Ranking produced by SELFCLEAN for atelectasis label errors in CheXpert, of which the top-15 are shown along with the respective rank, index, and original label.

## M.3 PATCHCAMELYON

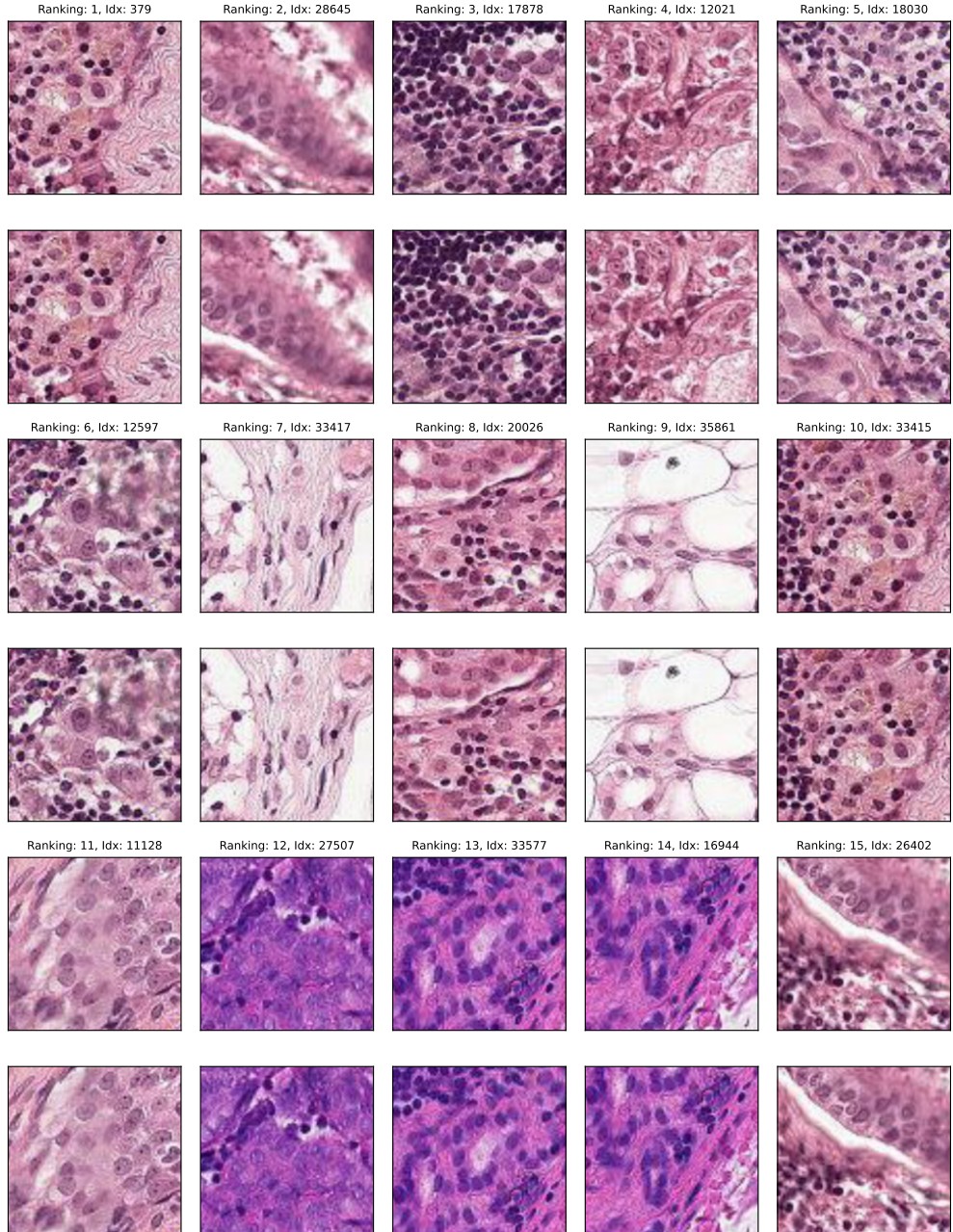

Figure 21: Ranking produced by SELFCLEAN for near duplicates in PatchCamelyon, of which the top-15 are shown along with the respective rank and index.

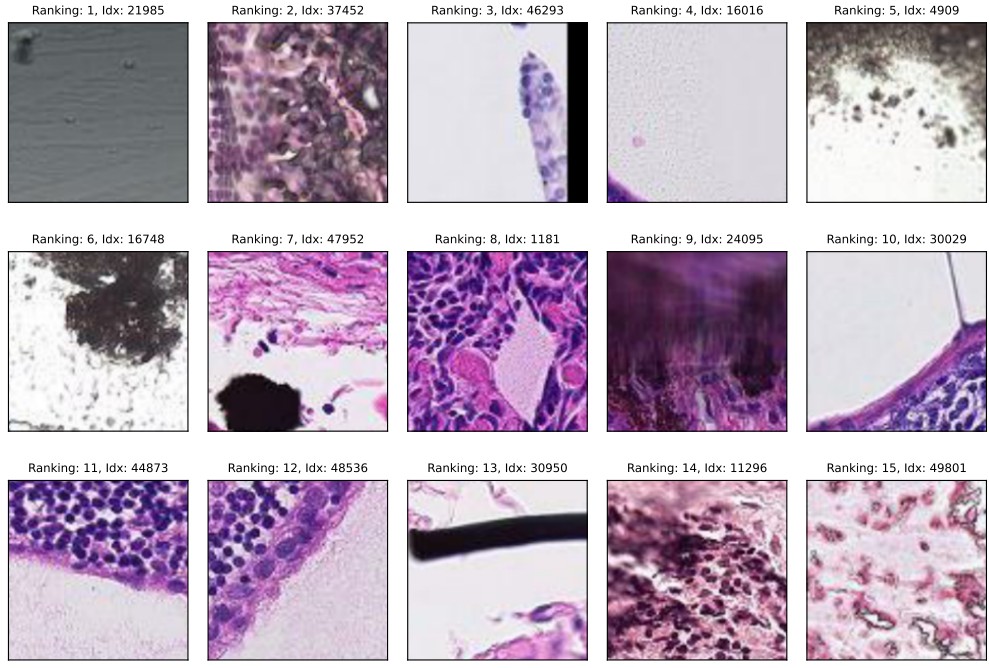

Figure 22: Ranking produced by SELFCLEAN for irrelevant samples in PatchCamelyon, of which the top-15 are shown along with the respective rank and index.

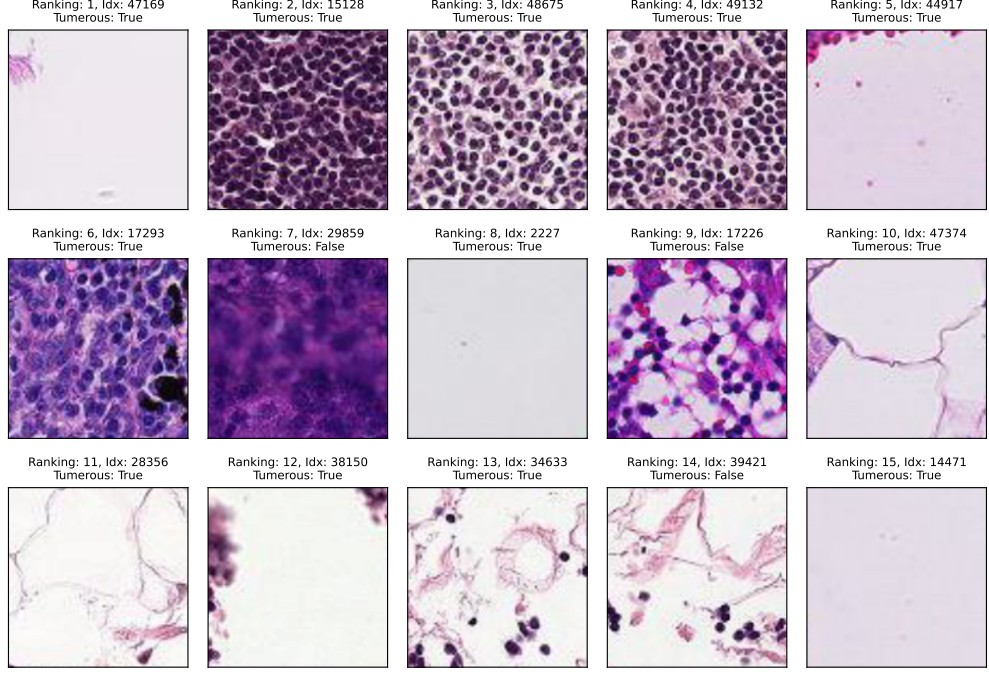

Figure 23: Ranking produced by SELFCLEAN for label errors in PatchCamelyon, of which the top-15 are shown along with the respective rank, index, and original label, i.e. if the patch is tumerous.

## M.4 FITZPATRICK17K

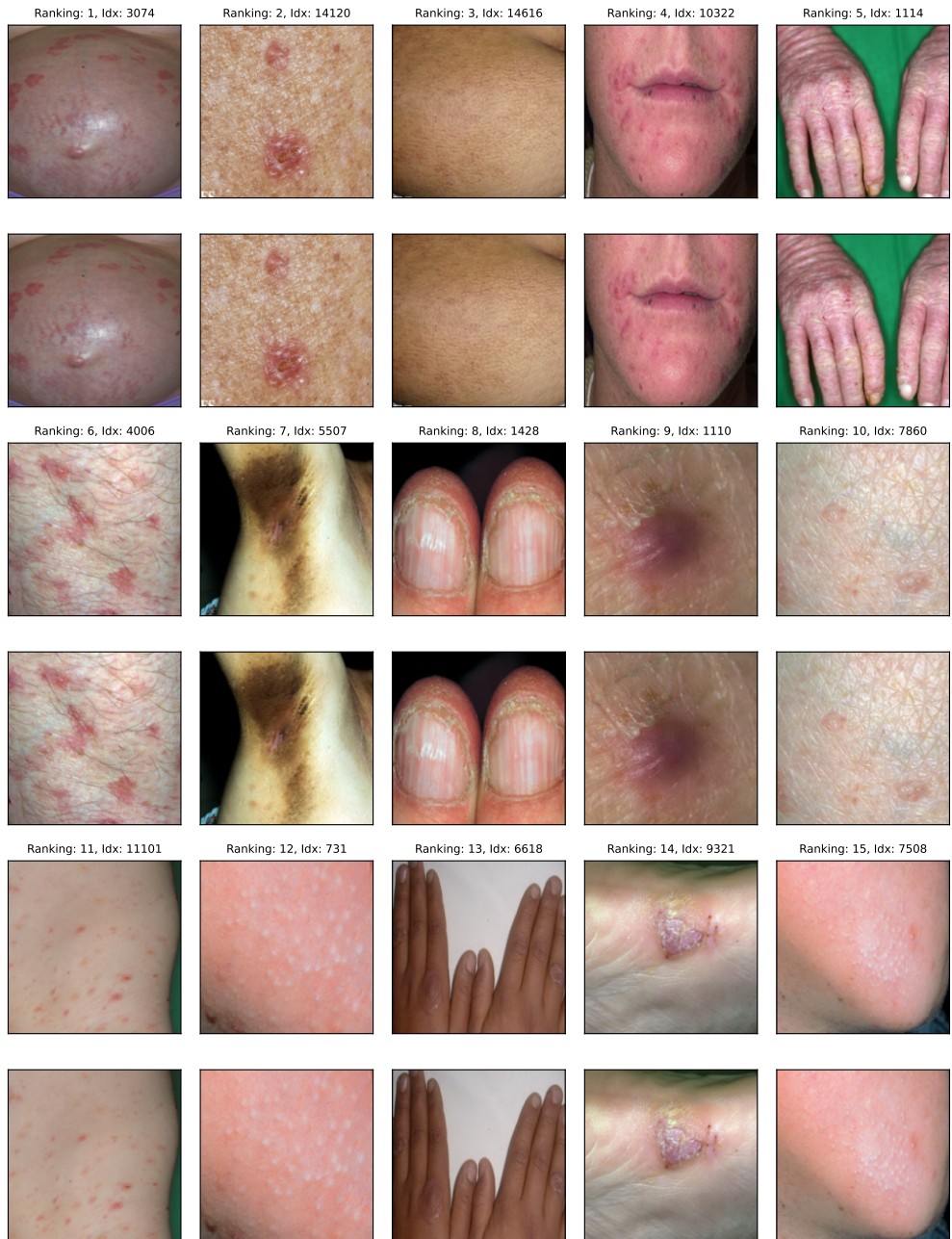

Figure 24: Ranking produced by SELFCLEAN for near duplicates in the Fitzpatrick17k, of which the top-15 are shown along with the respective rank and index.

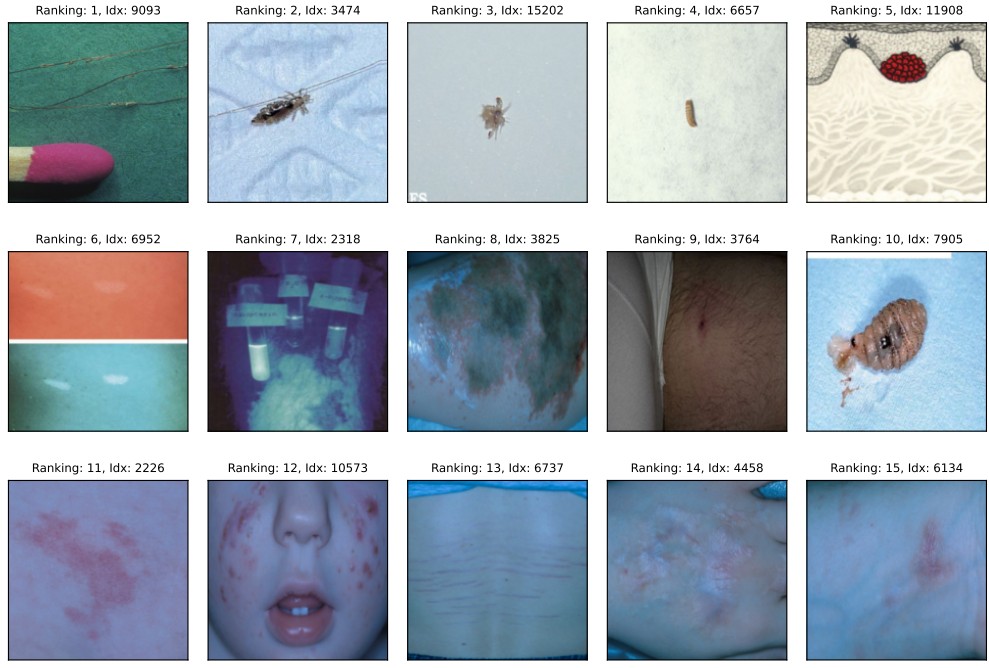

Figure 25: Ranking produced by SELFCLEAN for irrelevant samples in the Fitzpatrick17k, of which the top-15 are shown along with the respective rank and index.

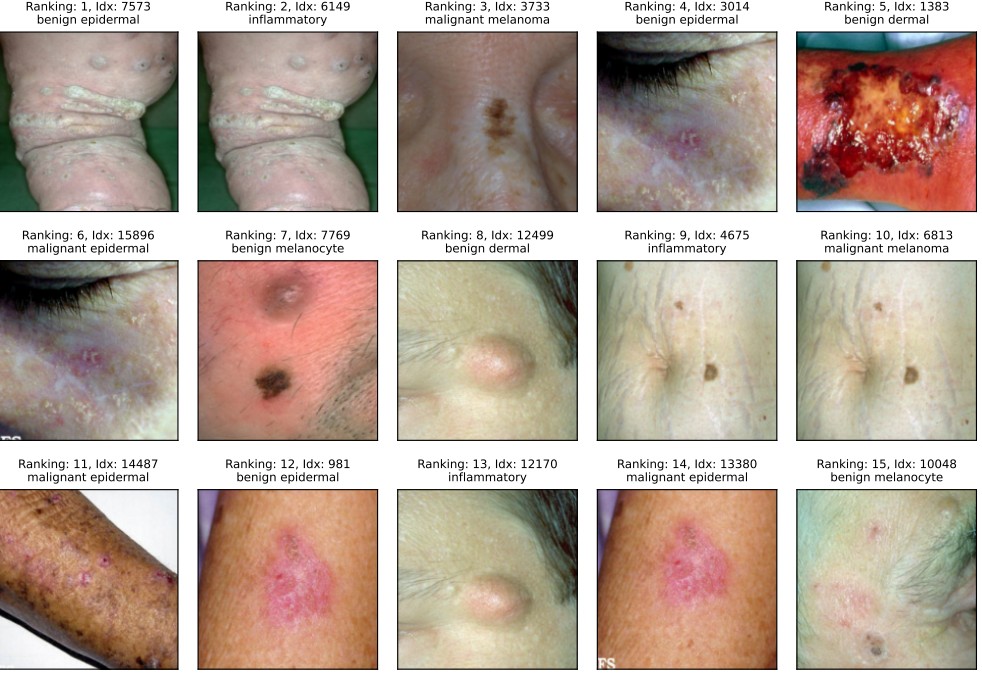

Figure 26: Ranking produced by SELFCLEAN for label errors in the Fitzpatrick17k, of which the top-15 are shown along with the respective rank, index, and original label.

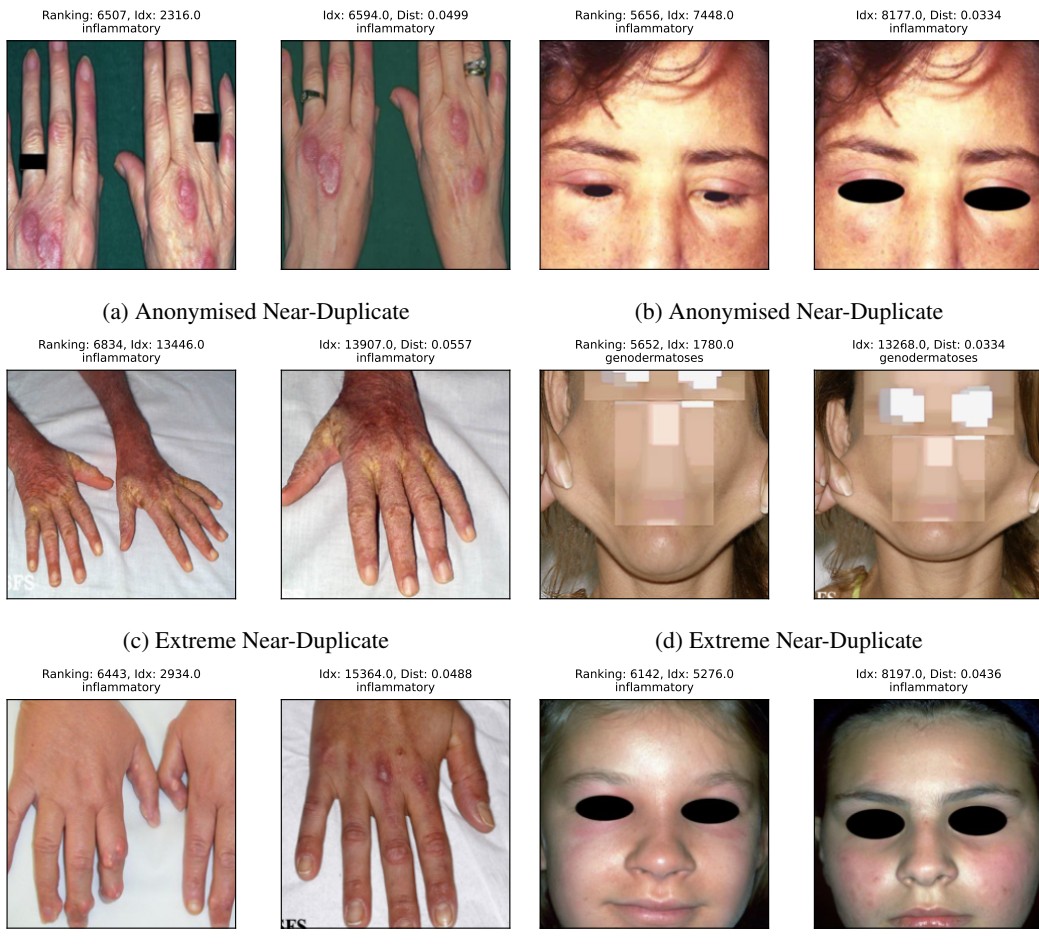

Figure 27: The figure shows multiple near-duplicates found in the Fitzpatrick17k dataset (Groh et al., 2021). Figures 27a and 27b show that SELFCLEAN can find near-duplicates where the images were anonymized. Further, figures 27c and 27d show extreme cases of near-duplicates where the method found crops of the same image. Finally, figures 27e and 27f show examples where the near-duplicate detection failed.

