# OpenReview forum: "SelfClean: A Self-Supervised Data Cleaning Strategy"
_ICLR.cc/2024/Conference — Submitted to ICLR 2024_

### Official Review · Reviewer_psYA · 2023-10-27

**Soundness:** 2 fair
**Presentation:** 4 excellent
**Contribution:** 3 good
**Rating:** 6
**Confidence:** 4

**Summary:**

This paper considers the problem of cleaning up large datasets used for training and evaluation. The paper considers three kinds of imperfections: (1) samples which are irrelevant, (2) samples which are near duplicates of other samples, and (3) samples with incorrect labels. The proposed approach is to use self-supervised learning methods to learn an embedding of the data and then compute scores for each of the imperfection categories above based on these embeddings. Then, samples are deemed to be flawed based on either a threshold-based criterion or human review. The method is evaluated using datasets with artificially injected flaws, as well as "in the wild" datasets (in which metadata and human review are used as ground truth).

I liked this paper because of the problem it considers and the ambition and thoroughness of its approach. However, there are some technical issues and some issues with framing. If tweaked to be a bit more thoughtful along a few axes, it could be a great contribution to ICLR.

I will flag the fact that I don't know the related work well enough to know whether all key baselines are included. On this, I will defer to more knowledgeable reviewers.

# References (used later)

@article{xiao2020should,
  title={What should not be contrastive in contrastive learning},
  author={Xiao, Tete and Wang, Xiaolong and Efros, Alexei A and Darrell, Trevor},
  journal={arXiv preprint arXiv:2008.05659},
  year={2020}
}

@inproceedings{cole2022does,
  title={When does contrastive visual representation learning work?},
  author={Cole, Elijah and Yang, Xuan and Wilber, Kimberly and Mac Aodha, Oisin and Belongie, Serge},
  booktitle={Proceedings of the IEEE/CVF Conference on Computer Vision and Pattern Recognition},
  pages={14755--14764},
  year={2022}
}

**Strengths:**

* The paper is generally well written and interesting to read.
* Very polished - I didn't spot one typo!
* The task under consideration - cleaning up datasets for training and evaluation - is a timely and important topic.
* The proposed methodology seems reasonable enough - see below for some caveats.
* The experiments are thorough and thoughtful.
* The paper includes a limitations section, which is always helpful for the reader.
* Ample supporting details and examples can be found in the appendix.

**Weaknesses:**

I have a few technical concerns:
* The paper is built on augmentation-driven SSL methods. The representations learned by these methods are driven by the augmentations they use, which generally need to be chosen in a domain-specific way [xiao2020should]. These seems like a liability for the proposed method, and should be addressed. In practice, how is someone supposed to pick?
* The SSL encoders which are the foundation of this paper are all trained with default or near-default parameters. Given that those parameters were tuned for ImageNet in the original papers, why shouldn't we worry that the encoders in this work are trained to varying levels of quality, confounding the results?

The paper's goals can seem a bit detached from real use cases, despite being framed in a very practical way. For instance:
* One of the three types of data fault considered in the paper is *near duplicates*. Given that the paper considers many medical use cases, this seems potentially problematic. What about adjacent B-scans in an OCT volume? Or adjacent tissue sections in an H&E block? Or longitudinal X-rays from a patient? All of these should be near-duplicates, but may be quite important.
* One of the three types of data fault considered in this paper is *irrelevance*. This seems like a loaded and insufficiently defined term. Isn't relevance in the eye of the beholder? When does it make sense to have a sample that is correctly labeled but "irrelevant" to the classification task at hand? Wouldn't it be obvious from the label that such a sample is irrelevant?
* Adding X-ray images to a dataset of images from a different domain seems like a contrived task of low difficulty. (One could probably tell an X-ray image from a photograph of a skin lesion using classical image processing methods.) It seems like the "real" use case would be more like adding X-ray images of one disease to a dataset consisting of X-ray images from a very different disease. More broadly, it seems like concept granularity is an important missing piece in this work - see e.g. the granularity-depending usefulness of SimCLR features described in [cole2022does].

## Minor comments
* This is not an accurate definition for SSL in general: "...self-supervised learning (SSL), which produces meaningful latent spaces by training features to be invariant to augmentations." Consider, say, BYOL or MAE.
* The text discussing the merging strategy for irrelevant samples is a bit hard to follow.
* It would be nice if e.g. the metrics in Table 1 could be translated into an estimate of how much human effort is saved. This would help us understand what percentage improvement is necessary to make a meaningful difference between two methods.
* The "Comparison with human annotators" section was a little hard to parse for me. Consider making the metrics and outcomes clearer. Also, "more often than random" is a pretty low bar - if something stronger can be said, it should be said.
* It might be mentioning in the main paper that all of the experiments are done on ViTs (and are therefore confounded with architecture).

**Questions:**

0. See "weaknesses" section.
1. Why should we not be concerned about the choice of augmentations underlying the self-supervised learning used in this paper?
2. Does the synthetic case of adding X-ray images to a dataset of images from a different domain model a realistic use case?
3. The paper notes that "the batch size cannot be large" for SimCLR - why is this?
4. Table 1 shows that sometimes SimCLR and DINO lead to very different performance and sometimes they do not. Is there any insight to be had into why this is?
5. Why shouldn't we worry about the proposed method removing difficult but important images or groups of images, especially in a medical context?

---

> ### Author Response · Authors · 2023-11-15
> **Response to Reviewer psYA**
>
> **Questions:**
>
> > Q1: Why should we not be concerned about the choice of augmentations underlying the self-supervised learning used in this paper?
> > W1: The paper is built on augmentation-driven SSL methods. The representations learned by these methods are driven by the augmentations they use, which generally need to be chosen in a domain-specific way [xiao2020should]. These seems like a liability for the proposed method, and should be addressed. In practice, how is someone supposed to pick?
>
> We appreciate this question which is valid and well formulated.
> Before answering it, there are two things to be observed.
> First, SSL methods that are not based on augmentations are a valid alternative for data cleaning, and their evaluation in the context of SelfClean is an interesting future direction.
> Second, in our analysis the same standard settings were effective on 10 image datasets of different sizes and semantics, which suggests the method is quite robust to the choice of augmentation.
> Unfortunately, we used different augmentations compared to the original papers as these provided too weak of a learning signal for small datasets.
>
> That said, the choice of augmentations is an important inductive bias as it establishes semantic equivalence, and in general requires domain knowledge.
> For instance, in a dataset of X-ray scans, a flipped image may be considered problematic for data quality or not.
> Therefore this step cannot, in our opinion, be automated - although luckily, in practice, there are very generic settings which lead to good results in a broad range of situations.
> In general, with more specific augmentations we expect our competitive results to improve even further.
>
> > Q2: Does the synthetic case of adding X-ray images to a dataset of images from a different domain model a realistic use case?
>
> Yes, unfortunately this is encountered often in the medical domain as some datasets are extracted from atlases.
> These atlases are however usually used for educational purposes and thus not only feature a single modality such as e.g. dermoscopy, but also complementary ones such as histopathology or X-rays of the patient.
> Our synthetic augmentations were heavily influenced by examples of real noise we encountered in the medical domain, and by domain knowledge of medical practitioners.
> Please consult our answer to W3.1 of reviewer zncP for more information.
>
> > Q3: The paper notes that "the batch size cannot be large" for SimCLR - why is this?
>
> Because in some of our experiments, the whole dataset has fewer examples than the usual batch size chosen for SimCLR.
>
> > Q4: Table 1 shows that sometimes SimCLR and DINO lead to very different performance and sometimes they do not. Is there any insight to be had into why this is?
>
> The dataset size seems to be crucial for SimCLR to obtain good performance, as we note in passing towards the end of section 5.1.
>
> > Q5: Why shouldn't we worry about the proposed method removing difficult but important images or groups of images, especially in a medical context?
>
> SelfClean is meant as a tool to identify potential issues.
> The action to be taken remains in the responsibility of the user, which we argue is in a better position to decide with the additional knowledge provided by our method.
> We added a comment in the Recommended use paragraph of section 6 to highlight this, and included a preliminary study on biases for minority groups in Appendix G.3.
>
> **Weaknesses:**
>
> > W2: The SSL encoders which are the foundation of this paper are all trained with default or near-default parameters. Given that those parameters were tuned for ImageNet in the original papers, why shouldn't we worry that the encoders in this work are trained to varying levels of quality, confounding the results?
>
> That is a very fair observation, which we purposefully neglected as our goal was not to show the optimal performance that can be achieved with significant time spent on hyperparameter tuning.
> However, we expect the quality of the encoders to positively correlate with the performance of SelfClean, thereby suggesting that even better results may be achieved by tuning hyper-parameters in a dataset-specific way.
> This is now noted in the Limitations paragraph of section 6.

---

> > ### Author Response · Authors · 2023-11-15
> > **Response to Reviewer psYA**
> >
> > > W3: One of the three types of data fault considered in the paper is near duplicates. Given that the paper considers many medical use cases, this seems potentially problematic. What about adjacent B-scans in an OCT volume? Or adjacent tissue sections in an H&E block? Or longitudinal X-rays from a patient? All of these should be near-duplicates, but may be quite important.
> >
> > We fully agree with the observation and the examples of near duplicates that should not be removed.
> > In our paper we only give the definition of near duplicates without formulating a recommendation for their treatment, because no general suggestion can be made as correctly pointed out.
> > However, SelfClean enables practitioners to find relations among data samples, including longitudinal assessments and spatially close images, even if such information was not included in the metadata.
> > This can be extremely useful for instance when merging data from different clinics in the same geographical region or when performing data splitting without leakage.
> > We also observed for example that SelfClean may be able to identify dermatologic pictures of the same limbs from the front and from the back.
> > When appropriately used, in our opinion, this information greatly adds to the value of the method, as we now explicitly mention in the Recommended use paragraph of section 6.
> >
> > > W4: One of the three types of data fault considered in this paper is irrelevance. This seems like a loaded and insufficiently defined term. Isn't relevance in the eye of the beholder? When does it make sense to have a sample that is correctly labeled but "irrelevant" to the classification task at hand? Wouldn't it be obvious from the label that such a sample is irrelevant?
> >
> > This observation is well justified as our definition of irrelevant samples is dense, and we could not unpack it because of space constraints.
> > The answer is that there are indeed examples that are clearly correctly labeled but irrelevant.
> > We encountered this for instance with histopathology images that were crawled together with the corresponding skin lesion: they indicate the correct disease, but they are just not suitable to be classified in the same way as, for example, dermatoscopy pictures.
> > Likewise, in FOOD-101N we found abundant images of menus labeled with the correct category of food, which are arguably not amenable to the same classification as pictures of the food itself.
> >
> > > W5: Adding X-ray images to a dataset of images from a different domain seems like a contrived task of low difficulty. (One could probably tell an X-ray image from a photograph of a skin lesion using classical image processing methods.) It seems like the "real" use case would be more like adding X-ray images of one disease to a dataset consisting of X-ray images from a very different disease. More broadly, it seems like concept granularity is an important missing piece in this work - see e.g. the granularity-depending usefulness of SimCLR features described in [cole2022does].
> >
> > The comment on granularity is very valuable, and it is included now in the Limitations paragraph in section 6.
> > Throughout our experiments, we identified many datasets that have simple types of irrelevant samples, for instance, the histopathology or menu images as mentioned above.
> > We therefore argue that identifying even this type of simple issue is very valuable, especially when they are rare and one does not know what to look for.
> >
> > **Minor Comments:**
> >
> > > C1: This is not an accurate definition for SSL in general: "...self-supervised learning (SSL), which produces meaningful latent spaces by training features to be invariant to augmentations." Consider, say, BYOL or MAE.
> >
> > Thank you for pointing this out.
> > We have revised this sentence.
> >
> > > C2: The text discussing the merging strategy for irrelevant samples is a bit hard to follow.
> >
> > Thank you for this helpful comment.
> > We will have another pass at this section and revise it.
> >
> > > C3: It would be nice if e.g. the metrics in Table 1 could be translated into an estimate of how much human effort is saved. This would help us understand what percentage improvement is necessary to make a meaningful difference between two methods.
> >
> > It is indeed possible to measure precisely how much human effort is saved.
> > However, to evaluate this in a way which only depends on the ranking requires a new metric.
> > More precisely, we would suggest the fraction of time spent confirming negative samples for a given recall, averaged over recall.
> > This is exactly zero for a perfect model which ranks all problems before valid samples, exactly one for the worst model which ranks all problems, and equal to the fraction of valid samples for a random model.
> > The results are still pending and will be included in the next days.

---

> > > ### Author Response · Authors · 2023-11-15
> > > **Response to Reviewer psYA**
> > >
> > > > C4: The "Comparison with human annotators" section was a little hard to parse for me. Consider making the metrics and outcomes clearer. Also, "more often than random" is a pretty low bar - if something stronger can be said, it should be said.
> > >
> > > Thank you for pointing it out.
> > > We have reformulated the sentence.
> > >
> > > > C5: It might be mentioning in the main paper that all of the experiments are done on ViTs (and are therefore confounded with architecture).
> > >
> > > Thanks for pointing this out.
> > > We have added a statement about the choice of encoder architecture in Section 3.

---

> ### Comment · Reviewer_psYA · 2023-11-22
> **Response to authors**
>
> I appreciate your detailed response to my comments!
>
> At present, my rating is unchanged (weak accept). That is, I still think the paper is likely to be of interest at ICLR. However, I also still think it's not a clear accept in its current form because of:
> * ongoing concerns about confounding in the experiments, due to the use of default hyperparameters in training the encoders (W2);
> * concerns about how practically useful the method is, due to somewhat contrived tasks (W5); and
> * a lack of empirical insight into the relationship between encoder quality / invariances and SelfClean performance (Q1/W1).

---

> > ### Author Response · Authors · 2023-11-22
> > **Response to Reviewer psYA**
> >
> > Dear reviewer psYA,
> >
> > As previously mentioned in response to your minor comment C3, we estimated how much human effort is saved thanks to ranking samples.
> > You can find the details in Appendix G.7 "Annotation Effort Saved", with results reported numerically in a table and graphically in a set of plots.
> > Using synthetic mixed contamination of all considered types at 10\% level, on the DDI dataset and for the specific random seed used,
> > SelfClean reduces the human confirmation effort by a **factor of 5 for irrelevant samples** compared to the most competitive method IN + HBOS,
> > reduces effort by a **factor of 100 for near duplicates** compared to the most competitive approach SSIM,
> > and increases effort by a factor of 2 for label errors compared to FastDup (while still marginally better than IN + Confident Learning).
> > We hope that this is helpful to translate our metrics in expected annotation effort reduction.
> >
> > We would also like to thank you for commenting on our reply.
> > We are of the opinion that tuning hyperparameters differently for each dataset would confound results more rather than less, as each optimization would have a different degree of success.
> > We also hold the view that the practical utility of SelfClean in realistic settings is clearly demonstrated by the comparison with metadata in Section 5, Appendix G.2 (now extended) and by the samples of data quality issues found in the datasets of Appendix M.

---

### Official Review · Reviewer_N64F · 2023-11-01

**Soundness:** 2 fair
**Presentation:** 2 fair
**Contribution:** 1 poor
**Rating:** 1
**Confidence:** 5

**Summary:**

The paper proposes an data label noise detection and removal technique based on self-supervised trained embeddings and three basic heuristics to identify potential outliers, near duplicates with conflicting labels and other labeling errors using agglomerative clustering, pairwise distance comparisons (thresholding) and "intra/extra" class distance ratios. The method is evaluated on 10 different datasets including popular computer vision benchmarks (such as ImageNet, Food101, CelebA) and seven other medical image datasets, mainly through synthetic experiments. The results are compared to some other known methods such as pHash, SSIM, FastDup.

**Strengths:**

- Investigates label noise detection and removal on ten different images datasets including from both computer vision and medical imaging domains.

**Weaknesses:**

- Lack of novelty. The described methods of clustering, distance thresholding and intra/extra class distance ratios in the embedding space trained on the target dataset have been widely known and applied in the space. Combining these approaches is not sufficient for justifying novelty.
- Lack of proper evaluation setup. The experiments are primarily performed using the synthetic label noise which is also devised by the paper.
- Lack of proper baselines.
- The paper is hard to follow.

**Questions:**

- Dataset size and data distribution would largely impact the hyper-parameters (alpha, q etc) used for identifying near duplicates, outliers etc. It is not clear how well the proposed heuristics would generalize for other datasets or other applied settings.
- Page 9: "Recommended use" section. The paper's proposal is to use SelfClean to identify the errors and fix them using human in the loop. However, the process would still introduce sampling bias originating from SelfClean. This should be noted.
- What was the rationale for choosing the current baselines? Below is the suggested baseline which uses pre-trained embeddings and vector similarity to identify label noise using Markov Random Fields (most similar to the proposed approach)
Sharma et al, "NoiseRank: Unsupervised Label Noise Reduction with Dependence Models", ECCV, 2020
-  The paper is way too dense and hard to follow. Proposed novelties also needs to be clearly noted in the abstract.

---

> ### Author Response · Authors · 2023-11-15
> **Response to Reviewer N64F**
>
> **Summary and Strengths:**
>
> > S1.1: The paper proposes a [...] technique [...] to identify potential outliers, near duplicates with conflicting labels and other labeling errors [...].
>
> This remark is not completely accurate.
> In fact, we identify near duplicates irrespective of whether they have conflicting labels and argue that this is useful in general.
> The case of near duplicates with conflicting labels is however particularly relevant as it essentially results in data poisoning.
>
> > S1.2: Investigates label noise detection and removal [...].
>
> While formally correct, this statement is incomplete.
> Indeed, we do not simply consider label noise but also two other types of data quality issues that are not related to labels.
>
> **Questions:**
>
> > Q1: Dataset size and data distribution would largely impact the hyper-parameters (alpha, q etc) used for identifying near duplicates, outliers etc. It is not clear how well the proposed heuristics would generalize for other datasets or other applied settings.
>
> This is a very valid point, which however we address by considering 10 datasets spanning different sizes (from 200 to 300,000 samples) and domains (from dermatology to X-ray scans to food and celebrity faces).
> We demonstrate that the same heuristics yield good rankings for candidate issues across all of these datasets.
> Furthermore, we are able to fix the hyper-parameters mentioned in the question, which are only used for fully automatic cleaning, to the same values throughout.
> Additionally, we report a detailed study on the influence of these hyper-parameters in Appendix H, sections H.3 and H.4.
> Thus, there is evidence that the proposed heuristics generalize to several different domains and settings.
>
> > Q2: Page 9: "Recommended use" section. The paper's proposal is to use SelfClean to identify the errors and fix them using human in the loop. However, the process would still introduce sampling bias originating from SelfClean. This should be noted.
>
> We fully agree with the observation that bias is not entirely eliminated, even in our recommended use.
> We touch on this theme in the mentioned "Recommended use" section ("The conflict between resolving data quality issues and the veto against the examination of evaluation data, mentioned in the introduction, has no easy resolution. We suggest the following compromise [...] as an improvement to the current situation.").
> To highlight the point even more, the paragraph now includes a dedicated remark.
>
> > Q3.1: What was the rationale for choosing the current baselines?
>
> We included methods from other frameworks that address multiple types of data quality issues, i.e. FastDup.
> Additionally, for each of the three noise types we selected one or more recent competitive baselines with open-source implementations.
> Furthermore, as Confident Learning has compared their methodology against seven other techniques for label error detection and showed their superiority, there is reason to believe that by outperforming Confident Learning in our experiments our approach is also competitive with those.
> It is almost impossible to exhaustively evaluate against other approaches, so despite all of our efforts, we may still have missed some that are particularly relevant for SelfClean.
>
> > Q3.2: Below is the suggested baseline which uses pre-trained embeddings and vector similarity to identify label noise using Markov Random Fields (most similar to the proposed approach) Sharma et al, "NoiseRank: Unsupervised Label Noise Reduction with Dependence Models", ECCV, 2020
>
> We are grateful for this suggestion and have included NoiseRank in our comparison.
> In Appendix G.1, you can find the results of the method in our synthetic scenario, and in Table 6 of Appendix G.2 the results of the comparison with metadata.
>
> > Q4.1: The paper is way too dense and hard to follow.
>
> We are sorry about this finding and keep this comment in mind during revision.
> As the other reviewers (97xm, zncP, psYA) unanimously agree that the paper is well written, we kindly ask you to point out specific parts that would in your opinion profit from a better formulation.
>
> > Q4.2: Proposed novelties also needs to be clearly noted in the abstract.
>
> Thank you very much for raising this point.
> We reformulated the abstract to make it clear that features are learned in the context of the dataset only, that the combination of simple heuristics we identify is surprisingly effective, and that we perform a thorough evaluation which shows competitive results and practical utility.

---

> > ### Author Response · Authors · 2023-11-15
> > **Response to Reviewer N64F**
> >
> > **Weaknesses:**
> >
> > > W1: Lack of novelty. The described methods of clustering, distance thresholding and intra/extra class distance ratios in the embedding space trained on the target dataset have been widely known and applied in the space. Combining these approaches is not sufficient for justifying novelty.
> >
> > We are not aware of published work that leverages self-supervised features learned on the target dataset to find multiple data quality issues and presents a detailed evaluation of the data cleaning procedure.
> > If specific, prior relevant references are provided, we will reconsider our work's novelty.
> >
> > > W2: Lack of proper evaluation setup. The experiments are primarily performed using the synthetic label noise which is also devised by the paper.
> >
> > We thank the reviewer for highlighting that the comparison with other methods is focused on synthetic datasets.
> > The reason to take this as a starting point is that evaluating data cleaning is very difficult in real scenarios.
> > The experiments in Appendix I, the comparison with metadata in Appendix G.2, and the many qualitative examples shown in Appendix M are evidence that SelfClean also works on natural contamination.
> > However, to address this concern, we have now extended the comparison with metadata of Appendix G.2 to include competing approaches in Table 6, albeit resorting to subsampling in the case of near duplicates because of time constraints.
> > Furthermore, in a follow-up work that is now accepted for publication, we analyzed four additional small to medium-size datasets with increasing levels of curation finding AUROCs of 86.2%, 99.9%, 100%, and 97.7% for irrelevant samples, 99.9%, 100%, 99.9%, and 100% for near duplicates, and 82.7%, 93.5%, 99.2%, and 98.8% for label errors.
> > For full disclosure, these numbers may be positively biased due to the details of the verification procedure, but still constitute strong evidence that the method performs well.
> > Finally, we once more highlight that our work is not limited to "label noise" only.

---

> > > ### Comment · Reviewer_N64F · 2023-11-22
> > >
> > > Sincere thanks to the authors for the discussions. The use of SSL-based models for feature extraction alone is not sufficient enough to claim novelty and meet the paper acceptance bar. The absence of such baselines does not necessarily make the contributions significant enough to meet the acceptance bar.  At least one more reviewer stated a similar concern (citing below), I am not in favor of accepting the paper at this time. Not changing the initial rating.
> > >
> > > Reviewer (zncP): "My primary reservation concerns the novelty of the cleaning approach. The main contribution is the use of SSL-based models for feature extraction. Beyond this, I struggle to identify other novel aspects."

---

> > > > ### Author Response · Authors · 2023-11-23
> > > > **Response to Reviewer N64F**
> > > >
> > > > We are thankful for the reviewer's reply and for joining the discussion.
> > > >
> > > > To the best of our understanding, the reviewer is arguing that the absence of peer-reviewed results on SSL for data cleaning is not sufficient to make the work novel.
> > > > However, we challenge the reviewer, who claimed that the method is well-known in the community, to produce even unreviewed work exploiting **dataset-specific SSL for cleaning** or achieving performance similar to ours across data quality issues.
> > > > Furthermore, we clearly demonstrate in figure 2 that SSL features alone are not sufficient to reap all the benefits of SelfClean, but they need to be **combined with the right detection methods**.
> > > > In our synthetic and natural contamination experiment, **SelfClean outperforms all 9 competitors**.
> > > >
> > > > The average improvement in AP compared to the best competitor is 84\%, 19\%, and 12\% on synthetic contamination, while on natural contamination we gain a factor of 2 for known label errors and on average a factor 15 for near-duplicates.
> > > > These factors are increased by a large margin when comparing label error detection with NoiseRank, the competitor method indicated by the reviewer.
> > > > We therefore argue that SelfClean is a **considerable improvement compared to the state of the art**.
> > > >
> > > > We believe these elements need to be considered when assessing **novelty**.

---

### Official Review · Reviewer_zncP · 2023-11-01

**Soundness:** 2 fair
**Presentation:** 3 good
**Contribution:** 2 fair
**Rating:** 6
**Confidence:** 4

**Summary:**

The authors of this work introduce a new dataset cleaning strategy. They utilize in-domain self-supervised learning to address challenges associated with human biases/errors, task biases, and issues present in other methods that use supervised learning e.g. semantic collapse. Their primary focus is on eliminating irrelevant examples, near duplicates, and correcting labeling errors. They benchmark their approach on an array of datasets and tasks, showing that their method outperforms previous works, by a large margin in some cases while they also provide valuable insights.

**Strengths:**

The authors of this paper set out to tackle the prominent, and always relevant, problem of dataset cleaning. I find myself in agreement with the authors' narrative and intuition about the use of self-supervised features, which I believe address several issues encountered by past methods.  The authors benchmark their approach primarily on medical datasets but they also provide valuable insights on the application of their method to mainstream natural datasets. In general, the paper is well written and easy to go through.

**Weaknesses:**

My primary reservation concerns the novelty of the cleaning approach. The main contribution is the use of SSL-based models for feature extraction. Beyond this, I struggle to identify other novel aspects.

While the experiments are detailed, they appear somewhat limited in scope. The authors mainly concentrate on dermatology datasets and their experiments to natural images or other domains seems rather limited. Based on the novelty of the paper I would expect a stronger experimental section.

The first and second contamination strategies, while intuitive and useful, don't seem to address real-world noise. In the context of medical images, irrelevant examples often include images that are out-of-focus, images with doctor’s annotations, or images with parts from an apparatus. They do not contain random ImageNet examples, PowerPoint slides or data from a different medical modality. Similarly for the duplicates, a simple rotation, resizing etc. is a rather a simplification of real-world duplicates.

The impact of dataset cleaning appears to be less significant than anticipated. While there are instances where cleaning the evaluation set boosts performance, there are equally instances where has a negative impact on the results.  Most importantly, when the training set gets cleaned, the performance decreases in 3 out of 5 cases. In the remaining 2 out of 5 cases, the gains in performance are rather small. This is important especially considering real-world applications – dataset cleaning is not only for benchmarking. Only the training set is accessible. The test set remains unknown and probably contains corrupted data.

When k-NN evaluation is used, cleaning the training set seems to help consistently (although marginally in most cases). However, this is rather expected given the nature of k-NN evaluation.  However, in reality we care about the performance of linear classifiers and not k-NN. Most importantly, a fine-tuning step, which is a standard expectation in real-world scenarios, seems to be missing from this analysis.

**Questions:**

A significant challenge in medical diagnosis is that of intra-observer variability, which complicate thing further when comes to labeling errors. How does one address this complexity when it is nearly impossible to define a ground truth?

---

> ### Author Response · Authors · 2023-11-15
> **Response to Reviewer zncP**
>
> **Questions:**
>
> > Q1: A significant challenge in medical diagnosis is that of intra-observer variability, which complicate thing further when comes to labeling errors. How does one address this complexity when it is nearly impossible to define a ground truth?
>
> We agree that intra-observer variability, i.e. the difference in repeated measurements by the same observer, is an important issue especially in the medical domain.
> In cases where the opinion of an expert shows significant variations over time, or even if multiple experts cannot reach a consensus, it is desirable to get another source of truth (e.g. other diagnostics, follow ups).
> If the truth is not positively defined for a sample, it does not meet our definition of a label error, and should rather be considered an ambiguous sample.
> Integrating a notion of ambiguity in the SelfClean framework is a very interesting future direction of research.
>
> **Weaknesses:**
>
> > W1: My primary reservation concerns the novelty of the cleaning approach. The main contribution is the use of SSL-based models for feature extraction. Beyond this, I struggle to identify other novel aspects.
>
> The main novelty of SelfClean is that we describe precisely how to combine self-supervised learning with simple indicators to detect data quality issue competitively across different types of noise, while many alternatives do not work as effectively.
> Notably, we suggest dataset-specific SSL to learn concepts of data quality issues in the context of the collection at hand and demonstrate that this works for three types of noise.
> We also demonstrate that not all SSL strategies work equally well in low-data regimes and suggest DINO as a general baseline.
> As to the simple strategies to find issues, they share the same extendable principle and perform consistently well.
> For instance, using hierarchical clustering allows for joint treatment of isolated and grouped irrelevant samples, and is for this reason much more efficient than e.g. the local outlier factor.
> Or again, the average intra-/extra-class distance ratio is not nearly as good as the more local single-neighbor ratio we suggest.
>
> > W2: While the experiments are detailed, they appear somewhat limited in scope. The authors mainly concentrate on dermatology datasets and their experiments to natural images or other domains seems rather limited. Based on the novelty of the paper I would expect a stronger experimental section.
>
> In the paper we representatively include ImageNet, Food-101N, and CelebA for natural images, plus CheXpert and PatchCamelyon for other medical domains.
> The main reason why our evaluation mostly focuses on small specific datasets is the challenge in finding standard image collections that are truly clean.
> Nevertheless, to address this concern, we are running the synthetic mixed-contamination experiment both for subsets of both ImageNet and CheXpert.
> Moreover, the comparison with metadata (including CelebA, ImageNet-1k, and Food-101N) is now performed also for other baselines, which better highlights the effectiveness of SelfClean.
> We agree that it would be nice to have more, but we believe the empirical evidence to be sufficient for publication of the method.

---

> > ### Author Response · Authors · 2023-11-15
> > **Response to Reviewer zncP**
> >
> > > W3.1: The first and second contamination strategies, while intuitive and useful, don't seem to address real-world noise. In the context of medical images, irrelevant examples often include images that are out-of-focus, images with doctor’s annotations, or images with parts from an apparatus. They do not contain random ImageNet examples, PowerPoint slides or data from a different medical modality.
> >
> > There are indeed many types of irrelevant samples, and some might be more challenging than the ones we analyze.
> > Our selection is based on our own experience with public medical datasets, where crawling often leads to including completely extraneous images from websites (hence ImageNet samples) or pictures related to clinical understanding but not to the task (e.g. histopathology of a dermatologic lesion, or X-ray scan of the corresponding limb).
> > Such irrelevant types are found for instance in the Fitzpatrick17k dataset [1], as can be seen from Figure 23 in Appendix M.4.
> > Furthermore, clinicians often maintain their collection of rare medical cases in PowerPoint slides [2], hence the addition of slides that can occur due to image extraction from them.
> > Moreover, features like clinician annotations or e.g. rulers may or may not be considered irrelevant samples according to their prevalence in the dataset at hand and are arguably parts of valid samples in many cases we consider.
> > Finally, our qualitative results in the appendix demonstrate that SelfClean can actually find rather complex cases, such as arguably invalid CheXpert scans (Figure 17 of Appendix M.2) or color alterations in DDI.
> >
> > However we are currently running experiments with an additional synthetic strategy where irrelevant samples consist of existing samples augmented with a strong amount of blur, making the object on these images invisible.
> > Results are still pending and will be added in the coming days.
> >
> > > W3.2: Similarly for the duplicates, a simple rotation, resizing etc. is a rather a simplification of real-world duplicates.
> >
> > This is a fair observation.
> > Our suggestion is a starting point, for lack of a better solution, which finds application in many crawling scenarios where the same image might appear with different post-processing.
> > However, we have also performed extensive investigation in real-world scenarios, such as comparison with metadata and human validation, where our method found some very non-trivial near duplicates, such as images of the same patient potentially from very different angles (see e.g. Figure 25 in Appendix M.4 and also Section G.2).
> >
> > > W4.1: The impact of dataset cleaning appears to be less significant than anticipated.
> >
> > Albeit small in absolute terms, the effect is significant in 5/10 cases with a linear classifier and in 9/10 cases for kNN.
> > Furthermore, for the two uncurated datasets Fitzpatrick17 and Food-101N, it is significant in 4/4 for a linear classifier and 3/4 for kNN.
> > A reason why score variations are small is that we purposefully set automatic cleaning to be very conservative.
> > For instance, in HAM10000 fully automatic cleaning with default parameters finds only one near duplicate pair,
> > while a review of the SelfClean ranking by multiple domain experts identified 16.
> >
> > > W4.2: While there are instances where cleaning the evaluation set boosts performance, there are equally instances where has a negative impact on the results.
> >
> > This is expected when cleaning evaluation sets, where removing unwanted samples makes evaluation more correct but might pull the scores in any direction according to whether those examples were easy or hard.
> >
> > > W4.3: Most importantly, when the training set gets cleaned, the performance decreases in 3 out of 5 cases. In the remaining 2 out of 5 cases, the gains in performance are rather small.
> >
> > This is true for linear evaluation, and the paired permutation test does not attribute significance to any of these three cases, while it does to the other two.
> > In addition, we argue that even when cleaning the training set results in lower scores, investigation of why optimizing for corrupted data improves performance is due.
> >
> > > W4.4: This is important especially considering real-world applications – dataset cleaning is not only for benchmarking. Only the training set is accessible. The test set remains unknown and probably contains corrupted data.
> >
> > We agree that in real-world applications true generalization performance is given by truly unseen data in production.
> > Our experiment with benchmarks is only meant to assess if there is a significant different in that case.
> > However our methodology is not limited to cleaning data collections prior to training, but also has the possibility to be run at inference time to filter unwanted samples, as observed at the end of section 7.

---

> > > ### Author Response · Authors · 2023-11-15
> > > **Response to Reviewer zncP**
> > >
> > > > W5.1: When k-NN evaluation is used, cleaning the training set seems to help consistently (although marginally in most cases). However, this is rather expected given the nature of k-NN evaluation. However, in reality we care about the performance of linear classifiers and not k-NN.
> > >
> > > We agree that, since kNN is influenced by individual data points and SelfClean ranks samples based on their nearest neighbor, a higher impact is expected on kNN.
> > > However we would also like to point out that kNN and linear evaluation are currently the primary evaluation criterion to measure the performance of latent representations [3,4,5].
> > > Moreover, generic models can learn more complex decision boundaries than mere hyper-planes.
> > > Therefore, by showing the performance difference in both settings we argue that it gives a somewhat more complete picture.
> > >
> > > > W5.2: Most importantly, a fine-tuning step, which is a standard expectation in real-world scenarios, seems to be missing from this analysis.
> > >
> > > Our main rationale for focusing on kNN and linear evaluation was to investigate the general impact on possibly weaker evaluations and not to provide the most significant differences achievable.
> > > However, we find this comment very helpful and are currently extending the investigation to feature the fine-tuned performance.
> > > We will be back with results as soon as we have them.
> > >
> > > **References:**
> > >
> > > [1] Groh, M., et al. (2021). Evaluating Deep Neural Networks Trained on Clinical Images in Dermatology with the Fitzpatrick 17k Dataset.
> > > [2] Tschandl, P., et al. (2018). The HAM10000 dataset, a large collection of multi-source dermatoscopic images of common pigmented skin lesions.
> > > [3] Caron, M., et al. (2021). Emerging Properties in Self-Supervised Vision Transformers.
> > > [4] Zhou, J., et al. (2021). IBOT: Image BERT Pre-Training with Online Tokenizer.
> > > [5] Oquab, M., et al. (2023). DINOv2: Learning Robust Visual Features without Supervision.

---

> > > > ### Author Response · Authors · 2023-11-22
> > > > **Response to Reviewer zncP**
> > > >
> > > > Dear reviewer zncP,
> > > >
> > > > To demonstrate the effectiveness of SelfClean compared to other approaches more effectively outside of dermatology (W2), we identified the labeled subset of STL-10 as a relative clean dataset of **natural images for an additional synthetic noise experiment**.
> > > > Indeed, we could unfortunately not rely on the ImageNet-1k validation set, which is estimated to have around 5\% of irrelevant samples and 20\% problematic labels using different combinations of automatic methods and human annotation [1,2,3,4].
> > > > Results are reported in Appendix G.8, where we find that **SelfClean outperforms competitive approaches by a large margin** in detecting synthetic irrelevant samples, generated with strong blurring to simulate bad focus, and synthetic near duplicates, obtained through augmentations.
> > > > We observe however that FastDup is considerably better than SelfClean in detecting label errors simulated with random label flips, but very revealing the intra-/extra-class distance ratio based on ImageNet features also performs much better than DINO self-supervised features.
> > > > Models which were trained on ImageNet have indeed already seen STL-10 images *with their labels*, and have therefore an unfair advantage on label error detection in this case.
> > > > It is reasonable to think that this was the case for the model used by FastDup.
> > > >
> > > > We also reconsidered and reshaped our experiment to demonstrate that cleaning also has a **significant impact on models which are not closely related to SelfClean**'s detection criteria, as you pointed out to be the case for kNN classifier (W5.2).
> > > > Indeed, we realized that standard evaluation with fine-tuned models is not meaningful in our setting, due to our assessment of uncertainties by repeating splits, cleaning, and fine-tuning many times.
> > > > The mechanisms triggered are that (A) the same hyperparameters do not guarantee proper convergence of all fine-tuning runs and (B) the computational load is too high to gather a sufficiently large statistical sample.
> > > > For these reasons, we decided to switch to a type of model that is very stable, fast, and not based on distance: random forests.
> > > > The results are reported in the table below.
> > > > It is clear that, also for this model class, cleaning the evaluation set impacts model performance estimates significantly, and cleaning the training set has a significant positive effect in one case out of five.
> > > > This indicates that **in general cleaning has a sizeable effect**, and the fact that effects are different for different models makes it even more important to obtain correct results.
> > > >
> > > > | Dataset |  |  | Random Forest Classifier |  |  |
> > > > | :---: | :---: | :---: | :---: | :---: | :---: |
> > > > |  |  | Scores (%) |  | Differences (%) |  |
> > > > |  | Cont + Cont | Cont + Clean | Clean + Clean | Clean Eval | Clean Train |
> > > > | DDI | $56.1_{-8.0}^{+6.9}$ | $53.8_{-8.1}^{+7.4}$ | $53.8_{-7.8}^{+5.7}$ | $-2.2_{-6.3}^{+7.0 \ ^{***}}$ | $-0.5_{-6.3}^{+7.8}$ |
> > > > | HAM10000 | $48.8_{-3.4}^{+2.6}$ | $48.0_{-2.3}^{+3.1}$ | $48.1_{-2.2}^{+3.2}$ | $-0.6_{-2.3}^{+2.4} \ ^{***}$ | $-0.1_{-2.7}^{+2.7}$ |
> > > > | Fitzpatrick17k | $50.2_{-1.5}^{+1.7}$ | $43.9_{-1.9}^{+1.7}$ | $44.5_{-1.6}^{+1.6}$ | $-6.4_{-2.0}^{+1.5} \ ^{***}$ | $+0.6_{-1.4}^{+1.0}\ ^{***}$ |
> > > > | Food-101N | $38.1_{-0.5}^{+0.6}$ | $38.0_{-0.6}^{+0.5}$ | $37.8_{-0.5}^{+0.5}$ | $-0.1_{-0.7}^{+0.4} \ ^{***}$ | $-0.1_{-0.5}^{+0.4}$ |
> > > > | ImageNet-1k | $23.1_{-0.5}^{+0.8}$ | $23.1_{-0.6}^{+0.4}$ | $23.2_{-0.6}^{+0.5}$ | $-0.0_{-0.5}^{+0.6}$ | $+0.1_{-0.6}^{+0.6}$ |
> > > >
> > > > **References:**
> > > > [1] Northcutt, Curtis G., Anish Athalye, and Jonas Mueller. "Pervasive label errors in test sets destabilize machine learning benchmarks." (2021).
> > > > [2] Kertész, Csaba. "Automated cleanup of the imagenet dataset by model consensus, explainability and confident learning." (2021).
> > > > [3] Beyer, Lucas, et al. "Are we done with imagenet?." (2020).
> > > > [4] Shankar, Vaishaal, et al. "Evaluating machine accuracy on imagenet." (2020).

---

### Official Review · Reviewer_97xm · 2023-11-01

**Soundness:** 3 good
**Presentation:** 3 good
**Contribution:** 3 good
**Rating:** 6
**Confidence:** 4

**Summary:**

The authors present a method to curate datasets by finding near duplicates, bad labeled and out of distribution images.
To do so, they first train a self-supervised DINO model on the dataset to filter and then use distance and clustering based methods for each kind of anomaly detection.
They compare their method against multiple baseline both on synthetic and real anomaly detection benchmarks.

**Strengths:**

- Clear explanation of the method
- Simple method that use the properties of SSL models to capture all factor of variations, which means that they don't require labels
- Lot of ablation studies
- The method seems robust to hyperparameter changes
- Good evaluation setup of a ill posed problem

**Weaknesses:**

All the methods are based on the notion that distance between images represents meaningful factors of variation. The closer the images are in the latent space, the more similar they should be, both in term of appearance but also semantically.
However, while SSL probably captures most factor of variations (and usually more than with supervised learning), it is still not possible to control the hierarchy in term of distance in the latent space between the factors. Meaning that if an image is a sketch of a dog, it is hard to tell if it will match more a real picture of a dog or a sketch of a cat using self-supervised features.
This means that images from under represented groups or from rare classes will be removed or wrongly flagged as anomaly which is a big issue in medical imaging.

**Questions:**

1) Could you ablate the weakness I'm talking about, for exemple by looking at the proportion of images removed from the clean version of the dataset wrt. the frequency of their labels, or grouped by multiple metadata such as demographics, device etc. (CheXpert demographics: https://github.com/biomedia-mira/chexploration https://stanfordaimi.azurewebsites.net/datasets/192ada7c-4d43-466e-b8bb-b81992bb80cf)

Optional:
2) We can see on Table 2 that the gains are higher with kNN than with linear evaluation with a clean train set. This seems to be because kNN eval is directly based on distance, such as your filtering method. Given that zero shot evaluation of CLIP is based on the distance between the text and image encoder, it is possible that your method can enhance zero-shot evals of CLIP model.
Could you test it in a zero-shot setup such as CLIP image/text cosine similarity matching ?

---

> ### Author Response · Authors · 2023-11-15
> **Response to Reviewer 97xm**
>
> **Questions:**
>
> > Q1: Could you ablate the weakness I'm talking about, for exemple by looking at the proportion of images removed from the clean version of the dataset wrt. the frequency of their labels, or grouped by multiple metadata such as demographics, device etc.
>
> Thank you for suggesting this experiment which you can now find in Appendix G.3.
> Before delving into the details, it is important to note that the danger of systematically removing underrepresented groups is considerably higher in the fully automatic mode.
> Fully automatic mode must be used with a lot of care to avoid undesired effects such as the ones you describe.
> This is in line with our recommendation in Section 6, where we suggest to rely on the manual cleaning mode to mitigate these effects and highlight important features such as minority groups.
>
> In Appendix G.3, we investigate if rare attributes of a dataset are systematically considered more likely to be irrelevant.
> As suggested, we include some demographic attributes of CheXpert but also consider skin types in DDI and Fitzpatrick17k.
> We compare the alignment of these features with the irrelevant sample ranking using AP and AUROC, similar to the comparison with metadata in Appendix G.2.
> The results show that overall no systematic bias driving underrepresented groups in the earlier rankings is present, as the average precision is very close to the non-informed baseline i.e. the group's prevalence.
> While this is by no means conclusive proof that biases are prevented, it illustrates that at least in a few example situations, irrelevant samples can be detected without disfavoring minority groups.
>
> > Q2: We can see on Table 2 that the gains are higher with kNN than with linear evaluation with a clean train set. This seems to be because kNN eval is directly based on distance, such as your filtering method. Given that zero shot evaluation of CLIP is based on the distance between the text and image encoder, it is possible that your method can enhance zero-shot evals of CLIP model. Could you test it in a zero-shot setup such as CLIP image/text cosine similarity matching ?
>
> We appreciate the valuable suggestion and agree that this experiment would be very interesting.
> However, considering the other experiments to be performed during the rebuttal phase, we think we will be able to carry out this analysis only after the discussion period.
> Furthermore we would like to point out that, although kNN is more strongly influenced by our cleaning as it is also distance-based, it is a standard evaluation protocol for measuring the quality of latent spaces.
> Please also see our comment to W5.1 of reviewer zncP.
>
> **Weaknesses:**
>
> > W1: [...] It is not possible to control the hierarchy in term of distance between factors. Meaning that if an image is a sketch of a dog, it is hard to tell if it will match more a real picture of a dog or a sketch of a cat using self-supervised features.
>
> We agree with this comment.
> Indeed, in a related project, we observe that pictograms of dermatological diseases can be used to retrieve real skin lesion images by exploiting the learned embedding space.
> However, we would like to stress that in our case the hierarchy is built in the specific context of the dataset at hand.
> Thus, we expect samples that are easily distinguishable from the majority to be separated from them by a greater distance.

---

### Author Response · Authors · 2023-11-15
**Response to All Reviewers**

We thank all reviewers for their attentive read of our original submission,
and for their comments which greatly improved our manuscript.

We are encouraged that they recognize data cleaning as a topic that is important (psYA), prominent, and always **relevant** (zncP).
It gives us confidence to know that our approach to the problem agrees with their **intuition** (zncP) and to see the ambition of this research effort recognized (psYA).
Likewise, we are grateful for the appreciation of our method's **simplicity**, its independence of labels, and its robustness to hyperparameter changes (97xm).
It is great to hear that the **evaluation** setup is good despite the fact that data cleaning is often an ill-posed problem.
Reviewers also acknowledge that the submission contains a lot of ablation studies and details (97xm, psYA) and consider our experiments as **thorough** and thoughtful (psYA).
Last but not least, we are very happy that our submission has been praised as an interesting read (psYA) with **clear** explanations (97xm) and easy to go through (zncP).

We also appreciate the valuable and constructive critique of our work.
We **carefully considered all points** the reviewers raised and addressed them in this discussion by providing details or running additional experiments.
Specifically, comments led to the following improvements:

- Extension of the comparison with metadata to all considered competitive approaches to give a more complete picture of the performance in natural settings.
- Inclusion of "NoiseRank'' for label error detection.
- Investigation of the impact of irrelevant sample ranking for underrepresented groups, such as rare skin type or ethnicity.
- Assessment of irrelevant sample detection for an additional synthetic noise type consisting of strong blurring.
- Extension of the influence of cleaning to fine-tuned setting.

Should additional details, explanations, or clarifications be needed, we will be happy to provide them.

---

> ### Author Response · Authors · 2023-11-18
> **Updated Version of Submission**
>
> Dear reviewers,
>
> We have just uploaded an updated version of our submission, which includes:
> - An additional experiment on further synthetic augmentation mimicking out-of-focus pictures with strong Gaussian blur (see Appendix G.6), as requested by reviewer zncP.
> - A more complete picture of the impact of cleaning on model performance thanks to evaluation on downstream tasks with fine-tuned models (see Table 2 and Appendix G.4), as requested by reviewer zncP. Results are currently only available for DDI, HAM10000, and Fitzpatrick17k and will be added for Food-101N and ImageNet-1k in the next few days.
> - An improved formulation of Appendix J on the score for irrelevant samples, as requested by reviewer psYA.
>
> In case these revisions do not sufficiently address your points, we encourage you to mention it and, if possible, specify alternative suggestions.
> If instead we managed to effectively answer at least some of them, we kindly ask you to consider increasing your score.

---

> > ### Author Response · Authors · 2023-11-22
> > **Updated Version of Submission**
> >
> > Dear reviewers
> >
> > We have uploaded an updated version of our paper, which includes:
> > - A quantification of the **annotation effort saved** in a synthetic setting (see Appendix G.7), in response to reviewer psYA.
> > - An additional comparison of SelfClean with competitive approaches on **natural images from STL-10** augmented with mixed synthetic noise (see Appendix G.8), in response to reviewer zncP.
> > - A shift of the additional (so far incomplete) experiment on the impact of cleaning, requested by reviewer zncP, to a reviewer reply since its non-standard settings may be confusing for readers.
> >
> > For details, please consult the individual replies, referenced sections of the manuscript, or feel free to ask.
> > We are looking forward to your feedback on the revised manuscript and on the updated results of our experiments.

---

### Meta-Review · Area_Chair_2Zfb · 2023-12-17

**Metareview:**

This work presents a method that aims to automatically remove issues in datasets that could hinder their practical use. Specifically the authors aim to "clean" the data of mislabeled, duplicate, and "irrelevant" data. The cleaning is based on training a latent space, the distances within are used to identify distribution outliers and duplicates. The authors perform a number of ablation studies and apply their method to a number of cases, which presents a number of strengths to their work that were appreciated by the reviewers. The reviewers, however, raised a number of weaknesses as well, including concerns of novelty, utility, and generalizability in terms of the specific definitions of what does and does not constitute "clean" data. The authors provided a number of clarifications and responses during the discussion period, which I believe clarify some of, but does not address all of the points. In particular there are a number of thorny issues such as what constitutes an irrelevant data-point, a better distinction of the method from past methods to explain the authors novelty claims, and how the choices in pre-trained networks etc. might bias the results. I feel that given more time the authors can address these, but as of now I believe that this work could greatly benefit from increased clarity of presentation and incorporating these changes for a future submission.

**Justification For Why Not Higher Score:**

Numerous issues including:
 - Novelty clarification
 - Clarity on definitions
 - Exploring inherent biases

**Justification For Why Not Lower Score:**

N/A

---

### Decision · Program_Chairs · 2024-01-16

Reject